# Meta Learning for Support Recovery of High-Dimensional Ising Models

**Huiming Xie**                                              *xie339@purdue.edu*
*Department of Statistics*
*Purdue University*

**Jean Honorio**                                    *jhonorio@unimelb.edu.au*
*School of Computing and Information Systems*
*The University of Melbourne*

**Reviewed on OpenReview:** *`https://openreview.net/forum?id=n4OEWwis1j`*

## Abstract

In this paper, we consider the meta learning problem for estimating the graphs associated with high-dimensional Ising models, using the method of $\ell_1$-regularized logistic regression for neighborhood selection of each node. Our goal is to use the information learned from the auxiliary tasks in the learning of the novel task to reduce its sufficient sample complexity. To this end, we propose a novel generative model as well as an improper estimation method. In our setting, all the tasks are *similar* in their *random* model parameters and supports. By pooling all the samples from the auxiliary tasks to *improperly* estimate a single parameter vector, we can recover the true support union, assumed small in size, with a high probability with a sufficient sample complexity of $n = O(d^3 \log p/K)$ per task, for $K$ tasks of Ising models with $p$ nodes and a maximum neighborhood size $d$. This is very relevant for meta learning where there are many tasks $K = O(d^3 \log p)$, each with very few samples, i.e., $n = O(1)$, in a scenario where multi-task learning fails. We prove a matching information-theoretic lower bound for the necessary number of samples per task, which is $n = \Omega(d^3 \log p/K)$, and thus, our algorithm is minimax optimal. Finally, with the support for the novel task restricted to the estimated support union, we prove that consistent neighborhood selection for the novel task can be obtained with a sufficient sample complexity of $O(d^3 \log d)$. This reduces the original sample complexity of $n = O(d^3 \log p)$ for learning a single task. We also prove a matching information-theoretic lower bound of $\Omega(d^3 \log d)$ for the necessary number of samples.

## 1 Introduction

Markov random fields (MRF) are an important class of probability models that find its applications in a wide variety of fields spanning statistical physics (Ising, 1925) , social network analysis (Snell, 1980), computer vision (Geman & Geman, 1993) and natural language processing (Manning & Schutze, 1999). A Markov random field is an undirected graph where each node represents a random variable, and the graph structure carries certain assumptions about the conditional independence of these random variables. A prototypical example of Markov random field is the *Ising model*, where the random variables are discrete, and in particular, binary. Several other types of Markov random fields can be viewed as a general setting of the Ising model (Snell, 1980). Detecting statistical dependencies, which boils down to estimating the graph structure in the Ising model, is therefore a fundamentally significant problem to solve. Many efforts have been made for the *single-task* problem, among which Ravikumar et al. (2010) proved that with relatively low computational complexity, consistent model selection can be achieved with a sample size of $n = O(d^3 \log p)$ for a graph of $p$ nodes with a maximum neighborhood size $d$ by neighborhood selection for each node using $\ell_1$-regularized

logistic regression. This simple method with a theoretically supported reasonable performance has received considerable attention.

In practice, however, one may not be able to obtain as many samples for the high-dimensional settings where both $p$ and $d$ can be large. The more common situation in reality is that one has only a few samples for a task; nonetheless, there are usually *many related tasks* (each with very few samples) of a similar kind. For instance, for an Ising model problem, there could be only $n = O(1)$ samples per task for $K = O(d^3 \log p)$ tasks. Consider for instance the case of $K = 5000$ tasks but only a few 2 samples per task. Learning from related tasks has been previously considered in (Thrun & Pratt, 1998; Nichol et al., 2018). This challenge is also commonly met in the application of other machine learning algorithms (Vilalta & Drissi, 2002; Finn et al., 2017; Vanschoren, 2018). One general way to tackle this kind of difficulty is through *meta learning* (Vanschoren, 2018), or learning to learn (Lake et al., 2015), where we learn from multiple learning episodes that oftentimes cover a distribution of related tasks — a *family* of tasks, in order to gain some experience for the learning of a novel task (in that family), in the hope of reducing the sample complexity for the latter (Hospedales et al., 2022).

Meta learning has been widely used in machine learning problems to help increase sample efficiency, but a majority of prior works are experimental in nature, without theoretical guarantees (Lake et al., 2015; Lemke et al., 2015; Vinyals et al., 2016; Ravi & Larochelle, 2017; Finn et al., 2017; Snell et al., 2017; Grant et al., 2018; Yoon et al., 2018; Hospedales et al., 2022). There have been some efforts on building the theoretical foundation of meta-learning, but they only pertain generalization bounds in learning theory (Maurer, 2005; Pentina & Lampert, 2014; Amit & Meir, 2018; Denevi et al., 2018; Khodak et al., 2019; Huang et al., 2020; Tripuraneni et al., 2020; Farid & Majumdar, 2021; Chen & Chen, 2022; Guan et al., 2022) and convergence rates in optimization (Fallah et al., 2020; Finn et al., 2019; Khodak et al., 2019; Gao & Sener, 2020). In the above works, performance is only viewed in terms of risks (e.g., misclassification rate, mean squared error) and not in terms of support recovery. For Markov random fields, there have been some theoretical results for the models associated with Gaussian or sub-Gaussian random vectors, which is a continuous type of Markov random field (Zhang et al., 2021). For discrete type, or Ising model as its prototype, similar works on meta learning do not exist.

There has been some work on the *multi-task learning* problem on the Ising model (Gonçalves et al., 2015; Guo et al., 2015). Note that there is an intrinsic difference between *multi-task* learning and *meta* learning, where multi-task learning is learning one model for each of the different $K$ tasks simultaneously while for meta learning we learn a single model from different tasks for the easier learning of a novel task. The challenging situation of having many tasks but only a few samples per task (e.g., $K = 5000$ tasks each with 2 samples) would also render the multi-task learning method meaningless, as each task cannot be learned individually with these few samples per task.

To the best of our knowledge, we are the first to theoretically prove the sufficient sample complexity for the meta learning problem of the Ising model selection. For the Ising model, we follow the practice of Ravikumar et al. (2010) in using $\ell_1$-regularized logistic regression and converting the model selection problem into one of neighborhood selection. Based on this method, for the meta learning problem on Ising models, we propose a novel generative model by introducing *randomness* to the parameter vectors of the Ising models with reasonable and flexible assumptions for similarity among different tasks, which serves as a good generalization of the metadata for the Ising model, and hence our theoretical results can be applied to a wide range of distributions for the parameters in the Ising models under some mild conditions. We also propose an *improper estimation* method in the meta learning problem for Ising model selection where we pool all the samples from the auxiliary tasks together to estimate a single *common parameter vector* (see Definition 3.1), rather than estimate an individual parameter vector for each auxiliary task, and then we recover the *support union* from the single *common parameter vector*. Next, we estimate the parameter vector for the novel task by restricting its support in the estimated *support union*. While the sample complexity of multi-task learning is $n = O(d^3 \log p)$ (Guo et al., 2015), we have successfully shown that to learn the support union of all the tasks requires each task to have a sample complexity of only $n = O(d^3 \log p / K)$ per task for each of the $K$ tasks[1],

---

[1]For our result on meta learning, the sample complexity per task is $n = O(1/K)$ while for multi-task learning (Guo et al., 2015) the sample complexity per task is $n = O(1)$. Thus, for us, the more tasks the better, while for multi-task, the more tasks, the more data is needed.

and in fact the more tasks the better for solving this problem. Here $d$ is the maximum neighborhood size in the graphs of the Ising models for all tasks, and $p$ is the number of nodes in a graph, which is also the same for all tasks. Furthermore, we have proved that with the knowledge from the auxiliary tasks, learning on a novel task for neighborhood selection requires a sufficient sample complexity of only $O(d^3 \log d)$, independent of $p$, which is far less than that without restriction, $O(d^3 \log p)$ (Ravikumar et al., 2010), since we have assumed sparse graphical models in which $d \ll p$. In this sense, we have a smaller sufficient sample complexity for both auxiliary and novel tasks than those in all previous works, which have a magnitude depending on $p$, particularly in the form of $\log p$. We also show that $\Omega(d^3 \log p / K)$ samples per auxiliary task and $\Omega(d^3 \log d)$ samples for the novel task are necessary for the recovery success, which proves that our meta learning method is minimax optimal.

Finally, we want to point out that, as part of our contribution, we define a Bayesian generative model (see Definition 3.1). Bayesian meta learning has been previously studied (Grant et al., 2018; Yoon et al., 2018) but the focus is on proposing general problem-independent algorithms, in which a marginal likelihood (with respect to a common parameter) is approximated by estimating the parameter of each auxiliary task. In contrast, we avoid estimating each auxiliary task parameter. Furthermore, our main contribution is on theoretical guarantees of our procedure to efficiently recover the support union from auxiliary tasks through improper estimation, and to then estimate the parameter of the novel task, both with significantly better sample complexities.

## 2 Preliminaries

This section provides the background on the Ising model selection. The notations to be used throughout the paper is summarized in Table 1.

Table 1: Notation used in this paper.

| Notation | Description |
|---|---|
| $\text{sign}(x)$ | Sign of $x \in \mathbb{R}$ |
| $\|a\|_1$ | $\ell_1$-norm of vector $a \in \mathbb{R}^n$, i.e., $\sum_{i=1}^{n} |a_i|$ |
| $\|a\|_2$ | $\ell_2$-norm of vector $a \in \mathbb{R}^n$, i.e., $\sqrt{\sum_{i=1}^{n} a_i^2}$ |
| $\|a\|_\infty$ | $\ell_\infty$-norm of vector $a \in \mathbb{R}^n$, i.e., $\max_{i=1}^{n} |a_i|$ |
| $\|A\|_1$ | The $\ell_1$-norm of matrix $A \in \mathbb{R}^{m \times n}$, i.e., $\sum_{1 \le i \le m, 1 \le j \le n} |A_{ij}|$ |
| $\|A\|_\infty$ | $\ell_\infty$-operator-norm of matrix $A \in \mathbb{R}^{m \times n}$, i.e., $\max_{1 \le i \le m} \sum_{j=1}^{n} |A_{ij}|$ |
| $\Lambda_{\min}(A)$ | Minimum eigenvalue of matrix $A \in \mathbb{R}^{m \times m}$ |
| $\Lambda_{\max}(A)$ | Maximum eigenvalue of matrix $A \in \mathbb{R}^{m \times m}$ |
| $\|A\|_2$ | $\ell_2$-operator-norm of matrix $A \in \mathbb{R}^{m \times n}$, i.e., $\sqrt{\Lambda_{\max}(A^T A)}$ |
| $\text{supp}(a)$ | Support set of vector $a \in \mathbb{R}^p$, i.e., $\{i | a_i \ne 0, 1 \le i \le p\}$ |
| $\text{supp}(A)$ | Support set of matrix $A \in \mathbb{R}^{p \times p}$, i.e., $\{(i,j) | a_{ij} \ne 0, 1 \le i, j \le p\}$ |
| $|S|$ | Number of elements in set $S$ |
| $S^c$ | Complement of set $S$ |
| $a_S$ | Sub-vector of vector $a \in \mathbb{R}^n$ indexed by the entries in set $S$, i.e., $(a_i)_{i \in S}$ |
| $A_{S_1 S_2}$ | Sub-matrix of matrix $A^{m \times n}$ indexed by elements in $S_1 \times S_2$, i.e., $(A_{(i,j)})_{i \in S_1, j \in S_2}$ |
| $A \odot B \in \mathbb{R}^{m \times n}$ | The Hadamard product of $A, B \in \mathbb{R}^{m \times n}$, i.e., $[A \odot B]_{ij} = A_{ij} B_{ij}$ |
| $O(g(n))$ | $f(n) = O(g(n))$ if $0 \le f(n) \le K g(n)$ for some constant $0 < K < \infty$ |
| $\Omega(g(n))$ | $f(n) = \Omega(g(n))$ if $f(n) \ge K' g(n)$ for some constant $K' > 0$ |
| $\Theta(g(n))$ | $f(n) = \Theta(g(n))$ if $0 \le K' g(n) \le f(n) \le K g(n)$ for some constants $K' > 0, 0 < K < \infty$ |

### 2.1 Ising Model and Model Selection

In this paper, we focus on the Ising model, i.e., the binary pairwise Markov random fields, for which we provide a definition here.

**Definition 2.1.** *Let $X = (X_1, X_2, \ldots, X_p) \in \{-1, +1\}^p$ denote a $p$-dimensional binary random vector, and each random variable $X_s$ is associated with a vertex $s \in V$ of an undirected graph $G$ with vertex set*

$V = \{1, \ldots, p\}$ *and edge set* $E = V \times V$. *$X$ is said to form an Ising model associated with $G$ if the distribution takes the form*

$$\mathbb{P}_{\theta^*}(x) = \frac{1}{Z(\theta^*)} \exp \Big\{ \sum_{(s,t) \in E} \theta^*_{st} x_s x_t \Big\}, \tag{1}$$

*for some parameter* $\theta^*_{st} \in \mathbb{R}$, *where the partition function* $Z(\theta^*)$ *ensures that the distribution sums to one.*

A typical graphical model selection aims to infer the edge set $E$. A stronger criterion is *signed edge recovery*, i.e., to infer the edge sign vector

$$E_\pm := \begin{cases} \text{sign}(\theta^*_{st}), & \text{if } (s,t) \in E, \\ 0, & \text{otherwise.} \end{cases} \tag{2}$$

### 2.2 Neighborhood-based Logistic Regression

The signed edge recovery is equivalent to recovering, for each vertex $r \in V$, its neighborhood set $\mathcal{N}(r) := \{t \in V | (r,t) \in E\}$, together with the correct signs $\text{sign}(\theta^*_{rt})$ for all $t \in \mathcal{N}(r)$ (Ravikumar et al., 2010). Such information can be summarized as the *signed neighborhood set*

$$\mathcal{N}_\pm(r) := \{\text{sign}(\theta^*_{rt})t | t \in \mathcal{N}(r)\}, \tag{3}$$

which can be recovered naturally from the sign-sparsity pattern of the $(p-1)$-dimensional sub-vector of parameters

$$\theta^*_{\backslash r} := \{\theta^*_{ru}, u \in V \backslash r\}$$

associated with each vertex $r$. To estimate $\theta^*_{\backslash r}$, we can make use of the easily derived conditional distribution of $X_r$ given the other variables $X_{\backslash r} = \{X_t | t \in V \backslash r\}$:

$$\mathbb{P}_{\theta^*}(x_r | x_{\backslash r}) = \frac{\exp(2x_r \sum_{t \in V \backslash r} \theta^*_{rt} x_t)}{\exp(2x_r \sum_{t \in V \backslash r} \theta^*_{rt} x_t) + 1}. \tag{4}$$

The problem can thus be viewed as a logistic regression where $X_r$ is the response variable and all other variables $X_{\backslash r}$ act as the covariates. For a sparse problem, it is also natural to use the $\ell_1$-regularizer (Tibshirani, 1996). Formally stated, given $\mathfrak{X}^n_1 = \{x^{(1)}, x^{(2)}, \ldots, x^{(n)}\}$, a set of $n$ i.i.d. samples, the regularized regression problem is a convex program of the form (Ravikumar et al., 2010)

$$\min_{\theta_{\backslash r} \in \mathbb{R}^{p-1}} \ell(\theta_{\backslash r}; \mathfrak{X}^n_1) + \lambda \|\theta_{\backslash r}\|_1, \tag{5}$$

where $\ell(\theta_{\backslash r}; \mathfrak{X}^n_1) := -\frac{1}{n} \sum_{i=1}^n \log \mathbb{P}_{\theta_{\backslash r}}(x_r^{(i)} | x_{\backslash r}^{(i)})$ is the re-scaled negative log likelihood and $\lambda$ is a regularization parameter to be specified by the user. We can then use the estimate $\hat{\theta}_{\backslash r}$ to estimate the signed neighborhood according to

$$\hat{\mathcal{N}}_\pm(r) := \{\text{sign}(\hat{\theta}_{ru})u | u \in V \backslash r, \hat{\theta}_{ru} \neq 0\}. \tag{6}$$

## 3 Our Novel Generative Model and Improper Estimation Method

In this section, we introduce our novel generative model as well as our novel improper estimation method for the meta learning problem on Ising models.

### 3.1 Our Novel Generative Model for Meta Learning on Ising Models

We consider multiple Ising models whose parameters are generated randomly, which is more reasonable and flexible than the deterministic settings in previous work on multi-task learning (Guo et al., 2015). Formally, we define the following family of Ising models with random parameters:

**Definition 3.1.** *Let $X_1^{(k)}, X_2^{(k)}, \ldots, X_{n^{(k)}}^{(k)} \in \{-1, +1\}^p$ be $n^{(k)}$ i.i.d. p-dimensional random vectors for each $k \in \{1, 2, \ldots, K\}$. Let $X_{i,s}^{(k)}$ be the s-th entry of the p-dimensional vector $X_i^{(k)}$ for $1 \leq s \leq p$. Say we are given K undirected graphs with the same number of nodes p, i.e., $G^{(k)} = (V, E^{(k)})$ with the same vertex set $V = \{1, 2, \ldots, p\}$ and potentially different edge sets $E^{(k)} \subset V \times V$ for $1 \leq k \leq K$. Each random variable $X_{i,s}^{(k)}$ is associated with a vertex $s \in V$ in the k-th graph $G^{(k)}$. We say $\{X_i^{(k)}\}_{1 \leq i \leq n^{(k)}, 1 \leq k \leq K}$ forms a family of p-dimensional random Ising models of size K if, for $1 \leq k \leq K$,*

*(i) the samples $X_1^{(k)}, X_2^{(k)}, \ldots, X_{n^{(k)}}^{(k)} \overset{i.i.d.}{\sim} \mathbb{P}_{\bar{\theta}^{(k)}}$, where*

$$\mathbb{P}_{\bar{\theta}^{(k)}}(x_i^{(k)}) = \frac{1}{Z(\bar{\theta}^{(k)})} \exp\Big\{ \sum_{(s,t) \in E^{(k)}} \bar{\theta}_{st}^{(k)} x_{i,s}^{(k)} x_{i,t}^{(k)} \Big\} \tag{7}$$

*for some parameter $\bar{\theta}_{st}^{(k)} \in \mathbb{R}$ and the partition function $Z(\bar{\theta}^{(k)})$ ensuring that the distribution sums to one;*

*(ii)*

$$\bar{\theta}^{(k)} = \bar{\theta} + \Delta^{(k)}, \tag{8}$$

*with $\bar{\theta}, \Delta^{(k)} \in \mathbb{R}^{\binom{p}{2}}$, $\bar{\theta}$ deterministic and $\Delta^{(k)}, 1 \leq k \leq K$ are i.i.d random vectors drawn from distribution P; and*

*(iii) we have for $1 \leq k \leq K$,*

$$\mathbb{P}_{\Delta^{(k)} \sim P}[\text{supp}(\Delta^{(k)}) \subseteq \text{supp}(\bar{\theta})] = 1. \tag{9}$$

**Remark 3.2.** *With respect to Definition 3.1, a task k consist of: a graph $G^{(k)}$, an Ising model parameterized by $\bar{\theta}^{(k)}$ with support defined by the edge set of $G^{(k)}$, and samples $X_1^{(k)}, X_2^{(k)}, \ldots, X_{n^{(k)}}^{(k)}$ coming from the probability distribution defined by the Ising model parameterized by $\bar{\theta}^{(k)}$.*

For our meta learning problem, we have $K$ auxiliary tasks and one novel task, forming a family of $p$-dimensional random Ising models of size $K + 1$. We can refer to $\bar{\theta}$ as the *true common parameter vector* and $S := \text{supp}(\bar{\theta})$ is what we call the *true support union*. We can then understand the maximum neighborhood size $d$ for all graphs as $d := \max\{|S_1|, |S_2|, \ldots, |S_p|\}$, where $S_r$ is the neighborhood set of each node $r \in V$ in the *latent deterministic* graph parametrized by $\bar{\theta}$, defined as: $S_r := \{t \in V | (r, t) \in S\}$. We assume $d \ll p$ ($d$ is small in size compared to $p$), so that the graphs are fairly sparse. By restricting the parameter estimation of the novel task to the true support union that can be estimated using the auxiliary tasks, we can potentially reduce the sample complexity for the novel task by a large margin.

**Remark 3.3.** *Note that condition (9) restricts the support of the randomness in the parameter of each task to $\text{supp}(\bar{\theta})$, which guarantees that the support of each task $\text{supp}(\bar{\theta}^{(k)}) \subseteq \text{supp}(\bar{\theta})$ for $1 \leq k \leq K + 1$, with probability 1. For instance, for an arbitrary entry $(s, t) \in \text{supp}(\bar{\theta})$, we have two cases: for task k, if $\Delta_{st}^{(k)} = -\bar{\theta}_{st}$, then from equation (8) we have $\bar{\theta}_{st}^{(k)} = 0$, so that $(s, t) \notin \text{supp}(\bar{\theta}^{(k)})$ and $\text{supp}(\bar{\theta}^{(k)}) \subset \text{supp}(\bar{\theta})$, i.e., we get a proper subset; else if $\Delta_{st}^{(k)} \neq -\bar{\theta}_{st}$, then by the same token, we have $(s, t) \in \text{supp}(\bar{\theta}^{(k)})$. Either way we arrive at $\text{supp}(\bar{\theta}^{(k)}) \subseteq \text{supp}(\bar{\theta})$. Suppose on the contrary we do not impose condition (9), then it will be hard for us to estimate a common parameter or a support union useful for all the tasks. On the other hand, there is still great flexibility in the family of distributions since graphs from different tasks can have edge structures with no intersection with arbitrary probability, and we do not assume entries in $\Delta^{(k)}$ to be small in absolute value.*

## 3.2 Our Improper Estimation Method for Meta Learning on Ising Models

Our estimation procedure can be divided into two steps. The first step is to recover $S$ from the $K$ auxiliary tasks by estimating $\bar{\theta}$. The second step is the signed edge recovery for task $K + 1$ with its support restricted to the estimated support union. Considering the sparsity of the problem, we will use the $\ell_1$-regularized logistic regression in both steps.

**Estimating the Support Union from $K$ Tasks.** Here we *improperly* estimate a single parameter $\theta$ although we know that each of the auxiliary tasks $k = 1, \ldots, K$ has its own true parameter $\bar{\theta}^{(k)}$. Since we are only interested on the joint support of the auxiliary tasks, this improper estimation does not need to estimate $K$ parameters instead.

Specifically, for the first step, we pool all the samples from the $K$ tasks and estimate $\bar{\theta}$ by minimizing the $\ell_1$-regularized logistic loss between $\bar{\theta}$ and the estimate. For a clearer presentation, we assume that each auxiliary task has the same number of samples, i.e., $n^{(k)} = n$ for $1 \leq k \leq K$. Note that we do not assume that $n^{(K+1)} = n$ for the novel task. Then, for each node $r \in V$, given the samples from all the $K$ auxiliary tasks $\{X_i^{(k)}\}_{1 \leq i \leq n, 1 \leq k \leq K}$, for which we use a shorthand notation $\{\mathfrak{X}_1^n\}_1^K$, this regularized regression problem is a convex program of the form

$$\hat{\theta}_{\backslash r} = \arg\min_{\theta_{\backslash r} \in \mathbb{R}^{p-1}} \ell(\theta_{\backslash r}; \{\mathfrak{X}_1^n\}_1^K) + \lambda \|\theta_{\backslash r}\|_1, \tag{10}$$

where

$$\ell(\theta_{\backslash r}; \{\mathfrak{X}_1^n\}_1^K) = -\frac{1}{K}\sum_{k=1}^{K}\frac{1}{n}\sum_{i=1}^{n}\log \mathbb{P}_{\theta_{\backslash r}}(x_{i,r}^{(k)}|x_{i,\backslash r}^{(k)}) \tag{11}$$

is the averaged re-scaled negative log likelihood of all the auxiliary tasks and $\lambda > 0$ is a regularization parameter to be specified by the user, which potentially depends on $n, p, d$ and $K$. Note that (10) is an improper estimation as we estimate a single parameter vector using data from different distributions. We can further write $\ell(\theta_{\backslash r}; \{\mathfrak{X}_1^n\}_1^K) = \frac{1}{K}\sum_{k=1}^{K}\ell^{(k)}(\theta_{\backslash r}; \{\mathfrak{X}_1^n\}^{(k)})$, where $\{\mathfrak{X}_1^n\}^{(k)}$ is another shorthand notation for $\{X_i^{(k)}\}_{1 \leq i \leq n}$, and $\ell^{(k)}(\theta_{\backslash r}; \{\mathfrak{X}_1^n\}^{(k)}) := -\frac{1}{n}\sum_{i=1}^{n}\log \mathbb{P}_{\theta_{\backslash r}}(x_{i,r}^{(k)}|x_{i,\backslash r}^{(k)})$ for $1 \leq k \leq K$. With our estimate $\hat{\theta}$ for the true common parameter vector $\bar{\theta}$, we can use $\text{supp}(\hat{\theta})$ as the estimate of the true support union $S = \text{supp}(\bar{\theta})$.

**Estimating the Support of the Novel Task.** For the second step of our estimation, we aim to estimate the true parameter vector $\bar{\theta}^{(K+1)}$ with knowledge from auxiliary tasks. Suppose we have successfully recovered the true support union $S$ from the first step. Since the true support of the novel task parameter $\text{supp}(\theta^{K+1}) \subseteq S$ is assumed in the problem setting, we can perform a regularized logistic regression with an additional restriction:

$$\hat{\theta}_{\backslash r}^{(K+1)} = \arg\min_{\theta_{\backslash r} \in \mathbb{R}^{p-1}} \left\{\ell^{(K+1)}(\theta_{\backslash r}; \{\mathfrak{X}_1^{n^{(K+1)}}\}^{(K+1)}) + \lambda^{(K+1)}\|\theta_{\backslash r}\|_1\right\}$$

$$\text{s.t.} \quad \text{supp}(\theta_{\backslash r}) \subseteq \text{supp}(\hat{\theta}_{\backslash r}), \tag{12}$$

where

$$\ell^{(K+1)}(\theta_{\backslash r}; \{\mathfrak{X}_1^{n^{(K+1)}}\}^{(K+1)}) = -\frac{1}{n^{(K+1)}}\sum_{i=1}^{n^{(K+1)}}\log \mathbb{P}_{\theta_{\backslash r}}(x_{i,r}^{(K+1)}|x_{i,\backslash r}^{(K+1)}). \tag{13}$$

Here $\{\mathfrak{X}_1^{n^{(K+1)}}\}^{(K+1)} = \{X_i^{(K+1)}\}_{1 \leq i \leq n^{(K+1)}}$ denotes the $n^{(K+1)}$ samples from the $(K+1)$-th task, and $\lambda^{(K+1)}$ is the regularization parameter for the novel task, which potentially depends on $n^{(K+1)}, p, d$.

# 4 Theoretical Results

## 4.1 Assumptions

The success of our method requires some assumptions on the structure of the logistic regression, most of which are the dependency and incoherence conditions in the work by Ravikumar et al. (2010); Guo et al. (2015) generalized to our meta learning setting (see Assumptions 4.1, 4.2, 4.5, 4.6). We also make assumptions regarding the randomness in the parameters for each task, which, intuitively speaking, make the tasks similar in some sense (see Assumption 4.3).

### 4.1.1 Assumptions in Auxiliary Tasks

The assumptions for the support union estimation are stated in terms of the Hessian of the likelihood function $\mathbb{E}\{-\ell(\theta_{\backslash r}; \{\mathfrak{X}_1^n\}_1^K)\}$ evaluated at the true common parameter $\bar{\theta}_{\backslash r}$ for each node $r \in V$. More specifically, the Hessian for any fixed node $r \in V$ is a $(p-1) \times (p-1)$ matrix of the form $\bar{Q}_r := \mathbb{E}\{\frac{1}{K} \sum_{k=1}^K \nabla^2 \log \mathbb{P}_{\bar{\theta}}[X_r^{(k)} | X_{\backslash r}^{(k)}]\}$, which has an explicit expression

$$\bar{Q}_r = \frac{1}{K} \sum_{k=1}^K \mathbb{E}[\eta(X^{(k)}; \bar{\theta}) X_{\backslash r}^{(k)} (X_{\backslash r}^{(k)})^T], \tag{14}$$

where

$$\eta(u; \theta) := \frac{4 \exp(2u_r \sum_{t \in V \backslash r} \theta_{rt} u_t)}{(\exp(2u_r \sum_{t \in V \backslash r} \theta_{rt} u_t) + 1)^2}. \tag{15}$$

Here $\eta(u; \theta)$ is the variance function. Note that our expectation is taken with respect to the joint distribution of the data $\{X_i^{(k)}\}_{1 \le i \le n, 1 \le k \le K}$ and the random variables $\{\Delta^{(k)}\}_{k=1}^K$ in the parameters.

**Dependency Condition.** Following the dependency condition imposed by Ravikumar et al. (2010), we assume that the subset of the Fisher information matrix corresponding to the relevant covariates in the true support union has bounded eigenvalues. We have:

**Assumption 4.1.** *There exist constants $C_{\min} > 0$ and $D_{\max} > 0$ such that*

$$\Lambda_{\min}((\bar{Q}_r)_{S_r S_r}) \ge C_{\min}, \tag{16}$$

*for all $r \in V$ and*

$$\Lambda_{\max}\left(\frac{1}{K} \sum_{k=1}^K \mathbb{E}[X_{\backslash r}^{(k)} (X_{\backslash r}^{(k)})^T]\right) \le D_{\max}. \tag{17}$$

These assumptions make sure that the covariates are not excessively dependent.

**Incoherence Condition.** To prevent the large number of irrelevant covariates (the ones outside the support) having too strong an effect on the relevant covariates (the ones in the support), as pointed out by Ravikumar et al. (2010), the following assumption is required:

**Assumption 4.2.** *There exists an $\alpha \in (0, 1]$ such that*

$$\left\|\left\|(\bar{Q}_r)_{S_r^c S_r}((\bar{Q}_r)_{S_r S_r})^{-1}\right\|\right\|_\infty \le 1 - \alpha, \tag{18}$$

*for all $r \in V$.*

Our assumptions above, made only on the true common parameter $\bar{\theta}$ for meta-learning, are similar to assumptions made *for all tasks* in multi-task learning (Guo et al., 2015). Thus, our assumptions are less restrictive.

**Additional Assumptions on $\{\Delta^{(k)}\}_{1 \le k \le K}$.** The success of our method also relies on some reasonable and flexible assumptions on the centering of the random variables $\{\Delta^{(k)}\}_{1 \le k \le K}$ underlying the parameters of each task — reasonable in the sense that the tasks are similar enough to provide useful information, and flexible so that there is as little inductive bias as possible.

For simplicity, without writing down the samples for the auxiliary tasks, we use a shorthand notation $\nabla\ell(\bar{\theta}_{\backslash r})$ to denote the gradient of the loss function for the improper estimation, evaluated at the true common parameter vector $\bar{\theta}$; similarly $\nabla\ell^{(k)}(\bar{\theta}_{\backslash r}^{(k)})$ means the gradient of the loss function for the $k$-th auxiliary task evaluated at the true parameter for that particular task, i.e., $\bar{\theta}^{(k)}$. With a little abuse of notation in writing $\bar{\nabla}\ell := \frac{1}{K} \sum_{k=1}^K \nabla\ell^{(k)}(\bar{\theta}_{\backslash r}^{(k)})$, we have

**Assumption 4.3.** *For any $\varepsilon > 0$,*

$$\mathbb{P}\Big(\|\mathbb{E}[\nabla\ell(\bar{\theta}_{\backslash r}) - \bar{\nabla}\ell)]\|_\infty > \sqrt{\frac{8\log(2p/\varepsilon)}{nK}}\Big) \le \varepsilon, \tag{19}$$

*for all $r \in V$.*

The constants 8 and 2 above are just for ease of calculation, and can be substituted with other constants without harm. Note that $\bar{\nabla}\ell = \frac{1}{K}\sum_{k=1}^K \nabla\ell^{(k)}(\bar{\theta}_{\backslash r}^{(k)})$ is just the counterpart of $\nabla\ell(\bar{\theta}_{\backslash r})$, with the gradient of loss for each task evaluated at their own true parameters $\bar{\theta}_{\backslash r}^{(k)}$, which depend on each random $\Delta^{(k)}$, as opposed to all evaluated at the common parameter vector $\bar{\theta}_{\backslash r}$ in $\nabla\ell(\bar{\theta}_{\backslash r})$.

**Remark 4.4.** *In calculating the expectation over the joint distribution of $X$ and $\Delta$ in this assumption, $n$ and $K$ are eliminated due to the i.i.d. $\{\Delta^{(k)}\}_{1\le k\le K}$, and conditioned on which, the i.i.d. property of samples $\{X_i^{(k)}\}_{1\le i\le n,1\le k\le K}$ follows. Hence, the requirement on $K$ tasks is just one on the distribution $P$ of $\{\Delta^{(k)}\}_{1\le k\le K}$ for the family of Ising models. The assumption helps specify some symmetry for the family of distributions and is compatible with the primal-dual witness.*

**Illustrative Example.** Here we provide an illustrative example to help with intuitive comprehension: the quantity in the assumption can be written explicitly as $\|\mathbb{E}_{\Delta\sim P}\big[\mathbb{E}_{X\sim\bar{\theta}+\Delta}\big[X_{\backslash r}(\mathbb{E}_{X\sim\bar{\theta}}[X_r|X_{\backslash r}] - \mathbb{E}_{X\sim\bar{\theta}+\Delta}[X_r|X_{\backslash r}])|\Delta\big]\big]\|_\infty$. It can be checked that with a common latent graph with 3 nodes, 3 edges and $\bar{\theta} = (1,1,1)$, a setting of $\Delta$ resulting in the tasks parameter supports to have only 2 edges each with values $\bar{\theta} + \Delta \in \{(1.75, 1.75, 0), (1.75, 0, 1.75), (0, 1.75, 1.75)\}$ with equal probabilities will fulfill our condition with the desired quantity around 0. A straightforward example for higher dimensions is to have a graph with the above 3-node graph repeated multiple times, noticing that the combination of removing edges would also give rise to potentially numerous different tasks. More details of calculation and illustration can be found in Appendix C.

### 4.1.2 Assumptions in Novel Task

Our assumptions for the novel task is analogous to those for the improper estimation using auxiliary tasks, but with the parameters and the random variables restricted to the true support union $S$. We base the assumption on the Hessian of the likelihood function $\mathbb{E}\{-\ell^{(K+1)}(\theta_S; \{\mathfrak{X}_1^{n^{(K+1)}}\}_S^{(K+1)})\}$ evaluated at the true parameter for the $(K+1)$-th task, $\bar{\theta}_S^{(K+1)}$:

$$\bar{Q}_r^{(K+1)} := \mathbb{E}\{\nabla^2\log\mathbb{P}_{\bar{\theta}_S^{(K+1)}}[X_r^{(K+1)}|X_{S_r}^{(K+1)}]\}.$$

This is given as the explicit expression

$$\bar{Q}_r^{(K+1)} = \mathbb{E}[\eta(X_{S_r\cup\{r\}}^{(K+1)}; \bar{\theta}_S^{(K+1)})X_{S_r}^{(K+1)}(X_{S_r}^{(K+1)})^T]. \tag{20}$$

Let use $S^{(K+1)} := \text{supp}(\bar{\theta}^{(K+1)})$ to denote the true support of the $(K+1)$-th task which satisfies $S^{(K+1)} \subseteq S$. Similarly, we define $S_r^{(K+1)} := \{t \in V|(r,t) \in S^{(K+1)}\}$. We again follow the assumptions made by Ravikumar et al. (2010):

**Assumption 4.5** (Dependency Condition). *There exist constants $C_{\min}^{(K+1)} > 0$ and $D_{\max}^{(K+1)} > 0$ such that*

$$\Lambda_{\min}((\bar{Q}_r^{(K+1)})_{S_r^{(K+1)}S_r^{(K+1)}}) \ge C_{\min}^{(K+1)}, \tag{21}$$

*for all $r \in V$ and*

$$\Lambda_{\max}(\mathbb{E}[X_S^{(K+1)}(X_S^{(K+1)})^T]) \le D_{\max}^{(K+1)}. \tag{22}$$

**Assumption 4.6** (Incoherence Condition). *There exists an $\alpha^{(K+1)} \in (0,1]$ such that*

$$\left\|\left\|(\bar{Q}_r^{(K+1)})_{([S_r^{(K+1)}]^c \cap S)S_r^{(K+1)}}((\bar{Q}_r^{(K+1)})_{S_r^{(K+1)}S_r^{(K+1)}})^{-1}\right\|\right\|_\infty \leq 1 - \alpha^{(K+1)}, \tag{23}$$

*for all $r \in V$.*

Our assumptions above, made only on novel task parameter $\bar{\theta}^{(K+1)}$ for meta-learning, are similar to assumptions made *for all tasks* in multi-task learning (Guo et al., 2015). Thus, our assumptions are less restrictive.

### 4.2 Main Theorems

#### 4.2.1 Support Union Recovery

Our first theorem demonstrates that the sufficient sample complexity for the recovery of the true support union $S$ by our estimator in (10) is $n = O(d^3 \log p / K)$ per task for each of the $K$ tasks. This means that for the situation with numerous tasks $K = O(d^3 \log p)$, the sufficient sample complexity per task is as small as $O(1)$. From the condition we obtained on the regularizing parameter $\lambda$, we can also see that having more tasks will give a good estimate of the support union with less penalty, without having to increase the number of samples per task — the more tasks the better in this case.

**Theorem 4.7.** *For a family of $p$-dimensional random Ising models of size $K$ described in Definition 3.1 with $n^{(k)} = n$ for $1 \leq k \leq K$, suppose Assumptions 4.1, 4.2, 4.3 are satisfied. Let $\{\mathfrak{X}_1^n\}^{(k)}$ be a set of $n$ i.i.d. samples from the model specified by $\bar{\theta}^{(k)}$, and $\{\mathfrak{X}_1^n\}_1^K$ denote all the samples from the $K$ tasks. Suppose that the regularization parameter $\lambda$ is selected to satisfy $\lambda \geq \beta \sqrt{\frac{\log p}{nK}}$ for some constant $\beta > 0$, then there exists a positive constant $L$, independent of $(n, p, d, K)$, such that if $nK > Ld^3 \log p$, then, for estimating the Ising model with the true common parameter $\bar{\theta}$, for some constant $c > 0$, the following properties hold with probability at least $1 - O(\exp(-c\lambda^2 nK))$.*

*(a) For each node $r \in V$, the $\ell_1$-regularized logistic regression (10), given data $\{\mathfrak{X}_1^n\}_1^K$, has a unique solution, and so uniquely specifies a signed neighborhood $\hat{\mathcal{N}}_\pm(r)$.*

*(b) For each $r \in V$, the estimated signed neighborhood $\hat{\mathcal{N}}_\pm(r)$ correctly excludes all edges not in the true neighborhood, so that $\text{supp}(\hat{\theta}) \subseteq \text{supp}(\bar{\theta})$. Moreover, it correctly includes all edges $(r, t)$ for which $|\bar{\theta}_{rt}| \geq \frac{10}{C_{\min}}\sqrt{d}\lambda$, along with their correct sign.*

*Proof sketch for Theorem 4.7.* We use the primal-dual witness approach (Wainwright, 2009; Ravikumar et al., 2010) and the proof can be divided into two parts. The first part shows that imposing the dependence and incoherence assumptions (Assumptions 4.1 and 4.2) on the *population version* of the Fisher information matrix $\bar{Q}$ guarantees (with high probability) that analogous conditions hold for the *sample Fisher information matrix* $Q^N := \hat{\mathbb{E}}[-\nabla^2 \ell(\bar{\theta}_{\backslash r}; \{\mathfrak{X}_1^n\}_1^K)]$ (see (38) in Appendix); the second part of the proof is devoted to show that if the dependence condition and incoherence condition are imposed on the *sample Fisher information matrix* $Q^N$, then the growth condition and choice of $\lambda$ from Theorem 4.7 are sufficient to ensure that the graph associated with the true common parameter vector is recovered with high probability (Ravikumar et al., 2010).

The first part of the proof mainly use techniques such as norm inequalities and union bounds (Hoeffding, 1994; Ravikumar et al., 2010) to get a high probability $1 - O(\exp(-b\frac{nK}{d^3} + \log p))$ for some constant $b > 0$, which in turn yields the growth condition in on number of samples per task $n$ and number of tasks $K$.

In the second part, the key is to verify the strict dual feasibility (Rockafellar, 2015). Using some norm inequalities and following a method used in a different context (Rothman et al., 2008; Ravikumar et al., 2010), we show that it suffices to bound the random term $\|\nabla\ell(\bar{\theta}_{\backslash r}; \{\mathfrak{X}_1^n\}_1^K)\|_\infty = \|\frac{1}{K}\sum_{k=1}^K \nabla\ell^{(k)}(\bar{\theta}_{\backslash r}; \{\mathfrak{X}_1^n\}_1^K)\|_\infty$,

which we decompose into two parts as follows

$$\|\nabla\ell(\bar{\theta}_{\backslash r}; \mathfrak{X}_1^n\}_1^K)\|_\infty \leq \| \underbrace{\frac{1}{K}\sum_{k=1}^K \left\{ \nabla\ell^{(k)}(\bar{\theta}_{\backslash r}; \{\mathfrak{X}_1^n\}^{(k)}) - \nabla\ell^{(k)}(\bar{\theta}_{\backslash r}^{(k)}; \{\mathfrak{X}_1^n\}^{(k)}) \right\}}_{Y_1} \|_\infty$$

$$+ \| \underbrace{\frac{1}{K}\sum_{k=1}^K \nabla\ell^{(k)}(\bar{\theta}_{\backslash r}^{(k)}; \{\mathfrak{X}_1^n\}^{(k)})}_{Y_2} \|_\infty. \quad (24)$$

Using Hoeffding's Inequality with latent conditional independence (LCI) (Ke & Honorio, 2019) with latent variables $\{\Delta^{(k)}\}_{k=1}^K$, $\|Y_2\|_\infty$ can be bounded with high probability in the sense that

$$\mathbb{P}[\|Y_2\|_\infty > \delta] \leq 2\exp\left(-\frac{\delta^2 nK}{8} + \log p\right). \quad (25)$$

Also note that $Y_1 = \nabla\ell(\bar{\theta}_{\backslash r}) - \frac{1}{K}\sum_{k=1}^K \nabla\ell^{(k)}(\bar{\theta}_{\backslash r}^{(k)})$ using the shorthand notations in (19). We can then bound $\|Y_1\|_\infty$ by writing

$$\|Y_1\|_\infty = \|Y_1 - \mathbb{E}(Y_1) + \mathbb{E}(Y_1)\|_\infty \leq \|Y_1 - \mathbb{E}(Y_1)\|_\infty + \|\mathbb{E}(Y_1)\|_\infty. \quad (26)$$

Using Assumption 4.3, it is not hard to derive that

$$\mathbb{P}(\|\mathbb{E}(Y_1)\|_\infty \geq \delta) \leq 2\exp\left(-\frac{\delta^2 nK}{8} + \log p\right). \quad (27)$$

For the term $\|Y_1 - \mathbb{E}(Y_1)\|_\infty$, we can get the same rate of decay with respect to $(n, p, d, K)$ by using LCI Hoeffding's inequality (Ke & Honorio, 2019) again. Then applying union bounds and setting $\delta$ to be $\lambda$ times a constant, we get the rate $O(\exp(-c\lambda^2 nK))$ as in Theorem 4.7, as well as the condition on $\lambda$. The detailed proof is in Appendix D. $\qquad\square$

We also prove the following information-theoretic lower bound on the failure of support union recovery for some family of Ising models.

**Theorem 4.8.** *For some family of $p$-dimensional Ising models of size $K$ with parameters $\{\bar{\theta}^{(k)}\}_{k=1}^K$, suppose $p \geq 5$, $\bar{\theta}^{(k)} = H \odot \Gamma^{(k)}$ for $1 \leq k \leq K$ with $\Gamma^{(k)} \in [-1/d^4, 1/d^4]^{p\times p}$ symmetric, degree $d \in \mathbb{Z}^+$ even and $H \in \{0,1\}^{p\times p}$ such that $H$ is symmetric and $H_{ij} = 1$ iff $(i,j) \in E$. Thus $S := E$ is the support union of the $K$ Ising models. Assume $E$ is randomly generated in the following way:*

*(i) Obtain a permutation $\pi = (\pi_1, \pi_2, \ldots, \pi_p)$ of $V = \{1, 2, \ldots, p\}$ uniformly at random.*

*(ii) Let $\pi_{p+j} := \pi_j$ for $1 \leq j \leq d/2$*

*(iii) For $i = 1, \ldots, p$, add $(\pi_i, \pi_{i+j})$ to $E$ for $1 \leq j \leq d/2$.*

*Thus $d$ is the degree of the graphs in all tasks. Suppose that for each of the $K$ distributions, we have $n$ samples randomly drawn from them. Then for any estimate $\hat{S}$ of $S$, we have*

$$\mathbb{P}\{\hat{S} \neq S\} \geq 1 - \frac{2npK/d^3 + \log 2}{p\log p - p - \log 2p}. \quad (28)$$

The detailed proof is in Appendix E.

According to Theorem 4.8, if the sample size per distribution is $n \leq (d^3 \log p)/(4K) - d^3/(4K) - (d^3 \log(8p))/(4pK)$, then with probability larger than $1/2$, any method will fail to recover the support union of the Ising models specified in Theorem 4.8. Thus a sample complexity of $\Omega(d^3 \log p/K)$ per task is necessary for the support union recovery of the $p$-dimensional Ising models in $K$ tasks, which, combined with Theorem 4.7, indicates that our estimate (10) is minimax optimal with a necessary and sufficient sample complexity of $\Theta(d^3 \log p/K)$ per task.

### 4.2.2 Support Recovery for Novel Task

For the novel task, the following theorem provides the sufficient conditions and a probability lower bound for the sign-consistency of the estimate, from which we can conclude that using the knowledge learned from the auxiliary tasks, the consistent signed neighborhood selection for the novel task can be achieved in a sufficient sample complexity of $n^{(K+1)} = O(d^3 \log d)$.

**Theorem 4.9.** *Suppose we have recovered the true support union $S$ of a family of $p$-dimensional random Ising models of size $K$ described in Definition 3.1. Assume $|S| = O(d)$. For a novel task with parameter $\bar{\theta}^{(K+1)}$ such that $\mathrm{supp}(\bar{\theta}^{(K+1)}) \subseteq S$ and satisfying Assumptions 4.5, 4.6, suppose the regularization parameter is chosen such that $\lambda^{(K+1)} \geq \beta\sqrt{\frac{\log d}{n^{(K+1)}}}$ for some constant $\beta > 0$, then there exists a positive constant $L$, independent of $(n^{(K+1)}, p, d)$, such that if $n^{(K+1)} > Ld^3 \log d$, then for some constant $c > 0$, the following properties hold with probability at least $1 - O(\exp(-c(\lambda^{(K+1)})^2 n^{(K+1)}))$.*

*(a) For each node $r \in V$, the $\ell_1$-regularized logistic regression for estimating $\bar{\theta}^{(K+1)}$ in the novel task, given data $\{\mathfrak{X}_1^{n^{(K+1)}}\}^{(K+1)}$ has a unique solution $\hat{\theta}^{(K+1)}$, and so uniquely specifies a signed neighborhood $\hat{\mathcal{N}}_{\pm}^{(K+1)}(r) := \{\mathrm{sign}(\hat{\theta}_{ru}^{(K+1)})u | u \in V \setminus r, \hat{\theta}_{ru}^{(K+1)} \neq 0\}$*

*(b) For each $r \in V$, the estimated signed neighborhood $\hat{\mathcal{N}}_{\pm}^{(K+1)}(r)$ correctly excludes all edges not in the true neighborhood $\mathcal{N}_{\pm}^{(K+1)}(r) := \{\mathrm{sign}(\bar{\theta}_{ru}^{(K+1)})u | u \in V \setminus r, \bar{\theta}_{ru}^{(K+1)} \neq 0\}$. Moreover, it correctly includes all edges with $|\bar{\theta}_{rt}^{(K+1)}| \geq \frac{10}{C_{\min}^{(K+1)}}\sqrt{d}\lambda^{(K+1)}$, along with their correct sign.*

**Remark 4.10.** *Note that the constants $\beta, L$ and $c$ we use in the theorems are just some general constants for convenience of notation. The ones used in Theorem 4.7 are not related to those in Theorem 4.9.*

*Proof sketch for Theorem 4.9.* We use the primal-dual witness approach again (Wainwright, 2009; Ravikumar et al., 2010). We have supposed that we have recovered the true support union $S$ from our estimate for the true common parameter, $\hat{\theta}$. The constraint in (12) then enables us to convert the problem into one without the restriction and with a parameter and data of dimension $|S_r|$ with $|S_r| \leq d$ for all $r \in V$, for we can combine the constraint straightforward into the minimization problem.

$$(\hat{\theta}_{\setminus r}^{(K+1)})_{S_r} = \arg\min_{\theta_{S_r} \in \mathbb{R}^{|S_r|}} \left\{ \ell^{(K+1)}(\theta_{S_r}; \{\mathfrak{X}_{1,S}^{n^{(K+1)}}\}^{(K+1)}) + \lambda^{(K+1)}\|\theta_{S_r}\|_1 \right\}, \qquad (29)$$

and $(\hat{\theta}_{\setminus r}^{(K+1)})_{S_r^c} = 0$. In this way, we solve a convex program analogous to (5), which is learning on a single Ising model, but with dimension reduced from $p$ to $|S_r|$, and consequently to $d$. The detailed proof is in Appendix F. $\square$

We also prove the following information-theoretic lower bound for the failure of support recovery for some Ising model where the support set is a subset of a known set $S$.

**Theorem 4.11.** *For $n$ samples generated from some $p$-dimensional Ising model, suppose the true parameter matrix is $\bar{\theta} = H \odot \Gamma$ with $\Gamma \in [-\frac{1}{d^3 \log d}, \frac{1}{d^3 \log d}]^{p \times p}$ symmetric and $H \in \{0,1\}^{p \times p}$ such that $H$ is symmetric and $H_{ij} = 1$ iff $(i,j) \in E^{(K+1)}$. Thus $S^{(K+1)} := E^{(K+1)}$ is the support set of the Ising model. Assume $E^{(K+1)}$ is chosen uniformly at random from the edge set family $\mathcal{E} := \{E \subseteq S : (i,j) \in E \implies (j,i) \in E\}$ for a known edge set $S$. Assume $|S| = O(d)$. Then for any estimate $\hat{S}^{(K+1)}$ of $S^{(K+1)}$, we have*

$$\mathbb{P}\{\hat{S}^{(K+1)} \neq S^{(K+1)}\} \geq 1 - \frac{4n}{(\log 2)(d^3 \log d)} - \frac{2}{d}. \qquad (30)$$

The detailed proof is in the Appendix G.

According to Theorem 4.11, if $n \leq \frac{\log 2}{8}d^3 \log d - \frac{\log 2}{2}d^2 \log d$, then $\mathbb{P}\{S^{(K+1)} \neq \hat{S}^{(K+1)}\} \geq \frac{1}{2}$, which indicates that the necessary sample complexity for the support recovery of the novel task is $\Omega(d^3 \log d)$ and our estimate (12) is minimax optimal. Therefore, our two-step meta learning method is minimax optimal.

## 5 Experiments

**Synthetic Experiments.** To help illustrate and validate our theories, we conduct a group of synthetic experiments and report the success rates (over 100 repetitions) for recovery of the true support union. We run the experiments with different number of nodes $p \in \{50, 100, 200\}$ and degree $d = 3$, as well as with $p = 50$ nodes and three different degrees $d \in \{3, 5, 7\}$. We set the number of tasks scaling as $K = d^3 \log p$, with sample size for each auxiliary task $n = Cd^3 \log p/K$ for $C$ ranging from 1 to upwards of 200. Then based on the estimated support union using $C = 200$, we use different sample sizes $n^{(K+1)} = C'd^3 \log d$ for the novel task when $C'$ changes from 1 to 200 and calculate the success rates (over 100 repetitions) for signed edge recovery of the novel task. We plot the success rates against $C$ and $C'$ for the two steps respectively in Figure 1 and Figure 2. The curves approximately lie on top of one another as the success rates tend to 1 in each step, as predicted by Theorem 4.7 and 4.9. Our results compare favorably against alternative methods. See Appendix B.1 for more details.

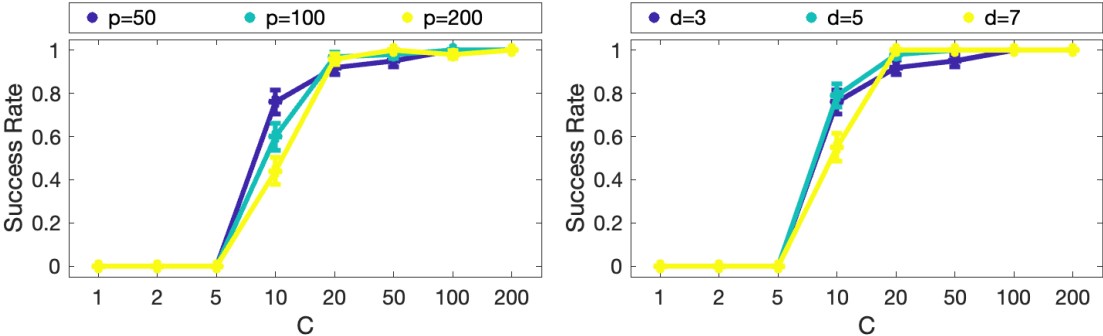

Figure 1: The success rates (over 100 repetitions) for support union recovery vs. the choice of $C$, for different number of nodes $p$ (left) and different degrees $d$ (right).

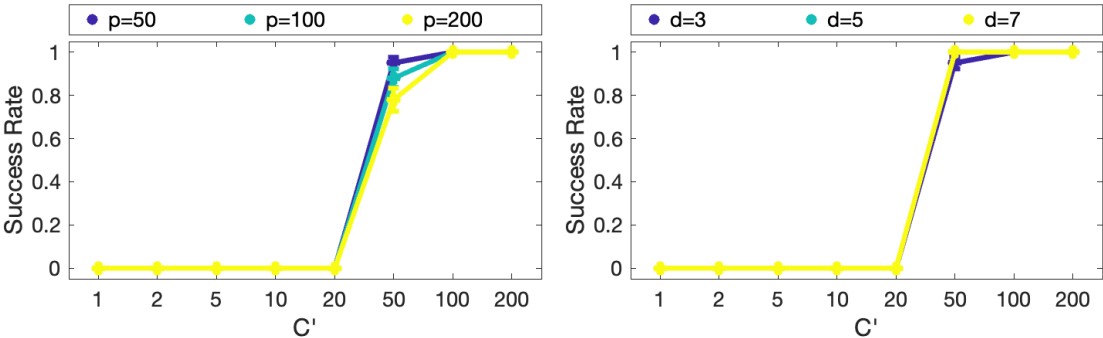

Figure 2: The success rates (over 100 repetitions) for signed edge recovery for novel task restricted to the estimated support union based on $C = 200$ vs. the choice of $C'$, for different number of nodes $p$ (left) and different degrees $d$ (right).

**Real-world Data Experiments.** As another motivation and validation of our method, we used the real-world dataset "1000 Functional Connectomes" at http://www.nitrc.org/projects/fcon_1000/ from 1128 subjects, 41 sites worldwide, and $p = 157$ brain regions. Each task comes from a different research lab with different magnetic resonance imaging scanners/equipment with different physical properties. There are labs around the world that collect MRI from different subsets of subjects/persons — one lab is a task, one brain region is a variable/node. Exploratory research was performed to unveil which brain region interactions (edges in the graph) are important. Similarly, different labs might not have the same underlying true graph (support), but it is reasonable to believe that they have some similarities. Specially, because it is partially

known that the "default mode network" is a graph that works when people are awake but resting — this data is resting state functional magnetic resonance imaging. We estimated the support union from $K = 40$ auxiliary tasks with precision 0.8821, recall 0.8918 and F1-score 0.8869. We used task 41 as the novel task, which support was estimated with precision 0.7133, recall 0.5528 and F1-score 0.6228. Both F1-score results are better than the ones from comparison methods. See Appendix B.2 for more details, and for interpretation of certain inter and intra symmetry between the left and right side of the brain.

## 6 Concluding Remarks

Our method and analysis in this paper can be extended to more general cases of Markov random fields. Since logistic regression can be generalized to multi-class logistic regression, the analysis performed on the meta learning problem for Ising model can find its analog in multi-class discrete Markov random fields (Ravikumar et al., 2010). Some other interesting directions for future work based on our method include but not limited to meta learning for general continuous Markov random fields, meta learning for graphical models with hyper-edges that can connect multiple nodes instead only two, or time-varying graphical models, etc. We believe our results can provide a solid foundation and open a novel perspective for meta learning in Markov random fields and related graphical models.

## Broader Impact

This theoretical work does not present any foreseeable societal consequence.

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
