## A   Summary of Shorthand Notations Used in the Appendix

In this appendix, we simplify the notation $\bar{Q}_r$ as $\bar{Q}$, and $(\bar{Q}_r)_{S_r S_r}$ as $\bar{Q}_{SS}$, since the reference node $r$ is used throughout the analysis and should be understood implicitly. We follow the same practice for similar shorthand notations in most part of the appendix to lighten the notations a little bit. This should not cause confusion since the elements in $S$ are pairs of nodes (two dimensional), while those in $S_r$ are individual nodes (one dimensional). Simlarly, we also write $\bar{Q}_r^{(K+1)}$ as $\bar{Q}^{(K+1)}$, and $(\bar{Q}_r^{(K+1)})_{S_r^{(K+1)} S_r^{(K+1)}}$ as $\bar{Q}_{S^{(K+1)} S^{(K+1)}}^{(K+1)}$. Finally, we write $\bar{Q}_{([S^{(K+1)}]^c \cap S) S^{(K+1)}}^{(K+1)}$ as $\bar{Q}_{[S^{(K+1)}]^c S^{(K+1)}}^{(K+1),S}$.

## B   Details of Experiments

### B.1   Synthetic Experiments

Given fixed values of $p$ and $d$, we simulate sparse random graphs by first randomly choosing whether an edge exists or not with a probability of $\frac{d}{p-1}$. At the end we check if the maximum neighborhood size $d$ is satisfied; if not, we redo the generating process until we get a random graph with maximum degree $d$. For the non-zero edge values, we use *mixed couplings* (Ravikumar et al., 2010), that is, each existent edge (edge in the true support union in our case) has value $\theta_{st} = \pm 0.5$ with equal probability. Then, to generate the random parameter of each task: for $1 \le k \le K+1$ and $(s,t) \in S$, we set $\bar{\theta}_{st}^{(k)} = \bar{\theta}_{st} X_{st}^{(k)}$ with $X_{st}^{(k)} \overset{i.i.d.}{\sim}$ Bernoulli(0.9). For the samples, we use Gibbs sampling (Casella & George, 1992) with 10 iterations to generate each $p$-dimensional data sample for the binary node values according to the specific distributions of Ising models (see (7)) using our simulated parameter values. Under each setting of the $(p, C)$ pair, we run the experiment 100 times to record whether or not it successfully recovers the neighborhood sets, and take the average of these 100 repetitions to calculate the success rate $\hat{\mathbb{P}}[\hat{\mathcal{N}}(r) = \mathcal{N}(r)]$. The regularization parameter $\lambda$ in the improper estimation (10) is set to be a constant factor of $\sqrt{\frac{\log p}{nK}}$ as suggested by Theorem 4.7. Here the constant factor is set to 1 by default, which works well. With $\lambda^{(K+1)}$ a constant factor (i.e., 1) of $\sqrt{\frac{\log d}{n^{(K+1)}}}$ in the restricted estimation (12), we then estimate the novel task parameter 100 times for $n^{(K+1)} = C' d^3 \log(d)$ with different values for $C'$, where the success rate for the novel task include sign information, i.e., it is calculated as $\hat{\mathbb{P}}[\hat{\mathcal{N}}_\pm^{(K+1)}(r) = \mathcal{N}_\pm^{(K+1)}(r)]$ over the 100 repetitions.

**More on Comparison.**   For learning the support union, we tried multi-task method of (Guo et al., 2015). We then joined all supports from each task. We show the results on Figure 3. Compared with our Figure 1 in our paper, we observe that multi-task learning fails to estimate the support union, with such few samples per task.

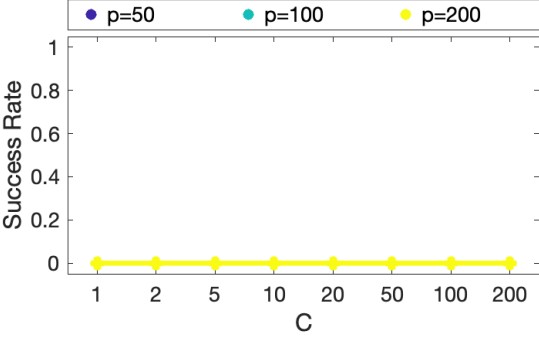

Figure 3: The success rates (over 100 repetitions) support union recovery vs. the choice of $C$ for multi-task learning (Guo et al., 2015) as a comparison to our method in Figure 1.

For estimating the novel task parameter, we evaluated the single-task method of (Ravikumar et al., 2010) on the novel task data only, using the same number of samples as in our experiments, and produced a resulting

plot here in Figure 4(left). Compared with our Figure 2 in our paper, we can see that given such few samples, the alternative method cannot succeed at learning, giving near zero success rates. We also tried pooling all data from auxiliary tasks together with the novel task, and using the single-task method of (Ravikumar et al., 2010). We report the results on Figure 4(right). While this method might be reasonably good for estimating the support union, it fails for estimating the correct signs and support of the novel task. This is due to the fact that the support of the novel task is a subset of the support union.

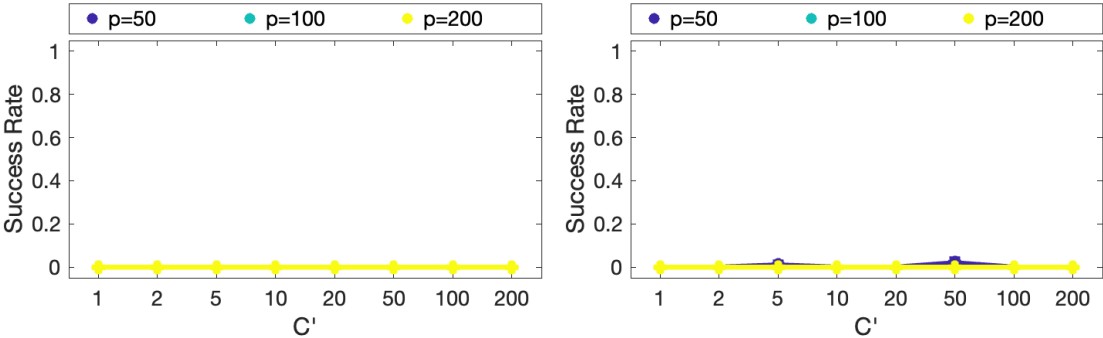

Figure 4: The success rates (over 100 repetitions) for signed edge recovery for novel task vs. the choice $C'$ for the single-task method of (Ravikumar et al., 2010) using novel task data only (left) and using all data from auxiliary tasks and novel task (right). Both serve as a comparison to our method in Figure 2.

## B.2 Real-world Data Experiments

For the real-world data experiment, the sample sizes for each individual task range from 300 to 4374, with an average size of around 1553 and standard deviation 914. We have an independent set with 68259 samples to retrieve the "true" support union as well as the "true" novel task support. When running the algorithm for support union recovery, we used 40 tasks. We used task 41 as the novel task. The constant factor in $\lambda$ was tuned to be 2 to get reasonably sparse graphs $d = 19$ compared to the number of nodes $p = 157$.

**More on Comparison.** We validated our results with comparison methods. For learning the support union, we tried multi-task method of (Guo et al., 2015). We then joined all supports from each task. This method obtained a precision 0.3916, recall 0.9938 and F1-score 0.5619, versus our F1-score of 0.8869. For the novel task, we tried the single-task method of (Ravikumar et al., 2010). This method obtained a precision 0.8170, recall 0.3472 and F1-score 0.4873, versus our F1-score of 0.6228. We also tried pooling all data from auxiliary tasks together with the novel task, and using the method of (Ravikumar et al., 2010). This method obtained a precision 0.2402, recall 0.9889 and F1-score 0.3865, versus our F1-score of 0.6228.

**Interpretation of Support Union.** In the data of Functional Connectomes for our real data experiments, we found that the support union shows some nice inter and intra symmetry between the left and right side of the brain. For inter symmetry, Broadmann areas in the left side of the brain interact similarly as the Broadmann areas in the right side of the brain (see Figure 5). For intra symmetry: One Broadmann area in the left is most likely to interact with its corresponding Broadmann area in the right (see Figure 6). This shows that estimating the support union is important as it reduces the search space for the novel task graph a lot in the real-world case.

## C Illustrative Example

To verify that Assumption 4.3 can be satisfied for a large family of distributions, we provide an illustrative example to demonstrate its viability. The infinity norm in the assumption can be written explicitly as

$$\|\mathbb{E}_{\Delta \sim P}\left[\mathbb{E}_{X \sim \bar{\theta}+\Delta}\left[X_{\setminus r}(\mathbb{E}_{X \sim \bar{\theta}}[X_r | X_{\setminus r}] - \mathbb{E}_{X \sim \bar{\theta}+\Delta}[X_r | X_{\setminus r}])|\Delta\right]\right]\|_{\infty}. \tag{31}$$

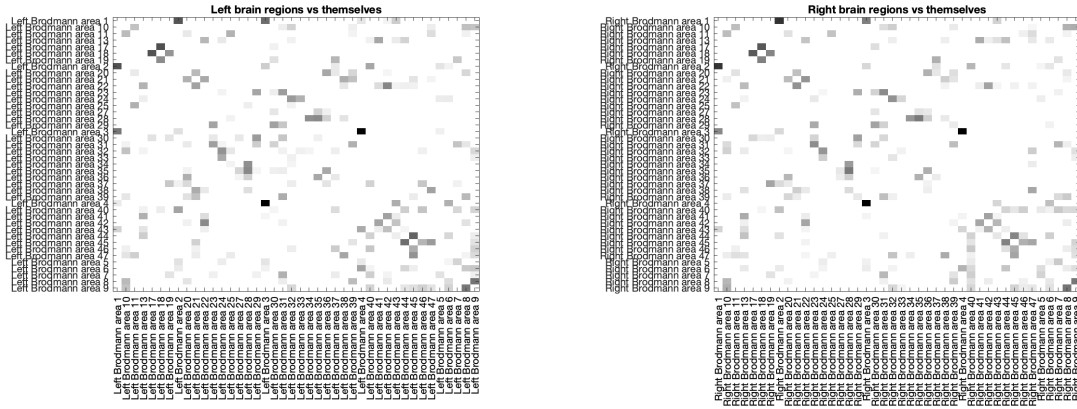

Figure 5: Inter symmetry

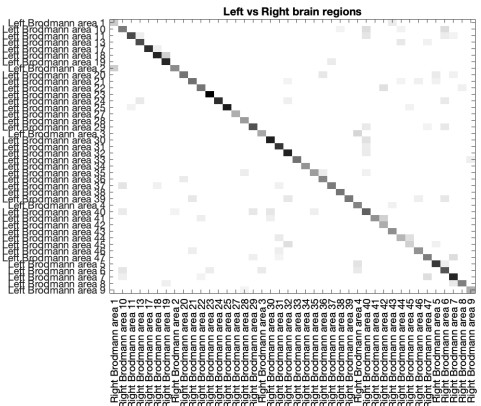

Figure 6: Intra symmetry

For a simple undirected graph with 3 nodes and 3 potential edges, we let the latent underlying graph have the parameter vector $\bar{\theta} = (\bar{\theta}_{12}, \bar{\theta}_{13}, \bar{\theta}_{23}) = (1, 1, 1)$. See Figure 7 for a graph illustration. Then we let the randomness in the parameter for the observable graphs to have the following pattern

$$\Delta = \begin{cases} (a-1, a-1, -1), & \text{with probability } \frac{1}{3} \\ (a-1, -1, a-1), & \text{with probability } \frac{1}{3} \\ (-1, a-1, a-1), & \text{with probability } \frac{1}{3}, \end{cases}$$

resulting in potentially 3 kinds of graphs, each with 2 edges with the same edge value $a$ (see Figure 8).

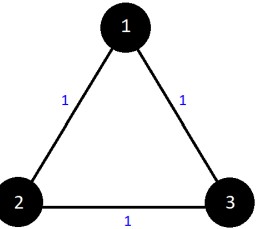

Figure 7: Latent common graph with deterministic edge vector $\bar{\theta} = (\bar{\theta}_{12}, \bar{\theta}_{13}, \bar{\theta}_{23}) = (1, 1, 1)$.

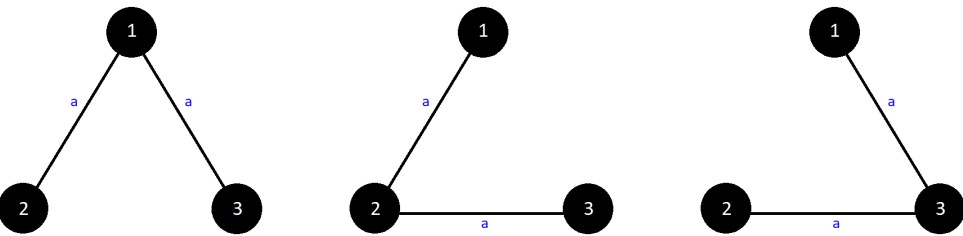

Figure 8: Observable graphs with edge vector $\bar{\theta} + \Delta$, each with 2 edges with the same value $a$.

Next, we need to find a value of $a$ that can fulfill our condition. Notice that the condition involves the expectation over both $\Delta$ and $X$, and thus we need to find out explicitly the probabilities of all combinations of the 3 binary states under each of the three graph settings.

In the following, we use $\mathbb{P}(x_1, x_2, x_3)$ to denote $\mathbb{P}(X_1 = x_1, X_2 = x_2, X_3 = x_3)$ for simplicity. Now consider node $r = 1$. In particular we will need the conditional distribution of $X_1$ given $X_2$ and $X_3$. For the first setting (the graph on the left in Figure 8), we have the joint distributions

$$\mathbb{P}(1, 1, 1) = \mathbb{P}(-1, -1, -1) = \frac{e^{2a}}{Z},$$

$$\mathbb{P}(1, 1, -1) = \mathbb{P}(1, -1, 1) = \mathbb{P}(-1, 1, -1) = \mathbb{P}(-1, -1, 1) = \frac{1}{Z},$$

$$\mathbb{P}(1, -1, -1) = \mathbb{P}(-1, 1, 1) = \frac{e^{-2a}}{Z},$$

where

$$Z = \frac{1}{4 + 2(e^{2a}e^{-2a})}$$

is the normalizing term. The joint distribution of $X_2$ and $X_3$ can be found to be

$$\mathbb{P}_{X_2, X_3}(1, 1) = \mathbb{P}_{X_2, X_3}(-1, -1) = \frac{e^{2a} + e^{-2a}}{Z},$$

$$\mathbb{P}_{X_2, X_3}(1, -1) = \mathbb{P}_{X_2, X_3}(-1, 1) = \frac{2}{Z}.$$

Then we can derive that the conditional expectation of $X_1$ given $X_2$ and $X_3$ are

$$\mathbb{E}[X_1 | X_2 = 1, X_3 = 1] = \frac{e^{2a} - e^{-2a}}{e^{2a} + e^{-2a}},$$

$$\mathbb{E}[X_1 | X_2 = 1, X_3 = -1] = \mathbb{E}[X_1 | X_2 = -1, X_3 = 1] = 0,$$

$$\mathbb{E}[X_1 | X_2 = -1, X_3 = -1] = \frac{e^{-2a} - e^{2a}}{e^{2a} + e^{-2a}}.$$

For the other 3 graph structures (middle and right in Figure 8), we can derive the probabilities and expectations similarly. Also note that for node $r = 1$, these two graph structures are symmetric to $X_1$. Finally with all these values we have, plugging them into the infinity norm in (31) and setting it to be small (e.g., 0 in this illustrative example), we have that $a \approx 1.75$. Since our setting design is symmetric for all $X_1$, $X_2$, $X_3$, the same result hold when $r = 2$ or $3$.

# D Proof of Theorem 4.7

## D.1 Primal-dual Witness for Recovery of the Latent Common Graph

The main technique we use throughout the theoretical proof is the primal-dual witness approach (Wainwright, 2009; Ravikumar et al., 2010) that relies on the Karush-Kuhn Tucker conditions in optimization and concentration inequalities in learning theory. Essentially, it constructs a *primal-dual pair*, i.e., a primal solution $\hat{\theta} \in \mathbb{R}^{p-1}$ and an associated sub-gradient vector $\hat{z} \in \mathbb{R}^{p-1}$ as a dual solution so that the sub-gradient optimality conditions in the convex program (10) are satisfied. We show that under the conditions on $(n, p, d, K)$ stated in the theorem, the primal-dual pair $(\hat{\theta}, \hat{z})$ can be constructed to act as a *witness* that guarantees the method correctly recovers the structure of the graph parametrized by the true common parameter $\bar{\theta}$.

For the convex program (10), the zero sub-gradient optimality condition (Rockafellar, 2015) has the form of

$$\nabla \ell(\hat{\theta}) + \lambda \hat{z} = 0, \tag{32}$$

where the dual (the sub-gradient vector) $\hat{z} \in \mathbb{R}^{p-1}$ must satisfy

$$\text{sign}(\hat{z}_{rt}) = \text{sign}(\hat{\theta}_{rt}) \quad \text{if } \hat{\theta}_{rt} \neq 0 \quad \text{and} \quad |\hat{z}_{rt}| \leq 1 \text{ otherwise.} \tag{33}$$

By convexity, a pair $(\hat{\theta}, \hat{z}) \in \mathbb{R}^{p-1} \times \mathbb{R}^{p-1}$ is a primal-dual optimal solution to the convex program *if and only if* the two conditions (32) and (33) are satisfied. Furthermore, this optimal primal-dual pair correctly specifies the signed neighborhood of node $r$ *if and only if*

$$\text{sign}(\hat{z}_{rt}) = \text{sign}(\bar{\theta}_{rt}) \quad \forall (r, t) \in S, \tag{34}$$

and

$$\hat{\theta}_{rt} = 0 \quad \forall (r, t) \in S^c. \tag{35}$$

The $\ell_1$-regularized logistic regression problem (10) is convex. The following lemma provides sufficient conditions for it to be *strictly* convex and hence the uniqueness of the optimal solution, as well as the shared sparsity among optimal solutions.

**Lemma D.1** (A generalization of Lemma 1 in Ravikumar et al. (2010))**.** *Suppose that there exists an optimal primal solution $\hat{\theta}$ with associated optimal dual vector $\hat{z}$ such that $\|\hat{z}_{S^c}\|_{\infty} < 1$. Then any optimal primal solution $\tilde{\theta}$ must have $\tilde{\theta}_{S^c} = 0$. Moreover, if the Hessian sub-matrix $[\nabla^2 \ell(\hat{\theta}; \{\mathfrak{X}_1^n\}_1^K)]_{SS}$ is strictly positive definite for the loss function defined in the paper, then $\hat{\theta}$ is the unique optimal solution.*

*Proof.* The proof follows exactly the same logic as that for Lemma 1 in Ravikumar et al. (2010), except that the loss function in our case is one more generalized — the average of the losses in each task, which does not change the property of strict convexity when it is present. To see this, note that the loss function in Ravikumar et al. (2010) corresponds to $\ell^{(k)}(\theta)$ we defined in the paper, the loss for each task in our case. □

Based on Lemma D.1, we construct a primal-dual witness pair $(\hat{\theta}, \hat{z})$ with the following steps.

**Step 1.** We set $\hat{\theta}_S$ as the minimizer of the $\ell_1$-penalized likelihood

$$\hat{\theta}_S = \underset{(\hat{\theta}_S, 0)}{\arg \min} \{\ell(\theta; \{\mathfrak{X}_1^n\}_1^K) + \lambda \|\theta_S\|_1\}, \tag{36}$$

and set $\hat{z}_S = \text{sign}(\hat{\theta}_S)$.

**Step 2.** We set $\hat{\theta}_{S^c} = 0$ so that condition (35) holds.

**Step 3.** We obtain $\hat{z}_{S^c}$ from (32) by substituting in the values of $\hat{\theta}_S$ and $\hat{z}_S$. At this point, our construction satisfies conditions (32) and (35).

**Step 4.** We need to show that the stated scaling of $(n, p, d, K)$ in Theorem 4.7 implies that, with high probability, the remaining conditions (33) and (34) are satisfied.

The last step is most challenging and is the goal of the majority of our proof. Our analysis guarantees that $\|\hat{z}_{S^c}\|_\infty < 1$ with high probability. Another condition to be satisfied is the positive definiteness stated in Lemma D.1, for which by Assumptions 4.1 and 4.2, we prove that the sub-matrix of the sample Fisher information matrix is strictly positive definite with high probability, so that the primal solution $\hat{\theta}$ is guaranteed to be unique. The next two subsections contribute exactly to these two parts of the proof.

## D.2 Uniform Convergence of Sample Information Matrices in Auxiliary Tasks

To satisfy the condition of positive definiteness in Lemma D.1 and to lay the foundation for the analysis under the assumptions of the sample information matrix of having bounded eigenvalues in the next subsection D.3, we aim to prove here that if the dependency and incoherence conditions from Assumptions 4.1 and 4.2 are imposed on the *population* Fisher information matrix then under the specified scaling of $(n, p, d, K)$, analogous bounds hold for the *sample* Fisher information matrix with probability converging to one.

Recall the definition of the *population* Fisher information matrix (dropping the subscript $r$) from Section 4.1.1, we have (see (14)):

$$\bar{Q} = \frac{1}{K}\sum_{k=1}^{K}\mathbb{E}[\eta(X^{(k)};\bar{\theta})X_{\backslash r}^{(k)}(X_{\backslash r}^{(k)})^T], \tag{37}$$

and its sample counterpart, i.e., the *sample* Fisher information matrix is defined as

$$Q^N := \hat{\mathbb{E}}[-\nabla^2\ell(\bar{\theta}_{\backslash r}; \{\mathfrak{X}_1^n\}_1^K)] = \frac{1}{K}\sum_{k=1}^{K}\frac{1}{n}\sum_{i=1}^{n}\eta(x_i^{(k)};\bar{\theta})x_{i,\backslash r}^{(k)}(x_{i,\backslash r}^{(k)})^T. \tag{38}$$

Here the $\mathbb{E}$ in $\bar{Q}$ is the population expectation under the joint distribution of the randomness in the model parameters $\{\Delta^{(k)}\}_{k=1}^{K}$ and the random samples $\{\mathfrak{X}_1^n\}_1^K$ for the $K$ auxiliary tasks, while $\hat{\mathbb{E}}$ in $Q^N$ denotes the empirical expectation, and the variance function is defined in (14).

### D.2.1 Uniform Convergence for Dependence Assumption

For the dependence assumption, we show that the eigenvalue bounds in Assumptions 4.1 hold with high probability for sample Fisher information matrix and sample covariance matrices in the following two lemmas:

**Lemma D.2.** *Suppose that Assumption 4.1 holds for the population Fisher information matrix $\bar{Q}$ and the pooled population covariance matrix $\mathbb{E}(\frac{1}{K}\sum_{k=1}^{K}X^{(k)}(X^{(k)})^T)$. For any $\delta > 0$ and some fixed constants $A$ and $B$, we have*

$$\mathbb{P}[\Lambda_{\min}(Q_{SS}^N) \le C_{\min} - \delta] \le 2\exp\left(-A\frac{\delta^2 nK}{d^2} + B\log(d)\right), \tag{39}$$

*and*

$$\mathbb{P}\left[\Lambda_{\max}\left[\frac{1}{K}\sum_{k=1}^{K}\frac{1}{n}\sum_{i=1}^{n}x_{i,\backslash r}^{(k)}(x_{i,\backslash r}^{(k)})^T\right] \ge D_{\max} - \delta\right] \le 2\exp\left(-A\frac{\delta^2 nK}{d^2} + B\log(d)\right). \tag{40}$$

The proof of this lemma is in Section H.1.1.

### D.2.2 Uniform Convergence for Incoherence Assumption

The following lemma is the analog for the incoherence assumption in Assumption 4.2 showing that the scaling of $(n, p, d, K)$ given in Theorem 4.7 guarantees that population incoherence implies sample incoherence.

**Lemma D.3.** *If the* pooled *population covariance satisfies* $\left\|\left|\bar{Q}_{S^c S}(\bar{Q}_{SS})^{-1}\right|\right\|_\infty \leq 1 - \alpha$ *with parameter* $\alpha \in (0, 1]$, *then the sample matrix satisfies an analogous version, with high probability in the sense that*

$$\mathbb{P}\left[\left\|\left|Q^N_{S^c S}(Q^N_{SS})^{-1}\right|\right\|_\infty \geq 1 - \frac{\alpha}{2}\right] \leq \exp\left(-B\frac{nK}{d^3} + \log(p)\right), \tag{41}$$

*for some fixed constant $B$.*

The proof of this lemma is in Section H.1.2.

### D.3 Analysis under Assumptions of Sample Information Matrices in Auxiliary Tasks

With the incoherence and dependence conditions guaranteed with high probability (proved in Section D.2), we then begin to establish model selection consistency when assumptions are imposed directly on the sample Fisher information matrix $Q^N$ as opposed to $\bar{Q}$. Recalling the definition (38) of the sample Fisher information matrix $Q^N$, we define the "good event"

$$\mathcal{M}(\{\mathfrak{X}^n_1\}^K_1) := \{\{\mathfrak{X}^n_1\}^K_1 \in \{-1, +1\}^{K \times n \times p} | Q^N \text{ satisfies Assumptions 4.1 and 4.2}\}. \tag{42}$$

As in the statement of Theorem 4.7, the quantities $L$ and $c_1$ refer to constants independent of $(n, p, d, K)$. With this notation, we have the following:

**Proposition D.4** (Fixed design for auxiliary tasks). *If the event $\mathcal{M}(\{\mathfrak{X}^n_1\}^K_1)$ holds, the sample size per task and number of tasks satisfy $nK > Ld^2 \log p$, and the regularization parameter is chosen such that $\lambda \geq \beta\sqrt{\frac{\log p}{nK}}$ for some fixed constant $\beta > 0$, then for recovering the true common parameter vector $\bar{\theta}$ of the latent common graph, with probability at least $1 - 6\exp(-c^2_\lambda nK) \to 1$ for some constant $c > 0$, the following properties hold,*

*(a) For each node $r \in V$, the $\ell_1$-regularized logistic regression for the improper estimation of $\bar{\theta}$ has a unique solution, and so uniquely specifies a signed neighborhood $\hat{N}_\pm(r)$.*

*(b) For each $r \in V$, the estimated signed neighborhood $\hat{N}_\pm(r)$ correctly excludes all edges not in the true support union. Moreover, it correctly includes all edges with $|\bar{\theta}_{rt}| \geq \frac{10}{C_{\min}}\sqrt{d}\lambda$, along with their correct sign.*

Intuitively, this result guarantees that if the sample Fisher information matrix is "good", then the probability of success for the recovery of the underlying latent graph parametrized by the true common parameter $\bar{\theta}$ converges to 1 at the specified rate. The following subsection is devoted to the proof of Proposition D.4.

#### D.3.1 Key Technical Results in the Proof of Proposition D.4

We follow the steps of primal-dual witness as stated at the beginning of Section D. Since the key is to guarantee the strict dual feasibility $\|\hat{z}_{S^c}\|_\infty < 1$ with high probability in **Step 4**, we make a series of deliberate constructions to find out the explicit expression of $\|\hat{z}_{S^c}\|_\infty$ and try to bound it.

Starting from the stationarity condition in (32): $\nabla\ell(\hat{\theta}; \{\mathfrak{X}^n_1\}^K_1) + \lambda\hat{z} = 0$, adding to both sides

$$W^N := -\nabla\ell(\bar{\theta}; \{\mathfrak{X}^n_1\}^K_1), \tag{43}$$

we get

$$\nabla\ell(\hat{\theta}; \{\mathfrak{X}^n_1\}^K_1) - \nabla\ell(\bar{\theta}; \{\mathfrak{X}^n_1\}^K_1) = W^N - \lambda\hat{z}. \tag{44}$$

Note that $W^N$ is just a shorthand notation for the $(p-1)$-dimensional score function. Then, applying the mean-value theorem coordinate-wise to the expansion (44) gives

$$\nabla^2\ell(\bar{\theta}; \{\mathfrak{X}^n_1\}^K_1)[\hat{\theta} - \bar{\theta}] = W^N - \lambda\hat{z} + R^N, \tag{45}$$

where the remainder term takes the form

$$R^N_j = -[\nabla^2\ell(\theta^{(j)}; \{\mathfrak{X}^n_1\}^K_1) - \nabla^2\ell(\bar{\theta}; \{\mathfrak{X}^n_1\}^K_1)]^T_j(\hat{\theta} - \bar{\theta}), \tag{46}$$

with $\theta^{(j)}$ being a parameter vector on the line between $\bar{\bar{\theta}}$ and $\hat{\theta}$, and with $[\cdot]_j^T$ denoting the $j$-th row of the matrix.

Recalling our shorthand notation $Q^N = -\nabla^2\ell(\bar{\theta}; \{\mathfrak{X}_1^n\}_1^K)$ and the fact that we have set $\hat{\theta}_{S^c} = 0$ in our primal-dual construction:

$$\begin{cases} -Q_{S^cS}^N[\hat{\theta}_S - \bar{\theta}_S] = W_{S^c}^N - \lambda\hat{z}_{S^c} + R_{S^c}^N \\ -Q_{SS}^N[\hat{\theta}_S - \bar{\theta}_S] = W_S^N - \lambda\hat{z}_S + R_S^N \end{cases}. \tag{47}$$

Since the matrix $Q_{SS}^N$ is invertible by assumption, it can be re-written as

$$Q_{S^cS}^N(Q_{SS}^N)^{-1}[W_S^N - \lambda\hat{z}_S + R_S^N] = W_{S^c}^N - \lambda\hat{z}_{S^c} + R_{S^c}^N, \tag{48}$$

by using the common parts $\hat{\theta}_S - \bar{\theta}_S$ in the equations. Rearranging yields:

$$\hat{z}_{S^c} = \frac{1}{\lambda}[W_{S^c}^N + R_{S^c}^N] - \frac{1}{\lambda}Q_{S^cS}^N(Q_{SS}^N)^{-1}[W_S^N + R_S^N] + Q_{S^cS}^N(Q_{SS}^N)^{-1}\hat{z}_S \tag{49}$$

By the assumptions $\left\|\left|Q_{S^cS}^N(Q_{SS}^N)^{-1}\right\|\right|_\infty \leq 1 - \alpha$, and the fact that $\|\hat{z}_S\|_\infty = 1$, we have

$$\|\hat{z}_{S^c}\|_\infty \leq (1 - \alpha) + (2 - \alpha)\left[\frac{\|R^N\|_\infty}{\lambda} + \frac{\|W^N\|_\infty}{\lambda}\right]. \tag{50}$$

**Strict Dual Feasibility.** Now, to satisfy the *strict dual feasibility* $\|\hat{z}_{S^c}\|_\infty < 1$, we need to bound $\frac{\|W^N\|_\infty}{\lambda}$ and $\frac{\|R^N\|_\infty}{\lambda}$. The following two lemmas show that $\frac{\|W^N\|_\infty}{\lambda}$ decays to 0 at an exponential rate and $\|R^N\|_\infty$ can be bounded deterministically accordingly under some conditions.

**Lemma D.5** (Decaying behavior of $W^N$). *For the specified mutual incoherence parameter $\alpha \in (0, 1]$ and a fixed constant $c$, we have*

$$\mathbb{P}\left(\frac{2 - \alpha}{\lambda}\|W^N\|_\infty > \frac{\alpha}{4}\right) \leq 6\exp\left(-\frac{\alpha^2\lambda^2}{c(2 - \alpha)^2}nK + log(p)\right), \tag{51}$$

*which converges to 0 at rate $\exp(-c'\lambda^2 nK)$ for some fixed constant $c'$, as long as $\lambda \geq \frac{\sqrt{2c}(2-\alpha)}{\alpha}\sqrt{\frac{\log p}{nK}}$.*

The proof of this lemma is in Section H.2.1

**Lemma D.6** (Control on the remainder term $R^N$). *If $\lambda d \leq \frac{C_{\min}^2}{100D_{\max}}\frac{\alpha}{2-\alpha}$ and $\|W^N\|_\infty \leq \frac{\lambda}{4}$, then*

$$\frac{\|R^N\|_\infty}{\lambda} \leq \frac{25D_{\max}}{C_{\min}^2}\lambda d \leq \frac{\alpha}{4(2 - \alpha)}. \tag{52}$$

The proof of this lemma is in Section H.2.2.

Next, applying Lemmas D.5 and D.6, we have the *strict dual feasibility* as

$$\|\hat{z}_{S^c}\|_\infty \leq (1 - \alpha) + \frac{\alpha}{4} + \frac{\alpha}{4} = 1 - \frac{\alpha}{2},$$

with probability converging to one.

**Correct Sign Recovery.** For the statement of *correct sign recovery* in Proposition D.4, we show here that our primal sub-vector $\hat{\theta}_S$ defined by (36) satisfies sign consistency $\text{sign}(\hat{\theta}_S) = \text{sign}(\bar{\theta}_S)$, which suffices to show that

$$\|\hat{\theta}_S - \bar{\theta}_S\|_\infty \leq \frac{\bar{\theta}_{\min}}{2},$$

where $\bar{\theta}_{\min} := \min_{(r,t)\in S}|\bar{\theta}_{rt}|$. The following lemma is used in the proof here, which establishes that the sub-vector $\hat{\theta}_S$ is an $\ell_2$-consistent estimate of the true common sub-vector $\bar{\theta}_S$.

**Lemma D.7** ($\ell_2$-consistency of primal sub-vector). *If* $\lambda d \leq \frac{C_{\min}^2}{10D_{\max}}$ *and* $\|W^N\|_\infty \leq \frac{\lambda}{4}$, *then*

$$\|\hat{\theta}_S - \bar{\theta}_S\|_2 \leq \frac{5}{C_{\min}}\sqrt{d}\lambda \tag{53}$$

The proof of this lemma is in Section H.2.3.

By Lemma D.7, we can write

$$\frac{2}{\bar{\theta}_{\min}}\|\hat{\theta}_S - \bar{\theta}_S\|_\infty \leq \frac{2}{\bar{\theta}_{\min}}\|\hat{\theta}_S - \bar{\theta}_S\|_2$$
$$\leq \frac{2}{\bar{\theta}_{\min}}\frac{5}{C_{\min}}\sqrt{d}\lambda,$$

which is less than 1 as long as $|\bar{\theta}_{rt}| \geq \frac{10}{C_{\min}}\sqrt{d}\lambda$.

Now it is clear that the uniform convergence of sample information matrices (in Section D.2) together with Proposition D.4 (from Section D.3) completes the proof of Theorem 4.7.

## E    Proof of Theorem 4.8

For $\mathbf{\Gamma} = \{\Gamma^{(k)}\}_{k=1}^K$, we know that there is a bijection between $\mathcal{E}$ and the set of all circular permutations of nodes $V = \{1, \ldots, p\}$. Thus $|\mathcal{E}|$, i.e., the size of $\mathcal{E}$, is the total number of circular permutations of $p$ elements, which is $C_E := (p-1)!/2$. Since $E$ is uniformly distributed on $\mathcal{E}$, the entropy of $E$ given $\mathbf{\Gamma}$ is $H(E|\mathbf{\Gamma}) = \log C_E$.

Consider a family of $p$-dimensional Ising models of size $K$ with parameters $\{\bar{\theta}^{(k)}\}_{k=1}^K$ generated according to Theorem 4.8. We use $\mathbf{X} := \{X_t^{(k)}\}_{1 \leq t \leq n, 1 \leq k \leq K}$ to denote the collection of $n$ samples from each of the $K$ tasks. Then for the mutual information $\mathbb{I}(\mathbf{X}; E|\mathbf{\Gamma})$. We have the following bound:

$$\mathbb{I}(\mathbf{X}; E|\mathbf{\Gamma}) \leq \frac{1}{C_E^2}\sum_E\sum_{E'}\mathbb{KL}(P_{\mathbf{X}|E,\mathbf{\Gamma}}\|P_{\mathbf{X}|E',\mathbf{\Gamma}})$$
$$= \frac{1}{C_E^2}\sum_E\sum_{E'}\sum_{k=1}^K\sum_{t=1}^n\mathbb{KL}(P_{X_t^{(k)}|E,\Gamma^{(k)}}\|P_{X_t^{(k)}|E',\Gamma^{(k)}}) \tag{54}$$

According to Lemma 19 in (Honorio, 2011), $P_{X_t^{(k)}|E,\Gamma^{(k)}}$ is $(\ell_\infty, 2)$-Lipschitz continuous for $\forall E \in \mathcal{E}$ and $1 \leq k \leq K$. Then by Theorem 7 in (Honorio, 2011), we have

$$\mathbb{KL}\left(P_{X_t^{(k)}|E,\Gamma^{(k)}}\middle\|P_{X_t^{(k)}|E',\Gamma^{(k)}}\right) \leq 2\|\bar{\theta}^{(k)} - \bar{\theta}'^{(k)}\|_1 \leq 2p/d^3, \tag{55}$$

where the second inequality follows by the definition of $\bar{\theta}^{(k)}$ and $\Gamma^{(k)} \in [-1/d^4, 1/d^4]^{p \times p}$ in Theorem 4.8. Putting (55) back to (54) gives

$$\mathbb{I}(\mathbf{X}; E|\mathbf{\Gamma}) \leq \frac{1}{C_E^2}\sum_E\sum_{E'}\sum_{k=1}^K\sum_{t=1}^n 2p/d^3 = 2npK/d^3 \tag{56}$$

For any estimate $\hat{S}$ of $S$, define $\hat{E} = \{(i,j) : (i,j) \in \hat{S}, i \neq j\}$. Since $E \subseteq S$, we have $\mathbb{P}\{S \neq \hat{S}\} \geq \mathbb{P}\{E \neq \hat{E}\}$. Then by applying Theorem 1 in (Ghoshal & Honorio, 2017), we get

$$\mathbb{P}\{S \neq \hat{S}\} \geq \mathbb{P}\{E \neq \hat{E}\}$$
$$\geq 1 - \frac{\mathbb{I}(\mathbf{X}; S|\mathbf{\Gamma}) + \log 2}{H(S|\mathbf{\Gamma})}$$
$$\geq 1 - \frac{2npK/d^3 + \log 2}{\log[(p-1)!/2]}$$

For $\log((p-1)!)$, we have:

$$\begin{aligned}
\log((p-1)!) &= \sum_{i=1}^{p-1} \log i \\
&\geq \int_1^{p-1} \log x \, dx \\
&= (p-1)\log(p-1) - p + 2 \\
&= (p-1)\log p + (p-1)\log\frac{p-1}{p} + 2 - p
\end{aligned}$$

Since

$$(p-1)\log\frac{p-1}{p} + 2 = 2 - (p-1)\log\left(1 + \frac{1}{p-1}\right) \geq 2 - 1 > 0$$

we have

$$\log((p-1)!) \geq (p-1)\log p - p = p\log p - p - \log p$$

$$\log((p-1)!/2) = \log((p-1)!) - \log 2 \geq p\log p - p - \log 2p$$

For $p \geq 5$, $p\log p - p - \log 2p > 0$, thus we have

$$\mathbb{P}\{S \neq \hat{S}\} \geq 1 - \frac{npK/d^3 + \log 2}{\log[(p-1)!/2]} \geq 1 - \frac{2npK/d^3 + \log 2}{p\log p - p - \log 2p}$$

which completes our proof of Theorem 4.8.

# F    Proof of Theorem 4.9

We have supposed that we have recovered the true support union $S$ from our estimate for the true common parameter, $\hat{\theta}$. The constraint in (12) then enables us to convert the problem into one without the restriction and with a parameter of dimension $|S_r|$ with $|S_r| \leq d$ for all $r \in V$, for we can combine the constraint straightforward into the minimization problem. With some abuse of notation using $S$ to denote $S_r$ as before, we can write

$$\hat{\theta}_S^{(K+1)} = \arg\min_{\theta_S \in \mathbb{R}^{p-1}} \left\{ \ell^{(K+1)}(\theta_S; \{\mathfrak{X}_{1,S}^{n^{(K+1)}}\}^{(K+1)}) + \lambda^{(K+1)}\|\theta_S\|_1 \right\}, \tag{57}$$

and $\hat{\theta}_{S^c}^{(K+1)} = 0$, since we know that

$$S^{(K+1)} \subseteq S. \tag{58}$$

This simplifies the problem to a great extent, and our proof henceforth takes on a similar pattern as the proof without restriction in Ravikumar et al. (2010), but with reduced dimensions.

## F.1    Primal-dual Witness for Graph Recovery in the Novel Task

We again use the primal-dual witness approach (Wainwright, 2009; Ravikumar et al., 2010) as stated in the proof of Theorem 4.7. See Section D. With the loss function, parameter and data changed for only one task — the novel task.

For the convex program (57), the zero sub-gradient optimality condition (Rockafellar, 2015) has the form of

$$\nabla\ell^{(K+1)}(\hat{\theta}_S^{(K+1)}) + \lambda^{(K+1)}\hat{z}_S^{(K+1)} = 0, \tag{59}$$

where the dual (the sub-gradient vector) $\hat{z}_S^{(K+1)} \in \mathbb{R}^{|S_r|}$ must satisfy

$$\text{sign}(\hat{z}_{rt}^{(K+1)}) = \text{sign}(\hat{\theta}_{rt}^{(K+1)}) \quad \text{if } \hat{\theta}_{rt} \neq 0 \quad \text{and} \quad |\hat{z}_{rt}| \leq 1 \text{ otherwise.} \tag{60}$$

By convexity, a pair $(\hat{\theta}_S^{(K+1)}, \hat{z}_S^{(K+1)}) \in \mathbb{R}^{|S_r|} \times \mathbb{R}^{|S_r|}$ is a primal-dual optimal solution to the convex program *if and only if* the two conditions (59) and (60) are satisfied. Furthermore, this optimal primal-dual pair correctly specifies the signed neighborhood of node $r$ *if and only if*

$$\text{sign}(\hat{z}_{rt}^{(K+1)}) = \text{sign}(\bar{\theta}_{rt}^{(K+1)}) \quad \forall (r,t) \in S^{(K+1)}, \tag{61}$$

and

$$\hat{\theta}_{rt}^{(K+1)} = 0 \quad \forall (r,t) \in [S^{(K+1)}]^c. \tag{62}$$

For this restricted problem, we have a similar lemma as D.1 to for the uniqueness of the solution and shared sparsity.

**Lemma F.1** (Lemma 1 in Ravikumar et al. (2010) with reduced dimensions)**.** *Suppose that there exists an optimal primal solution $\hat{\theta}_S^{(K+1)}$ with associated optimal dual vector $\hat{z}_S^{(K+1)}$ such that $\|\hat{z}_{[S^{(K+1)}]^c}^{(K+1)}\|_\infty < 1$. Then any optimal primal solution $\tilde{\theta}_S^{(K+1)}$ must have $\tilde{\theta}_{[S^{(K+1)}]^c}^{(K+1)} = 0$. Moreover, if the Hessian sub-matrix $[\nabla^2 \ell^{(K+1)}(\hat{\theta}_S^{(K+1)}; \{\mathfrak{X}_{1,S}^{n^{(K+1)}}\}^{(K+1)})]_{S^{(K+1)}S^{(K+1)}}$ is strictly positive definite, then $\hat{\theta}_S^{(K+1)}$ is the unique optimal solution.*

*Proof.* See proof of Lemma 1 in Ravikumar et al. (2010). The case in this convex program has a loss function $\ell^{(K+1)}$ carrying the same meaning as those in Ravikumar et al. (2010)), only with the dimensions of the parameter vector and our samples reduced since they are restricted to the true support union $S$ (see (58)). $\square$

Based on Lemma F.1, we construct a primal-dual witness pair $(\hat{\theta}_S^{(K+1)}, \hat{z}_S^{(K+1)})$ with the following steps.

**Step 1.** We set $\hat{\theta}_{S^{(K+1)}}^{(K+1)}$ as the minimizer of the $\ell_1$-penalized likelihood

$$\hat{\theta}_{S^{(K+1)}} = \arg\min_{(\hat{\theta}_{S^{(K+1)}}, 0)} \{\ell(\theta_S; \{\mathfrak{X}_{1,S}^n\}_1^K) + \lambda^{(K+1)} \|\theta_{S^{(K+1)}}\|_1\}, \tag{63}$$

and set $\hat{z}_{S^{(K+1)}}^{(K+1)} = \text{sign}(\hat{\theta}_{S^{(K+1)}}^{(K+1)})$.

**Step 2.** We set $\hat{\theta}_{[S^{(K+1)}]^c}^{(K+1)} = 0$ so that condition (62) holds.

**Step 3.** We obtain $\hat{z}_{[S^{(K+1)}]^c}^{(K+1)}$ from (32) by substituting in the values of $\hat{\theta}_{S^{(K+1)}}^{(K+1)}$ and $\hat{z}_{S^{(K+1)}}^{(K+1)}$ so that our construction satisfies conditions (59) and (62).

**Step 4.** We need to show that the stated scaling of $(n^{(K+1)}, d)$ in Theorem 4.7 implies that, with high probability, the remaining conditions (60) and (61) are satisfied.

Our analysis in the last step guarantees that $\|\hat{z}_{[S^{(K+1)}]^c}^{(K+1)}\|_\infty < 1$ with high probability. Another condition to be satisfied is the positive definiteness stated in Lemma F.1, for which by Assumptions 4.5 and 4.6, we prove that the sub-matrix of the sample Fisher information matrix is strictly positive definite with high probability, so that the primal solution $\hat{\theta}_S^{(K+1)}$ is guaranteed to be unique. The next two subsections contribute to these two parts of the proof.

## F.2 Uniform Convergence of Sample Information Matrices in Novel Task

To satisfy the condition of positive definiteness in Lemma F.1 and to prepare for the analysis under the assumptions of the sample information matrix of having bounded eigenvalues in the next subsection F.3, we will prove in this subsection that if the dependency and incoherence conditions from Assumptions 4.5 and 4.6 are imposed on the *population* Fisher information matrix then under the specified scaling of $(n^{(K+1)}, d)$, analogous bounds hold for the *sample* Fisher information matrix with probability converging to one.

Recall the definition of the *population* Fisher information matrix (dropping the subscript $r$) from (4.1.2), we have (see (20)):

$$\bar{Q}^{(K+1)} = \mathbb{E}[\eta(X_S^{(K+1)}; \bar{\theta}_S^{(K+1)}) X_S^{(K+1)} (X_S^{(K+1)})^T]. \tag{64}$$

and its sample counterpart, i.e., the *sample* Fisher information matrix is defined as

$$Q^{(K+1)} := \hat{\mathbb{E}}[-\nabla^2 \ell^{(K+1)}(\bar{\theta}_S^{(K+1)}; \{\mathfrak{X}_{1,S}^n\}_1^K)] = \frac{1}{n^{(K+1)}} \sum_{i=1}^{n^{(K+1)}} \eta(x_{i,S}^{(K+1)}; \bar{\theta}_S^{(K+1)}) x_{i,S}^{(K+1)} (x_{i,S}^{(K+1)})^T. \tag{65}$$

Here the $\mathbb{E}$ in $\bar{Q}^{(K+1)}$ is the population expectation under the joint distribution of the randomness in the model parameter $\Delta^{(K+1)}$ and the random samples $\{\mathfrak{X}_1^n\}^{(K+1)}$ for the the novel task. $\hat{\mathbb{E}}$ in $Q^{(K+1)}$ denotes the empirical expectation, and the variance function is defined in (14).

### F.2.1 Uniform Convergence for Dependence Assumption

For the dependence assumption, we show that the eigenvalue bounds in Assumptions 4.5 hold with high probability for sample Fisher information matrix and sample covariance matrices in the following two lemmas:

**Lemma F.2.** *Suppose that Assumption 4.5 holds for the population Fisher information matrix $\bar{Q}^{(K+1)}$ and population covariance matrix $\mathbb{E}(X_S^{(K+1)}(X_S^{((K+1)})^T)$. For any $\delta > 0$ and some fixed constants $A$ and $B$, we have*

$$\mathbb{P}\left[\Lambda_{\min}(Q_{S(K+1)S(K+1)}^{(K+1)}) \leq C_{\min}^{(K+1)} - \delta\right] \leq 2\exp\left(-A\frac{\delta^2 n^{(K+1)}}{d^2} + B\log(d)\right), \tag{66}$$

*and*

$$\mathbb{P}\left[\Lambda_{\max}(\frac{1}{n^{(K+1)}}\sum_{i=1}^{n^{(K+1)}} x_{i,S}^{(K+1)}(x_{i,S}^{(K+1)})^T) \geq D_{\max}^{(K+1)} - \delta\right] \leq 2\exp\left(-A\frac{\delta^2 n^{(K+1)}}{d^2} + B\log(d)\right) \tag{67}$$

The proof of this lemma is in Section I.1.1.

### F.2.2 Uniform Convergence for Incoherence Assumption

The following lemma is the analog for the incoherence assumption in Assumption 4.2 showing that the scaling of $(n, p, d, K)$ given in Theorem 4.7 guarantees that population incoherence implies sample incoherence.

**Lemma F.3.** *If the population covariance satisfies $\left\|\left\|\bar{Q}_{[S^{(K+1)}]^c S^{(K+1)}}^{(K+1),S}(\bar{Q}_{S^{(K+1)} S^{(K+1)}}^{(K+1)})^{-1}\right\|\right\|_{\infty} \leq 1 - \alpha$ with parameter $\alpha \in (0,1]$, then the sample matrix satisfies an analogous version, with high probability in the sense that*

$$\mathbb{P}[\left\|\left\|Q_{[S^{(K+1)}]^c S^{(K+1)}}^{(K+1),S}(Q_{S^{(K+1)} S^{(K+1)}}^{(K+1)})^{-1}\right\|\right\|_{\infty} \geq 1 - \alpha^{(K+1)}/2] \leq \exp\left(-B\frac{n^{(K+1)}}{d^3} + \log(d)\right) \tag{68}$$

*for some fixed constant $B$.*

The proof of this lemma is in Section I.1.2.

### F.3 Analysis under Assumptions of Sample Information Matrices

With the incoherence and dependence conditions guaranteed with high probability (proved in Section F.2), we can begin to establish model selection consistency when assumptions are imposed directly on the sample Fisher information matrix $Q^{(K+1)}$ as opposed to $\bar{Q}^{(K+1)}$. Recalling the definition (20) of the sample Fisher information matrix $Q^{(K+1)}$, we define a "good event" for the novel task

$$\mathcal{M}^{(K+1)}(\{\mathfrak{X}_{1,S}^{n^{(K+1)}}\}^{(K+1)})$$
$$:= \{\{\mathfrak{X}_{1,S}^{n^{(K+1)}}\}^{(K+1)} \in \{-1,+1\}^{n^{(K+1)} \times |S_r|} | Q^{(K+1)} \text{ satisfies Assumptions 4.5 and 4.6}\}. \tag{69}$$

As in the statement of Theorem 4.7, the quantities $L$ and $c_2$ refer to constants independent of $(n^{(K+1)}, p, d)$. With this notation, we have the following proposition:

**Proposition F.4** (Fixed design for novel task)**.** *Suppose we have recovered the true support union $S$. If the event $\mathcal{M}^{(K+1)}(\{\mathfrak{X}_{1,S}^{n^{(K+1)}}\}^{(K+1)})$ holds, the sample size satisfy $n^{(K+1)} > Ld^2 \log d$, and the regularization parameter is chosen such that $\lambda \geq 16\frac{(2-\alpha)}{\alpha}\sqrt{\frac{\log d}{n^{(K+1)}}}$, then for recovering the true common parameter vector $\bar{\theta}^{(K+1)}$ of the latent common graph, with probability at least $1 - 2\exp(-c\lambda^2 n^{(K+1)}) \to 1$ for some constant $c > 0$, the following properties hold,*

*(a) For each node $r \in V$, the $\ell_1$-regularized logistic regression for estimating $\bar{\theta}_S^{(K+1)}$ in the novel task, given data $\{\mathfrak{X}_1^{n^{(K+1)}}\}^{(K+1)}$ has a unique solution $\hat{\theta}_S^{(K+1)}$, and so uniquely specifies a signed neighborhood $\hat{\mathcal{N}}_{\pm}^{(K+1)}(r) := \{\text{sign}(\hat{\theta}_{ru}^{(K+1)})u | u \in V \setminus r, \hat{\theta}_{ru}^{(K+1)} \neq 0\}$*

*(b) For each $r \in V$, the estimated signed neighborhood $\hat{\mathcal{N}}_{\pm}^{(K+1)}(r)$ correctly excludes all edges not in the true neighborhood $\mathcal{N}_{\pm}^{(K+1)}(r) := \{\text{sign}(\bar{\theta}_{ru}^{(K+1)})u | u \in V \setminus r, \bar{\theta}_{ru}^{(K+1)} \neq 0\}$. Moreover, it correctly includes all edges with $|\bar{\theta}_{rt}^{(K+1)}| \geq \frac{10}{C_{\min}^{(K+1)}}\sqrt{d}\lambda^{(K+1)}$, along with their correct sign.*

Loosely stated, this result guarantees that if the sample Fisher information matrix is "good", then the probability of success for the recovery graph by converges to 1 at the specified rate. The following subsection is devoted to the proof of Proposition F.4.

### F.3.1 Key Technical Results in the Proof of Proposition F.4

We follow the steps of primal-dual witness as stated at the beginning of Section F. Since the key is to guarantee the strict dual feasibility $\|\hat{z}_{[S^{(K+1)}]^c}^{(K+1),S}\|_\infty < 1$ with high probability in **Step 4**, we first try to find out the explicit expression of $\|\hat{z}_{[S^{(K+1)}]^c}^{(K+1),S}\|_\infty$ and try to bound it.

Starting from the stationarity condition in (32): $\nabla\ell(\hat{\theta}_S^{(K+1)}) + \lambda\hat{z}_S^{(K+1)} = 0$, adding to both sides

$$W^{(K+1)} := -\nabla\ell^{(K+1)}(\bar{\theta}_S^{(K+1)}), \tag{70}$$

noticing that $\mathbb{E}[W^{(K+1)}] = 0$, and skipping writing down the sample $\{\mathfrak{X}_{1,S}^{n^{(K+1)}}\}^{(K+1)}$ in the loss function, we get

$$\nabla\ell^{(K+1)}(\hat{\theta}_S^{(K+1)}) - \nabla\ell^{(K+1)}(\bar{\theta}_S^{(K+1)}) = W^{(K+1)} - \lambda^{(K+1)}\hat{z}_S^{(K+1)}. \tag{71}$$

Note that $W^{(K+1)}$ is just a shorthand notation for the $|S_r|$-dimensional score function. Then, applying the mean-value theorem coordinate-wise to the expansion (71) gives

$$\nabla^2\ell^{(K+1)}(\bar{\theta}_S^{(K+1)})[\hat{\theta}_S^{(K+1)} - \bar{\theta}_S^{(K+1)}] = W^{(K+1)} - \lambda^{(K+1)}\hat{z}_S^{(K+1)} + R^{(K+1)}, \tag{72}$$

where the remainder term takes the form

$$R_j^{(K+1)} = -[\nabla^2\ell^{(K+1)}(\theta_S^{(K+1)j}) - \nabla^2\ell^{(K+1)}(\bar{\theta}_S^{(K+1)})]_j^T(\hat{\theta}_S^{(K+1)} - \bar{\theta}_S^{(K+1)}), \tag{73}$$

with $\theta_S^{(K+1)j}$ a parameter vector on the line between $\bar{\theta}_S^{(K+1)}$ and $\hat{\theta}_S^{(K+1)}$, and with $[\cdot]_j^T$ denoting the $j$-th row of the matrix.

Recalling our shorthand notation $Q^{(K+1)} = -\nabla^2\ell^{(K+1)}(\bar{\theta}_S^{(K+1)}; \{\mathfrak{X}_1^{n^{(K+1)}}\}^{(K+1)})$ and the fact that we have set $\hat{\theta}_{[S^{(K+1)}]^c}^{(K+1),S} = 0$ in our primal-dual construction:

$$\begin{cases} -Q_{[S^{(K+1)}]^c S^{(K+1)}}^{(K+1)}[\hat{\theta}_{S^{(K+1)}} - \bar{\theta}_{S^{(K+1)}}] = W_{[S^{(K+1)}]^c}^{(K+1)} - \lambda^{(K+1)}\hat{z}_{[S^{(K+1)}]^c}^{(K+1),S} + R_{[S^{(K+1)}]^c}^{(K+1)} \\ -Q_{S^{(K+1)} S^{(K+1)}}^{(K+1)}[\hat{\theta}_{S^{(K+1)}} - \bar{\theta}_{S^{(K+1)}}] = W_{S^{(K+1)}}^{(K+1)} - \lambda^{(K+1)}\hat{z}_{S^{(K+1)}}^{(K+1)} + R_{S^{(K+1)}}^{(K+1)} \end{cases} \tag{74}$$

Since the matrix $Q^{(K+1)}_{S(K+1)S(K+1)}$ is invertible by assumption, it can be re-written as

$$
\begin{aligned}
Q^{(K+1)}_{[S(K+1)]^c S(K+1)} (Q^{(K+1)}_{S(K+1)S(K+1)})^{-1} [W^{(K+1)}_{S(K+1)} - \lambda^{(K+1)} \hat{z}^{(K+1)}_{S(K+1)} + R^{(K+1)}_{S(K+1)}] \\
= W^{(K+1)}_{[S(K+1)]^c} - \lambda^{(K+1)} \hat{z}^{(K+1),S}_{[S(K+1)]^c} + R^{(K+1)}_{[S(K+1)]^c},
\end{aligned}
\tag{75}
$$

by using the common parts $\hat{\theta}^{(K+1)}_{S(K+1)} - \bar{\theta}^{(K+1)}_{S(K+1)}$ in the equations. Rearranging yields:

$$
\hat{z}^{(K+1),S}_{[S(K+1)]^c} = \frac{1}{\lambda^{(K+1)}} [W^{(K+1)}_{[S(K+1)]^c} + R^{(K+1)}_{[S(K+1)]^c}] \tag{76}
$$

$$
- \frac{1}{\lambda^{(K+1)}} Q^{(K+1)}_{[S(K+1)]^c S(K+1)} (Q^{(K+1)}_{S(K+1)S(K+1)})^{-1} [W^{(K+1)}_{S(K+1)} + R^{(K+1)}_{S(K+1)}] \tag{77}
$$

$$
+ Q^{(K+1)}_{[S(K+1)]^c S(K+1)} (Q^{(K+1)}_{S(K+1)S(K+1)})^{-1} \hat{z}^{(K+1)}_{S(K+1)}. \tag{78}
$$

By the assumptions $\left\lVert Q^{(K+1)}_{[S(K+1)]^c S(K+1)} (Q^{(K+1)}_{S(K+1)S(K+1)})^{-1} \right\rVert_\infty \leq 1 - \alpha^{(K+1)}$, and using the fact that $\lVert \hat{z}^{(K+1)}_{S(K+1)} \rVert_\infty = 1$, we have

$$
\lVert \hat{z}^{(K+1),S}_{[S(K+1)]^c} \rVert_\infty \leq (1 - \alpha^{(K+1)}) + (2 - \alpha^{(K+1)}) \left[ \frac{\lVert R^{(K+1)} \rVert_\infty}{\lambda^{(K+1)}} + \frac{\lVert W^{(K+1)} \rVert_\infty}{\lambda^{(K+1)}} \right]. \tag{79}
$$

**Strict Dual Feasibility.** Now, to satisfy the *strict dual feasibility* $\lVert \hat{z}^{(K+1),S}_{[S(K+1)]^c} \rVert_\infty < 1$, we need to bound $\frac{\lVert W^{(K+1)} \rVert_\infty}{\lambda^{(K+1)}}$ and $\frac{\lVert R^{(K+1)} \rVert_\infty}{\lambda^{(K+1)}}$. The following two lemmas show that $\frac{\lVert W^{(K+1)} \rVert_\infty}{\lambda^{(K+1)}}$ decays to 0 at an exponential rate and $\frac{\lVert R^{(K+1)} \rVert_\infty}{\lambda^{(K+1)}}$ can be bounded deterministically accordingly under some conditions.

**Lemma F.5** (Decaying behavior of $W^{(K+1)}$). *For the specified mutual incoherence parameter $\alpha^{(K+1)} \in (0,1]$, we have*

$$
\mathbb{P} \left[ \frac{2 - \alpha^{(K+1)}}{\lambda_n} \lVert W^{(K+1)} \rVert_\infty > \frac{\alpha^{(K+1)}}{4} \right] \leq 2 \exp \left( -\frac{(\alpha^{(K+1)})^2 (\lambda^{(K+1)})^2}{128(2 - \alpha^{(K+1)})^2} n^{(K+1)} + \log(d) \right), \tag{80}
$$

*which converges to 0 at rate $\exp(-c(\lambda^{(K+1)})^2 n^{(K+1)})$ for some constant $c$, as long as $\lambda^{(K+1)} \geq \frac{16(2 - \alpha^{(K+1)})}{\alpha^{(K+1)}} \sqrt{\frac{\log(d)}{n^{(K+1)}}}$.*

The proof of this lemma is in Section I.2.1

**Lemma F.6** (Control on the remainder term $R^{(K+1)}$). *If $\lambda^{(K+1)} d \leq \frac{(C^{(K+1)}_{\min})^2}{100 D^{(K+1)}_{\max}} \frac{\alpha^{(K+1)}}{2 - \alpha^{(K+1)}}$ and $\lVert W^{(K+1)} \rVert_\infty \leq \frac{\lambda^{(K+1)}}{4}$, then*

$$
\frac{\lVert R^{(K+1)} \rVert_\infty}{\lambda^{(K+1)}} \leq \frac{25 D^{(K+1)}_{\max}}{(C^{(K+1)}_{\min})^2} \lambda^{(K+1)} d \leq \frac{\alpha^{(K+1)}}{4(2 - \alpha^{(K+1)})} \tag{81}
$$

The proof of this lemma is in Section I.2.2.

Next, applying Lemmas F.5 and F.6, we have the *strict dual feasibility* as

$$
\lVert \hat{z}^{(K+1),S}_{[S(K+1)]^c} \rVert_\infty \leq (1 - \alpha^{(K+1)}) + \frac{\alpha^{(K+1)}}{4} + \frac{\alpha^{(K+1)}}{4}
$$

$$
= 1 - \frac{\alpha^{(K+1)}}{2},
$$

with probability converging to one.

**Correct Sign Recovery.** For the statement of *correct sign recovery* in Proposition F.4, we show here that our primal sub-vector $\hat{\theta}_{S^{(K+1)}}^{(K+1)}$ defined by (63) satisfies sign consistency $\text{sign}(\hat{\theta}_{S^{(K+1)}}^{(K+1)}) = \text{sign}(\bar{\theta}_{S^{(K+1)}}^{(K+1)})$, which suffices to show that

$$\|\hat{\theta}_{S^{(K+1)}}^{(K+1)} - \bar{\theta}_{S^{(K+1)}}^{(K+1)}\|_\infty \leq \frac{\bar{\theta}_{\min}^{(K+1)}}{2},$$

where $\bar{\theta}_{\min}^{(K+1)} := \min_{(r,t) \in S^{(K+1)}} |\bar{\theta}_{rt}^{(K+1)}|$. The following lemma is used in the proof here, which establishes that the sub-vector $\hat{\theta}_{S^{(K+1)}}^{(K+1)}$ is an $\ell_2$-consistent estimate of the true common sub-vector $\bar{\theta}_{S^{(K+1)}}^{(K+1)}$.

**Lemma F.7** ($\ell_2$-consistency of primal sub-vector)**.** *If* $\lambda^{(K+1)}d \leq \frac{C_{\min}^2}{10 D_{\max}^{(K+1)}}$ *and* $\|W^{(K+1)}\|_\infty \leq \frac{\lambda^{(K+1)}}{4}$, *then*

$$\|\hat{\theta}_{S^{(K+1)}}^{(K+1)} - \bar{\theta}_{S^{(K+1)}}^{(K+1)}\|_2 \leq 5 C_{\min}^{(K+1)} \sqrt{d} \lambda^{(K+1)} \tag{82}$$

The proof of this lemma is in Section I.2.3.

By Lemma F.7, we can write

$$\frac{2}{\bar{\theta}_{\min}^{(K+1)}} \|\hat{\theta}_{S^{(K+1)}}^{(K+1)} - \bar{\theta}_{S^{(K+1)}}^{(K+1)}\|_\infty \leq \frac{2}{\bar{\theta}_{\min}^{(K+1)}} \|\hat{\theta}_{S^{(K+1)}}^{(K+1)} - \bar{\theta}_{S^{(K+1)}}^{(K+1)}\|_2$$

$$\leq \frac{2}{\bar{\theta}_{\min}^{(K+1)}} \frac{5}{C_{\min}^{(K+1)}} \sqrt{d} \lambda^{(K+1)},$$

which is less than 1 as long as $|\bar{\theta}_{rt}^{(K+1)}| \geq \frac{10}{C_{\min}^{(K+1)}} \sqrt{d} \lambda^{(K+1)}$.

Then we can use the uniform convergence of sample information matrices (in Section F.2) and Proposition F.4 (from Section F.3) to finish the proof of Theorem 4.9.

## G   Proof of Theorem 4.11

For simplicity, assume $|S| = d$. (A similar proof can be carried out with $|S| = C_1 d$ and $\Gamma \in [-\frac{1}{C_1 d^3 \log d}, \frac{1}{C_1 d^3 \log d}]^{p \times p}$ instead.) According to the definition of $\mathcal{E}$, we know that $|\mathcal{E}| = 2^{|S|/2} = 2^{d/2}$. Since $E^{(K+1)}$ is uniformly distributed on $\mathcal{E}$, the entropy of $E^{(K+1)}$ given $\Gamma$ is

$$H(E^{(K+1)}|\Gamma) = \log |\mathcal{E}| \geq \frac{d}{2} \log 2 \tag{83}$$

Now let $\mathbf{X} := \{X_t\}_{1 \leq t \leq n}$ be the samples from a $p$-dimensional Ising models with parameters $\bar{\theta}$ generated according to Theorem 4.11. For the mutual information $\mathbb{I}(\mathbf{X}; E^{(K+1)}|\Gamma)$, we have the following bound:

$$\mathbb{I}(\mathbf{X}; E^{(K+1)}|\Gamma) \leq \frac{1}{|\mathcal{E}|^2} \sum_{E^{(K+1)}} \sum_{\tilde{E}^{(K+1)}} \mathbb{KL}(P_{\mathbf{X}|E^{(K+1)},\Gamma} \| P_{\mathbf{X}|\tilde{E}^{(K+1)},\Gamma})$$

$$= \frac{1}{|\mathcal{E}|^2} \sum_{E^{(K+1)}} \sum_{\tilde{E}^{(K+1)}} \sum_{t=1}^{n} \mathbb{KL}(P_{X_t|E^{(K+1)},\Gamma} \| P_{X_t|\tilde{E}^{(K+1)},\Gamma}) \tag{84}$$

According to Lemma 19 in (Honorio, 2011), $P_{X_t|E^{(K+1)},\Gamma}$ is $(\ell_\infty, 2)$-Lipschitz continuous for $\forall E^{(K+1)} \in \mathcal{E}$. Then by Theorem 7 in (Honorio, 2011), we have

$$\mathbb{KL}\left(P_{X_t|E^{(K+1)},\Gamma} \Big\| P_{X_t|\tilde{E}^{(K+1)},\Gamma}\right) \leq 2\|\bar{\theta}_S - \bar{\theta}_S'\|_1 \leq \frac{2}{d^2 \log d}, \tag{85}$$

where the second inequality follows by the definition of $\bar{\theta}$ and $\Gamma \in [-\frac{1}{d^3 \log d}, \frac{1}{d^3 \log d}]^{p \times p}$ in Theorem 4.11. Putting (85) back to (84) gives

$$\mathbb{I}(\mathbf{X}; E^{(K+1)}|\boldsymbol{\Gamma}) \leq \frac{1}{|\mathcal{E}|^2} \sum_E \sum_{E'} \sum_{t=1}^n \frac{2}{d^2 \log d} = \frac{2n}{d^2 \log d} \tag{86}$$

Define $\hat{E}^{(K+1)} := \{(i,j) \in \hat{S}^{(K+1)} : i \neq j\}$. By applying Theorem 1 in (Ghoshal & Honorio, 2017), we get

$$\begin{aligned}
\mathbb{P}\{S^{(K+1)} \neq \hat{S}^{(K+1)}\} \geq &\mathbb{P}\{E^{(K+1)} \neq \hat{E}^{(K+1)}\} \\
\geq &1 - \frac{\mathbb{I}(\mathbf{X}; E^{(K+1)}|\Gamma) + \log 2}{H(E^{(K+1)}|\Gamma)} \\
\geq &1 - \frac{\frac{2n}{d^2 \log d} + \log 2}{\log |\mathcal{E}|} \\
= &1 - \frac{\frac{2n}{d^2 \log d} + \log 2}{\frac{d}{2} \log 2} \\
= &1 - \frac{4n}{(\log 2)(d^3 \log d)} - \frac{2}{d}.
\end{aligned}$$

## H  Proof of Lemmas for Theorem 4.7

### H.1  Proof of Lemmas for Uniform Convergence of Sample Information Matrices in Auxiliary Tasks

#### H.1.1  Proof of Lemma D.2

*Proof.* The $(j,l)^{th}$ element of the difference matrix $Q^N(\bar{\theta}) - \bar{Q}(\bar{\theta})$ can be written as an i.i.d. sum of the form $Z_{jl} = \frac{1}{K} \sum_{k=1}^K \frac{1}{n} \sum_{i=1}^n Z_{jl,i}^{(k)}$, where each $Z_{jl,i}^{(k)}$ is zero-mean and bounded (in particular, $|Z_{jl,i}^{(k)}| \leq 4$). By the Azuma-Hoeffding's bound (Hoeffding, 1994), for any indices $j,l = 1, \ldots, d$ and for any $\varepsilon > 0$, we have

$$\mathbb{P}[(Z_{jl})^2 \geq \varepsilon^2] = \mathbb{P}\left[\left|\frac{1}{K}\sum_{k=1}^K \frac{1}{n}\sum_{i=1}^n Z_{jl,i}^{(k)}\right| \geq \varepsilon\right] \leq 2\exp\left(-\frac{\varepsilon^2 nK}{32}\right). \tag{87}$$

By the Courant-Fischer variational representation (Horn & Johnson, 2012),

$$\begin{aligned}
\Lambda_{\min}(\bar{Q}_{SS}) &= \min_{\|x\|_2=1} x^T \bar{Q}_{SS} x \\
&= \min_{\|x\|_2=1} \{x^T Q_{SS}^N x + x^T(\bar{Q}_{SS} - Q_{SS}^N)x\} \\
&\leq y^T Q_{SS}^N y + y^T(\bar{Q}_{SS} - Q_{SS}^N)y,
\end{aligned}$$

where $y \in \mathbb{R}^d$ is a unit-norm minimal eigenvector of $Q_{SS}^N$. Therefore, we have

$$\Lambda_{\min}(Q_{SS}^N) \geq \Lambda_{\min}(\bar{Q}_{SS}) - \left\|\left\|\bar{Q}_{SS} - Q_{SS}^N\right\|\right\|_2 \geq C_{\min} - \left\|\left\|\bar{Q}_{SS} - Q_{SS}^N\right\|\right\|_2.$$

Observe that

$$\left\|\left\|Q_{SS}^N - \bar{Q}_{SS}\right\|\right\|_2 \leq \left(\sum_{j=1}^d \sum_{l=1}^d (Z_{jl})^2\right)^{1/2}.$$

Setting $\varepsilon^2 = \delta^2/d^2$ in (87) and applying the union bound over the $d^2$ index pairs $(j,l)$ then yields

$$\mathbb{P}\left[\left\|\left\|Q_{SS}^N - \bar{Q}_{SS}\right\|\right\|_2 \geq \delta\right] \leq 2\exp\left(-\frac{\delta^2 nK}{32d^2} + 2\log(d)\right). \tag{88}$$

So, we have the first concentration inequality in Lemma H.1.1:

$$\mathbb{P}[\Lambda_{\min}(Q_{SS}^N) \leq C_{\min} - \delta] \leq 2\exp\left(-\frac{\delta^2 nK}{32d^2} + 2\log(d)\right). \tag{89}$$

Now, for the second concentration inequality about maximum eigenvalue of the sample covariance matrix, with the same reasoning from the Courant-Fischer variational representation (Horn & Johnson, 2012), we have, for $1 \leq k \leq K$,

$$
\begin{aligned}
\Lambda_{\max}(\mathbb{E}[\frac{1}{K}\sum_{k=1}^{K}X_{\backslash r}^{(k)}(X_{\backslash r}^{(k)})^T]) &= \max_{\|v\|_2=1} v^T\mathbb{E}[\frac{1}{K}\sum_{k=1}^{K}X_{\backslash r}^{(k)}(X_{\backslash r}^{(k)})^T]v \\
&= \max_{\|v\|_2=1}\{v^T(\frac{1}{K}\sum_{k=1}^{K}\frac{1}{n}\sum_{i=1}^{n}x_{i,\backslash r}^{(k)}(x_{i,\backslash r}^{(k)})^T)v \\
&\quad + v^T(\mathbb{E}[\frac{1}{K}\sum_{k=1}^{K}X_{\backslash r}^{(k)}(X_{\backslash r}^{(k)})^T] - \frac{1}{K}\sum_{k=1}^{K}\frac{1}{n}\sum_{i=1}^{n}x_{i,\backslash r}^{(k)}(x_{i,\backslash r}^{(k)})^T)v\} \\
&\geq u^T(\frac{1}{K}\sum_{k=1}^{K}\frac{1}{n}\sum_{i=1}^{n}x_{i,\backslash r}^{(k)}(x_{i,\backslash r}^{(k)})^T)u \\
&\quad + u^T(\mathbb{E}[\frac{1}{K}\sum_{k=1}^{K}X_{\backslash r}^{(k)}(X_{\backslash r}^{(k)})^T] - \frac{1}{K}\sum_{k=1}^{K}\frac{1}{n}\sum_{i=1}^{n}x_{i,\backslash r}^{(k)}(x_{i,\backslash r}^{(k)})^T)u,
\end{aligned}
$$

where $u \in \mathbb{R}^d$ is a unit-norm maximal eigenvector of $\frac{1}{K}\sum_{k=1}^{K}\frac{1}{n}\sum_{i=1}^{n}x_{i,\backslash r}^{(k)}(x_{i,\backslash r}^{(k)})^T$. Therefore, we have

$$
\begin{aligned}
&\Lambda_{\max}(\frac{1}{K}\sum_{k=1}^{K}\frac{1}{n}\sum_{i=1}^{n}x_{i,\backslash r}^{(k)}(x_{i,\backslash r}^{(k)})^T) \\
&\leq \Lambda_{\max}(\mathbb{E}[\frac{1}{K}\sum_{k=1}^{K}X_{\backslash r}^{(k)}(X_{\backslash r}^{(k)})^T]) + u^T(\frac{1}{K}\sum_{k=1}^{K}\frac{1}{n}\sum_{i=1}^{n}x_{i,\backslash r}^{(k)}(x_{i,\backslash r}^{(k)})^T - \mathbb{E}[\frac{1}{K}\sum_{k=1}^{K}X_{\backslash r}^{(k)}(X_{\backslash r}^{(k)})^T])u \\
&\leq D_{\max} + \left\|(\frac{1}{K}\sum_{k=1}^{K}\frac{1}{n}\sum_{i=1}^{n}x_{i,\backslash r}^{(k)}(x_{i,\backslash r}^{(k)})^T - \mathbb{E}[\frac{1}{K}\sum_{k=1}^{K}X_{\backslash r}^{(k)}(X_{\backslash r}^{(k)})^T])\right\|_2.
\end{aligned}
$$

The difference matrix $\frac{1}{K}\sum_{k=1}^{K}\frac{1}{n}\sum_{i=1}^{n}x_{i,\backslash r}^{(k)}(x_{i,\backslash r}^{(k)})^T - \mathbb{E}[\frac{1}{K}\sum_{k=1}^{K}X_{\backslash r}^{(k)}(X_{\backslash r}^{(k)})^T]$ can be written as an i.i.d. sum of the form $Y_{jl} = \frac{1}{K}\sum_{k=1}^{K}\frac{1}{n}\sum_{i=1}^{n}Y_{jl,i}^{(k)}$, where each $Y_{jl,i}^{(k)}$ is zero-mean and bounded (in particular, $|Y_{jl,i}^{(k)}| \leq 4$). By the Azuma-Hoeffding's bound (Hoeffding, 1994), for any indices $j, l = 1, \ldots, d$ and for any $\varepsilon > 0$, we have

$$
\mathbb{P}[(Y_{jl})^2 \geq \varepsilon^2] = \mathbb{P}\big[|\frac{1}{K}\sum_{k=1}^{K}\frac{1}{n}\sum_{i=1}^{n}Y_{jl,i}| \geq \varepsilon\big] \leq 2\exp\left(-\frac{\varepsilon^2 nK}{32}\right). \tag{90}
$$

Observe that

$$
\left\|\frac{1}{K}\sum_{k=1}^{K}\frac{1}{n}\sum_{i=1}^{n}x_{i,\backslash r}^{(k)}(x_{i,\backslash r}^{(k)})^T - \mathbb{E}[\frac{1}{K}\sum_{k=1}^{K}X_{\backslash r}^{(k)}(X_{\backslash r}^{(k)})^T]\right\|_2 \leq \left(\sum_{j=1}^{d}\sum_{l=1}^{d}(Y_{jl})^2\right)^{1/2}.
$$

Setting $\varepsilon^2 = \delta^2/d^2$ in (90), and applying the union bound over the $d^2$ index pairs $(j, l)$ then yields

$$
\mathbb{P}\left[\left\|\frac{1}{K}\sum_{k=1}^{K}\frac{1}{n}\sum_{i=1}^{n}x_{i,\backslash r}^{(k)}(x_{i,\backslash r}^{(k)})^T - \mathbb{E}[\frac{1}{K}\sum_{k=1}^{K}X_{\backslash r}^{(k)}(X_{\backslash r}^{(k)})^T]\right\|_2 \geq \delta\right] \leq 2\exp\left(-\frac{\delta^2 nK}{32d^2} + 2\log(d)\right).
$$

So we have

$$
\mathbb{P}\big[\Lambda_{\max}[\frac{1}{n}\sum_{i=1}^{n}x_{i,\backslash r}^{(k)}(x_{i,\backslash r}^{(k)})^T] \geq D_{\max} + \delta\big] \leq 2\exp\left(-\frac{\delta^2 nK}{32d^2} + 2\log(d)\right)
$$

as stated in the lemma. $\qquad\square$

### H.1.2 Proof of Lemma D.3

We begin the proof of this lemma by decomposing the sample matrix as the sum $Q^N_{S^cS}(Q^N_{SS})^{-1} = T_1 + T_2 + T_3 + T_4$, where we define

$$T_1 := \bar{Q}_{S^cS}[(Q^N_{SS})^{-1} - (\bar{Q}_{SS})^{-1}], \tag{91a}$$

$$T_2 := [Q^N_{S^cS} - \bar{Q}_{S^cS}](\bar{Q}_{SS})^{-1}, \tag{91b}$$

$$T_3 := [Q^N_{S^cS} - \bar{Q}_{S^cS}][(Q^N_{SS})^{-1} - (\bar{Q}_{SS})^{-1}], \tag{91c}$$

$$T_4 := \bar{Q}_{S^cS}(\bar{Q}_{SS})^{-1}. \tag{91d}$$

The fourth term is controlled by the incoherence assumption in Assumption 4.1:

$$\|\|T_4\|\|_\infty = \left\|\left\|\bar{Q}_{S^cS}(\bar{Q}_{SS})^{-1}\right\|\right\|_\infty \leq 1 - \alpha.$$

If we can show that $\|\|T_i\|\|_\infty \leq \frac{\alpha}{6}$ for the remaining indices $i = 1, 2, 3$, then by our four-term decomposition and the triangle inequality, the sample version can satisfy the desired bound (41). To deal with these remaining terms, we make use of the following lemma:

**Lemma H.1.** *For any $\delta > 0$, and constants $B, B_1, B_2$, the following bounds hold*

$$\mathbb{P}[\|\|Q^N_{S^cS} - \bar{Q}_{S^cS}\|\|_\infty \geq \delta] \leq 2 \exp\left(-B\frac{\delta^2 nK}{d^2} + \log(d) + \log(p)\right), \tag{92a}$$

$$\mathbb{P}[\|\|Q^N_{SS} - \bar{Q}_{SS}\|\|_\infty \geq \delta] \leq 2 \exp\left(-B\frac{\delta^2 nK}{d^2} + 2\log(d)\right), \tag{92b}$$

$$\mathbb{P}[\left\|\left\|(Q^N_{SS})^{-1} - (\bar{Q}_{SS})^{-1}\right\|\right\|_\infty \geq \delta] \leq 4 \exp\left(-B_1\frac{nK\delta^2}{d^3} + B_2\log(d)\right). \tag{92c}$$

See Section J.1 for the proof of these claims.

**Control of the first term.** For the first term, we re-factorize it as

$$T_1 = \bar{Q}_{S^cS}(\bar{Q}_{SS})^{-1}[\bar{Q}_{SS} - Q^N_{SS}](Q^N_{SS})^{-1}.$$

Then,

$$\|\|T_1\|\|_\infty \leq \left\|\left\|\bar{Q}_{S^cS}(\bar{Q}_{SS})^{-1}\right\|\right\|_\infty \|\|\bar{Q}_{SS} - Q^N_{SS}\|\|_\infty \left\|\left\|(Q^N_{SS})^{-1}\right\|\right\|_\infty$$

$$\leq (1 - \alpha)\|\|\bar{Q}_{SS} - Q^N_{SS}\|\|_\infty \{\sqrt{d}\left\|\left\|(Q^N_{SS})^{-1}\right\|\right\|_2\},$$

where we have used the incoherence assumption in Assumption 4.1. Using the bound (40) from Lemma (D.2) with $\delta = C_{\min}/2$, we have $\left\|\left\|(Q^N_{SS})^{-1}\right\|\right\|_2 = [\Lambda_{\min}(Q^N_{SS})]^{-1} \leq \frac{2}{C_{\min}}$ with probability greater than $1 - 2\exp(-BnK/d^2 + 2\log(d))$. Next, applying the bound (92b) with $\delta = c/\sqrt{d}$, we conclude that with probability greater than $1 - 2\exp(-BnKc^2/d^3 + 2\log(d))$, we have

$$\|\|\bar{Q}_{SS} - Q^N_{SS}\|\|_\infty \leq c/\sqrt{d}.$$

By choosing the constant $c > 0$ sufficiently small, we are guaranteed that

$$\mathbb{P}[\|\|T_1\|\|_\infty \geq \alpha/6] \leq 2\exp\left(-B\frac{nKc^2}{d^3} + \log(d)\right).$$

**Control of the second term.** To bound $T_2$, we first write

$$\|T_2\|_\infty \leq \sqrt{d} \left\|\left(\bar{Q}_{SS}\right)^{-1}\right\|_2 \|Q^N_{S^cS} - \bar{Q}_{S^cS}\|_\infty$$

$$\leq \frac{\sqrt{d}}{C_{\min}} \|Q^N_{S^cS} - \bar{Q}_{S^cS}\|_\infty.$$

Then apply the bound (92a) with $\delta = \frac{\alpha}{6} \frac{C_{\min}}{\sqrt{d}}$ to conclude that

$$\mathbb{P}[\|T_2\|_\infty \geq \alpha/6] \leq 2\exp\left(-B\frac{nK}{d^3} + \log(p)\right).$$

**Control of the third term.** Finally, in order to bound the third term $T_3$, we apply the bounds (92a) and (92c), both with $\delta = \sqrt{\alpha/6}$ and use the fact that $\log d \leq \log p$ to conclude that

$$\mathbb{P}[\|T_3\|_\infty \geq \alpha/6] \leq 4\exp\left(-B\frac{nK}{d^3} + \log(p)\right). \tag{93}$$

Putting together the four pieces, we conclude that

$$\mathbb{P}\left[\left\|Q^N_{S^cS}(Q^N_{SS})^{-1}\right\|_\infty \geq 1 - \alpha/2\right] = O\left(\exp\left(-B\frac{nK}{d^3} + \log(p)\right)\right)$$

## H.2 Proof of Lemmas for Proposition D.4

### H.2.1 Proof of Lemma D.5

*Proof.* By definition of $W^N$ (see (43)), we have

$$\|W^N\|_\infty = \|\nabla\ell(\bar{\theta}; \{\mathfrak{X}^n_1\}^K_1)\|_\infty = \|\frac{1}{K}\sum_{k=1}^K \nabla\ell^{(k)}(\bar{\theta}; \{\mathfrak{X}^n_1\}^K_1)\|_\infty. \tag{94}$$

which can be decompose into two parts as follows

$$\|\nabla\ell(\bar{\theta}; \mathfrak{X}^n_1\}^K_1)\|_\infty$$

$$\leq \|\underbrace{\frac{1}{K}\sum_{k=1}^K \left\{\nabla\ell^{(k)}(\bar{\theta}; \{\mathfrak{X}^n_1\}^{(k)}) - \nabla\ell^{(k)}(\bar{\theta}^{(k)}; \{\mathfrak{X}^n_1\}^{(k)})\right\}}_{Y_1}\|_\infty + \|\underbrace{\frac{1}{K}\sum_{k=1}^K \nabla\ell^{(k)}(\bar{\theta}^{(k)}; \{\mathfrak{X}^n_1\}^{(k)})}_{Y_2}\|_\infty. \tag{95}$$

We then bound the two terms $\|Y_1\|_\infty$ and $\|Y_2\|_\infty$ respectively.

**Bounding $\|Y_2\|_\infty$** .

Note that the conditional expectation of $Y_2$ given $\{\Delta^{(k)}\}^K_1$ is

$$\mathbb{E}[Y_2|\{\Delta^{(k)}\}^K_1] = \mathbb{E}[\frac{1}{K}\sum_{k=1}^K \nabla\ell^{(k)}(\bar{\theta}^{(k)}; \{\mathfrak{X}^n_1\}^{(k)})|\{\Delta^{(k)}\}^K_1]$$

$$= \frac{1}{K}\sum_{k=1}^K \mathbb{E}[\nabla\ell^{(k)}(\bar{\theta}^{(k)}; \{\mathfrak{X}^n_1\}^{(k)})|\Delta^{(k)}]$$

$$= \frac{1}{K}\sum_{k=1}^K 0$$

$$= 0,$$

where the second to last line comes from the fact that the expected gradient at the true parameter of each task is 0. This property can also be checked by expanding the expression of $Y_2$. Each entry of $Y_2$, denoted by $Y_{2,u}$, for $1 \le u \le p-1$, can be expressed as a sum of random variables $Z_{i,u}^{(k)}$:

$$Y_{2,u} = \frac{1}{K} \sum_{k=1}^{K} \frac{1}{n} \sum_{i=1}^{n} Z_{i,u}^{(k)}, \tag{96}$$

where

$$Z_{i,u}^{(k)} = x_{i,u}^{(k)} \{ x_{i,r}^{(k)} - \frac{\exp(\sum_{t \in V \backslash r} \bar{\theta}_{rt}^{(k)} x_{i,t}^{(k)}) - \exp(-\sum_{t \in V \backslash r} \bar{\theta}_{rt}^{(k)} x_{i,t}^{(k)})}{\exp(\sum_{t \in V \backslash r} \bar{\theta}_{rt}^{(k)} x_{i,t}^{(k)}) + \exp(-\sum_{t \in V \backslash r} \bar{\theta}_{rt}^{(k)} x_{i,t}^{(k)})} \}$$
$$= x_{i,u}^{(k)} \{ x_{i,r}^{(k)} - \mathbb{P}_{\bar{\theta}^{(k)}}[X_r^{(k)} = 1 | x_{i,\backslash r}^{(k)}] + \mathbb{P}_{\bar{\theta}^{(k)}}[X_r^{(k)} = -1 | x_{i,\backslash r}^{(k)}] \}.$$

We have the conditional expectation $\mathbb{E}[Z_{i,u}^{(k)} | \Delta^{(k)}] = 0$ by applying another law of total expectation (Weiss et al., 2005) with the inner conditional expectation of $X_r^{(k)}$ given $X_{\backslash r}^{(k)}$ and the outer total expectation being the marginal joint expectation of $X_{\backslash r}^{(k)}$. So we have the total expectation

$$\mathbb{E}[Z_{i,u}^{(k)}] = \mathbb{E}[\mathbb{E}[Z_{i,u}^{(k)} | \Delta^{(k)}]] = \mathbb{E}[0] = 0. \tag{97}$$

Also, from the expression of $Z_{i,u}^{(k)}$, since all samples are either $-1$ or $+1$, it is easy to see that $|Z_{i,u}^{(k)}| \le 2$. On the other hand, note that the total $nK$ samples $\{X_i^{(k)}\}_{1 \le i \le n, 1 \le k \le K}$ are conditionally independent given $\{\Delta^{(k)}\}_1^K$ ( $\{\Delta^{(k)}\}_1^K$ are the *latent random variables*). We can then apply the Hoeffding's Inequality with latent conditional independence (LCI), Corollary 1 in Ke & Honorio (2019) by conditioning on the *latent random variables* $\{\Delta^k\}_{k=1}^K$ to get

$$\mathbb{P}[\sum_{k=1}^{K} \sum_{i=1}^{n} (Z_{i,u}^{(k)} - 0) \ge \delta] \le \exp\left(-\frac{\delta^2}{8nK}\right), \tag{98}$$

for any $\delta > 0$. Substituting $Y_{2,u}$ in (96) and by the symmetry of it (resulting from the symmetry of the binary random variables $\{X_i^{(k)}\}_{1 \le i \le n, 1 \le k \le K}$), we have

$$\mathbb{P}[|Y_{2,u}| \ge \delta] = \mathbb{P}[Y_{2,u} \ge \delta \quad \text{or} \quad Y_{2,u} \le -\delta]$$
$$\le \mathbb{P}[Y_{2,u} \ge \delta] + \mathbb{P}[Y_{2,u} \le -\delta]$$
$$= 2\mathbb{P}[Y_{2,u} \ge \delta]$$
$$= 2\mathbb{P}[\frac{1}{nK} \sum_{k=1}^{K} \sum_{i=1}^{n} (Z_{i,u}^{(k)} - 0) \ge \delta]$$
$$\le 2\exp\left(-\frac{\delta^2}{8nK}\right).$$

After that, applying union bound over the indices $u$ of $Y_2$ yields

$$\mathbb{P}[\|Y_2\|_\infty \ge \delta] \le 2\exp\left(-\frac{\delta^2}{8nK} + \log p\right). \tag{99}$$

**Bounding $\|Y_1\|_\infty$** .

Note that $Y_1 = \nabla \ell(\bar{\theta}_{\backslash r}) - \frac{1}{K} \sum_{k=1}^{K} \nabla \ell^{(k)}(\bar{\theta}_{\backslash r}^{(k)})$ using the shorthand notations in (19). We can bound $\|Y_2\|_\infty$ by writing

$$\|Y_1\|_\infty = \|Y_1 - \mathbb{E}(Y_1) + \mathbb{E}(Y_1)\|_\infty$$
$$\le \|Y_1 - \mathbb{E}(Y_1)\|_\infty + \|\mathbb{E}(Y_1)\|_\infty. \tag{100}$$

Using Assumption 4.3 by setting $\delta = \sqrt{\frac{8\log(2p/\varepsilon)}{nK}}$, we have

$$\mathbb{P}(\|\mathbb{E}(Y_1)\|_\infty \geq \delta) \leq 2\exp\left(-\frac{\delta^2 nK}{8} + \log p\right). \tag{101}$$

Notice that for $Y_1$, we can also decompose it into a sum of random variables $Z_{i,u}^{\prime(k)}$

$$Y_{1,u} = \frac{1}{K}\sum_{k=1}^K \frac{1}{n}\sum_{i=1}^n Z_{i,u}^{\prime(k)}, \tag{102}$$

where

$$
\begin{aligned}
Z_{i,u}^{\prime(k)} &= x_{i,u}^{(k)}\{x_{i,r}^{(k)} - \frac{\exp(\sum_{t\in V\backslash r}\bar{\theta}_{rt}x_{i,t}^{(k)}) - \exp(-\sum_{t\in V\backslash r}\bar{\theta}_{rt}x_{i,t}^{(k)})}{\exp(\sum_{t\in V\backslash r}\bar{\theta}_{rt}x_{i,t}^{(k)}) + \exp(-\sum_{t\in V\backslash r}\bar{\theta}_{rt}x_{i,t}^{(k)})}\} \\
&= x_{i,u}^{(k)}\{x_{i,r}^{(k)} - \mathbb{P}_{\bar{\theta}}[X_r = 1|x_{i,\backslash r}^{(k)}] + \mathbb{P}_{\bar{\theta}}[X_r = -1|x_{i,\backslash r}^{(k)}]\}.
\end{aligned}
$$

Here $\mathbb{P}_{\bar{\theta}}$ denotes the conditional probability of the random variable associated with node $r$ taking on $-1$ or $+1$ given a $(p-1)$-dimensional data vector values $x_{i,\backslash r}^{(k)}$, supposing the true parameter is $\bar{\theta}_{\backslash r}$. In this way, we can write each entry of $Y_1 - \mathbb{E}(Y_1)$ as

$$
\begin{aligned}
Y_{1,u} - \mathbb{E}(Y_{1,u}) &= \frac{1}{K}\sum_{k=1}^K\frac{1}{n}\sum_{i=1}^n Z_{i,u}^{\prime(k)} - \mathbb{E}[\frac{1}{K}\sum_{k=1}^K\frac{1}{n}\sum_{i=1}^n Z_{i,u}^{\prime(k)}] \\
&= \frac{1}{K}\sum_{k=1}^K\frac{1}{n}\sum_{i=1}^n Z_{i,u}^{\prime(k)} - \mathbb{E}[Z_{i,u}^{\prime(k)}].
\end{aligned}
$$

Then we define random variable

$$H_{i,u}^{(k)} := Z_{i,u}^{\prime(k)} - \mathbb{E}[Z_{i,u}^{\prime(k)}] \tag{103}$$

for all $1\leq i\leq n, 1\leq k\leq K, 1\leq u\leq p-1$. We have

$$\mathbb{E}[H_{i,u}^{(k)}] = \mathbb{E}[Z_{i,u}^{\prime(k)} - \mathbb{E}[Z_{i,u}^{\prime(k)}]] = \mathbb{E}[Z_{i,u}^{\prime(k)}] - \mathbb{E}[Z_{i,u}^{\prime(k)}] = 0. \tag{104}$$

Since the expected value $\mathbb{E}[Z_{i,u}^{\prime(k)}]$ is deterministic, the randomness of $H_{i,u}^{(k)}$ takes on the same pattern as $Z_{i,u}^{\prime(k)}$, so they are conditionally independent given $\{\Delta^{(k)}\}_1^K$. In addition, $H_{i,u}^{(k)}$ is bounded in the sense that $|H_{i,u}^{(k)}| \leq 6$. By using LCI Hoeffding's inequality (Ke & Honorio, 2019) again, we get

$$\mathbb{P}[\sum_{k=1}^K\sum_{i=1}^n(H_{i,u}^{(k)} - 0) \geq \delta] \leq \exp\left(-\frac{\delta^2}{72nK}\right). \tag{105}$$

Using the same reasoning (symmetry and union bound) in proving the bound for $\|Y_2\|_\infty$, we get

$$\mathbb{P}[\|Y_1 - \mathbb{E}[Y_1]\|_\infty \geq \delta] \leq 2\exp\left(-\frac{\delta^2}{72nK} + \log p\right). \tag{106}$$

Next, putting the terms $\|\mathbb{E}[Y_1]\|_\infty$ and $\|Y_1 - \mathbb{E}[Y_1]\|_\infty$ together, we have

$$
\begin{aligned}
\mathbb{P}(\|Y_1\|_\infty > 2\delta) &= 1 - \mathbb{P}(\|Y_1\|_\infty < 2\delta) \\
&\leq 1 - \mathbb{P}(\|\mathbb{E}[Y_1]\|_\infty + \|Y_1 - \mathbb{E}[Y_1]\|_\infty < 2\delta) \\
&\leq 1 - \mathbb{P}(\|\mathbb{E}[Y_1]\|_\infty < \delta \quad \text{and} \quad \|Y_1 - \mathbb{E}[Y_1]\|_\infty < \delta) \\
&= \mathbb{P}(\|\mathbb{E}[Y_1]\|_\infty \geq \delta \quad \text{or} \quad \|Y_1 - \mathbb{E}[Y_1]\|_\infty \geq \delta) \\
&\leq \mathbb{P}(\|\mathbb{E}[Y_1]\|_\infty \geq \delta) + \|Y_1 - \mathbb{E}[Y_1]\|_\infty \geq \delta).
\end{aligned}
$$

By the same token, we get

$$\mathbb{P}(\|W^N\|_\infty > 3\delta) \le \mathbb{P}(\|Y_2\|_\infty > \delta) + \mathbb{P}(\|\mathbb{E}[Y_1]\|_\infty \ge \delta) + \|Y_1 - \mathbb{E}[Y_1]\|_\infty \ge \delta) \tag{107}$$

$$\le 4\exp\left(-\frac{\delta^2}{8nK} + \log p)\right) + 2\exp\left(-\frac{\delta^2}{72nK} + \log p\right) \tag{108}$$

$$\le 6\exp\left(-\frac{\delta^2}{72nK} + \log p\right). \tag{109}$$

Finally, setting $3\delta = \frac{\alpha\lambda}{4(2-\alpha)}$, we obtain

$$\mathbb{P}\left(\frac{2-\alpha}{\lambda}\|W^N\|_\infty > \frac{\alpha}{4}\right) \le 6\exp\left(-\frac{\alpha^2\lambda^2}{c(2-\alpha)^2}nK + log(p)\right), \tag{110}$$

for some fixed constant $c$ as in Lemma D.5. $\qquad\square$

### H.2.2 Proof of Lemma D.6

*Proof.* We first show that the remainder term $R^N$ satisfies the bound $\|R^N\|_\infty \le D_{\max}\|\hat\theta_S - \bar\theta_S\|_2^2$. Then the result of Lemma D.7, namely $\|\hat\theta_S - \bar\theta_S\|_2 \le \frac{5}{C_{\min}}\sqrt{d}\lambda$, can be used to conclude that

$$\frac{\|R^N\|_\infty}{\lambda} \le \frac{25D_{\max}}{C_{\min}^2}\lambda d$$

as claimed in Lemma D.6.

Focusing on element $R_j^N$ for some index $j \in \{1, \ldots, p\}$, we have

$$R_j^N = -[\nabla^2\ell(\theta^{(j)}; \mathfrak{X}) - \nabla^2\ell(\bar\theta; \mathfrak{X})]_j^T(\hat\theta - \bar\theta)$$

$$= \frac{1}{K}\sum_{k=1}^K \frac{1}{n}\sum_{i=1}^n [\eta(x_i^{(k)}; \theta^{(j)}) - \eta(x_i^{(k)}; \bar\theta))](\hat\theta - \bar\theta),$$

for some point $\theta^{(j)} = \mu_j\hat\theta + (1 - \mu_j)\bar\theta$ and $\mu_j \in [0, 1]$. Then we set $g(t) = \frac{4e^{2t}}{(e^{2t}+1)^2}$ by noting that $\eta(\theta, x) = g(x_r \sum_{t \in V\backslash r} \theta_{rt}x_t)$. By the chain rule and another application of the mean value theorem, we then have

$$R_j^N = \frac{1}{K}\sum_{k=1}^K \frac{1}{n}\sum_{i=1}^n g'((\theta'^{(j)})^T x_i^{(k)})(x_i^{(k)})^T[\theta^{(j)} - \bar\theta]\{x_{i,j}^{(k)}(x_i^{(k)})^T[\hat\theta - \bar\theta]\}$$

$$= \frac{1}{K}\sum_{k=1}^K \frac{1}{n}\sum_{i=1}^n \{g'((\theta'^{(j)})^T x_i^{(k)})x_{i,j}^{(k)}\}\{[\theta^{(j)} - \bar\theta]^T x_{i,j}^{(k)}(x_i^{(k)})^T[\hat\theta - \bar\theta]\},$$

where $\theta'^{(j)}$ is another point on the line joining $\hat\theta$ and $\bar\theta$. Setting $a_i^{(k)} := \{g'((\theta'^{(j)})^T x_i^{(k)})x_{i,j}^{(k)}\}$ and $b_i^{(k)} := \{[\theta^{(j)} - \bar\theta]^T x_{i,j}^{(k)}(x_i^{(k)})^T[\hat\theta - \bar\theta]\}$, and treating $a, b$ both as $nK$-dimensional vectors, we have

$$|R_j^N| = \frac{1}{nK}|\sum_{k=1}^K \sum_{i=1}^n a_i^{(k)}b_i^{(k)}| \le \frac{1}{nK}\|a\|_\infty\|b\|_1.$$

A calculation shows that $\|a\|_\infty \le 1$, and

$$\frac{1}{nK}\|b\|_1 = \frac{1}{K}\sum_{k=1}^K \mu_j[\hat\theta - \bar\theta]^T\left\{\frac{1}{n}\sum_{i=1}^n x_i^{(k)}(x_i^{(k)})^T\right\}[\hat\theta - \bar\theta]$$

$$= \frac{1}{K}\sum_{k=1}^K \mu_j[\hat\theta_S - \bar\theta_S]^T\left\{\frac{1}{K}\sum_{k=1}^K \frac{1}{n}\sum_{i=1}^n x_{i,S}^{(k)}(x_{i,S}^{(k)})^T\right\}[\hat\theta_S - \bar\theta_S]$$

$$\le D_{\max}\|\hat\theta_S - \bar\theta_S\|_2^2,$$

where the second line uses the fact that $\hat{\theta}_{S^c} = \bar{\theta}_{S^c} = 0$. Therefore, we have

$$\|R^N\|_\infty \leq D_{\max}\|\hat{\theta}_S - \bar{\theta}_S\|_2^2$$

$\square$

### H.2.3 Proof of Lemma D.7

*Proof.* Following the method of proof in Ravikumar et al. (2010) which was also previously used in another context (Rothman et al., 2008), we define the function $G : \mathbb{R}^d \to \mathbb{R}$ by

$$G(u_S) := \ell(\bar{\theta}_S + u_S; \{\mathfrak{X}_1^n\}_1^K) - \ell(\bar{\theta}_S; \{\mathfrak{X}_1^n\}_1^K) + \lambda_n(\|\bar{\theta}_S + u_S\|_1 - \|\bar{\theta}_S\|_1). \tag{111}$$

It can be seen that $\hat{u} = \hat{\theta}_S - \bar{\theta}_S$ minimizes $G$. Moreover, $G(0) = 0$ by construction; therefore, we must have $G(\hat{u}) \leq 0$. Note that $G$ is convex. Suppose we show for some radius $B > 0$, and for $u \in \mathbb{R}^d$ with $\|u\|_2 = B$, we have $G(u) > 0$, . we then claim that $\|\hat{u}\|_2 \leq B$. In fact, if $\hat{u}$ lay outside the ball of radius $B$, then the convex combination $t\hat{u} + (1-t)(0)$ would lie on the boundary of the ball, for an appropriately chosen $t \in (0, 1)$. By convexity,

$$G(t\hat{u} + (1-t)(0)) \leq tG(\hat{u}) + (1-t)G(0) \leq 0,$$

which contradicts the assumed strict positivity of $G$ on the boundary. It thus suffices to establish strict positivity of $G$ on the boundary of the ball with radius $B = M\lambda\sqrt{d}$, where $M > 0$ is a parameter to be chosen later in the proof. Let $u \in \mathbb{R}^d$ be an arbitrary vector with $\|u\|_2 = B$. Recalling the notation $W^N := -\nabla\ell(\bar{\theta}; \{\mathfrak{X}_1^n\}_1^K)$, by a Taylor series expansion of the log likelihood component of $G$, we have

$$G(u) = -(W_S^N)^T u + u^T[\nabla^2\ell(\bar{\theta}_S + \alpha u; \{\mathfrak{X}_1^n\}_1^K)]u + \lambda_n(\|\bar{\theta}_S + u_S\|_1 - \|\bar{\theta}_S\|_1)$$

for some $\alpha \in [0, 1]$. For the first term, we have the bound

$$|(W_S^N)^T u| \leq \|W_S^N\|_\infty\|u\|_1 \leq \|W_S^N\|_\infty\sqrt{d}\|u\|_2 \leq (\lambda_n\sqrt{d})^2\frac{M}{4}, \tag{112}$$

since $\|W_S^N\|_\infty \leq \frac{\lambda_n}{4}$ by assumption. For the last term, applying triangle inequality yields

$$\lambda_n(\|\bar{\theta}_S + u_S\|_1 - \|\bar{\theta}_S\|_1) \geq -\lambda_n\|u_S\|_1.$$

Since $\|u_S\|_1 \leq \sqrt{d}\|u_S\|_2$, we have

$$\lambda_n(\|\bar{\theta}_S + u_S\|_1 - \|\bar{\theta}_S\|_1) \geq -\lambda_n\sqrt{d}\|u_S\|_2 = -M(\sqrt{d}\lambda_n)^2. \tag{113}$$

Finally, turning to the middle Hessian term, we have

$$
\begin{aligned}
q^* &:= \Lambda_{\min}(\nabla^2\ell(\bar{\theta}_S + \alpha u); \{\mathfrak{X}_1^n\}_1^K)) \\
&\geq \min_{\alpha\in[0,1]} \Lambda_{\min}(\nabla^2\ell(\bar{\theta}_S + \alpha u_S); \{\mathfrak{X}_1^n\}_1^K)) \\
&= \min_{\alpha\in[0,1]} \Lambda_{\min}\left[\frac{1}{K}\sum_{k=1}^K\frac{1}{n}\sum_{i=1}^n \eta(x_i^{(k)}; \bar{\theta}_S + \alpha u_S)x_{i,S}^{(k)}(x_{i,S}^{(k)})^T\right].
\end{aligned}
$$

By a Taylor series expansion of $\eta(x_i^{(k)}; \cdot)$, we have, for some $\alpha_0 \in [0, \alpha]$, a lower bound of $q^*$:

$$\min_{\alpha\in[0,1]} \Lambda_{\min}\{\frac{1}{nK}\sum_{k=1}^K\sum_{i=1}^n[\eta(x_i^{(k)}; \bar{\theta}_S)x_{i,S}^{(k)}(x_{i,S}^{(k)})^T$$

$$+ \alpha g'(x_{i,r}^{(k)}\sum_{t\in S\backslash r}(\bar{\theta}_{rt} + \alpha_0 u_{rt})x_{i,t}^{(k)})x_{i,r}^{(k)}(u_S^T x_{i,S}^{(k)})x_{i,S}^{(k)}(x_{i,S}^{(k)})^T]\}$$

$$\geq \Lambda_{\min}\left[\frac{1}{K}\sum_{k=1}^{K}\frac{1}{n}\sum_{i=1}^{n}\eta(x_i^{(k)};\bar{\theta}_S)x_{i,S}^{(k)}(x_{i,S}^{(k)})^T\right]$$

$$+ \min_{\alpha\in[0,1]}\alpha\Lambda_{\min}\left[\frac{1}{K}\sum_{k=1}^{K}\frac{1}{n}\sum_{i=1}^{n}g'\left(x_{i,r}^{(k)}(\bar{\theta}_S+\alpha_0 u_S)^T x_{i,S}^{(k)}\right)x_{i,r}^{(k)}(u_S^T x_{i,S}^{(k)})x_{i,S}^{(k)}(x_{i,S}^{(k)})^T\right]$$

$$\geq \Lambda_{\min}(Q_{SS}^N) - \max_{\alpha\in[0,1]}\alpha\max_{\alpha_0\in[0,\alpha]}\left\|\!\left\|\frac{1}{K}\sum_{k=1}^{K}\frac{1}{n}\sum_{i=1}^{n}g'\left(x_{i,r}^{(k)}(\bar{\theta}_S+\alpha_0 u_S)^T x_{i,S}^{(k)}\right)(u_S^T x_{i,S}^{(k)})x_{i,S}^{(k)}(x_{i,S}^{(k)})^T\right\|\!\right\|_2$$

$$\geq \Lambda_{\min}(Q_{SS}^N) - \max_{\alpha\in[0,1]}\left\|\!\left\|\frac{1}{K}\sum_{k=1}^{K}\frac{1}{n}\sum_{i=1}^{n}g'\left(x_{i,r}^{(k)}(\bar{\theta}_S+\alpha u_S)^T x_{i,S}^{(k)}\right)(u_S^T x_{i,S}^{(k)})x_{i,S}^{(k)}(x_{i,S}^{(k)})^T\right\|\!\right\|_2$$

$$\geq C_{\min} - \max_{\alpha\in[0,1]}\left\|\!\left\|\underbrace{\frac{1}{K}\sum_{k=1}^{K}\frac{1}{n}\sum_{i=1}^{n}g'\left(x_{i,r}^{(k)}(\bar{\theta}_S+\alpha u_S)^T x_{i,S}^{(k)}\right)(\langle u_S, x_{i,S}^{(k)}\rangle)x_{i,S}^{(k)}(x_{i,S}^{(k)})^T}_{A(\alpha)}\right\|\!\right\|_2$$

It remains to control the spectral norm of the matrix $A(\alpha)$, for $\alpha\in[0,1]$. For any fixed $\alpha\in[0,1]$, and $y\in\mathbb{R}$ with $\|y\|_2 = 1$, we have

$$\langle y, A(\alpha)y\rangle = \frac{1}{K}\sum_{k=1}^{K}\frac{1}{n}\sum_{i=1}^{n}g'\left(x_{i,r}^{(k)}(\bar{\theta}_S+\alpha u_S)^T x_{i,S}^{(k)}\right)[\langle u_S, x_{i,S}^{(k)}\rangle][\langle x_{i,S}^{(k)}, y\rangle]^2$$

$$\leq \frac{1}{K}\sum_{k=1}^{K}\frac{1}{n}\sum_{i=1}^{n}\left|g'\left(x_{i,r}^{(k)}(\bar{\theta}_S+\alpha u_S)^T x_{i,S}^{(k)}\right)\right||\langle u_S, x_{i,S}^{(k)}\rangle|[\langle x_{i,S}^{(k)}, y\rangle]^2$$

Note that $\left|g'\left(x_{i,r}^{(k)}(\bar{\theta}_S+\alpha u_S)^T x_{i,S}^{(k)}\right)\right| \leq 1$, and

$$|\langle u_S, x_{i,S}^{(k)}\rangle| \leq \|u_S\|_1 \leq \sqrt{d}\|u_S\|_2 = M\lambda_n d.$$

Moreover, we have

$$\frac{1}{K}\sum_{k=1}^{K}\frac{1}{n}\sum_{i=1}^{n}(\langle x_{i,S}^{(k)}, y\rangle)^2 \leq \left\|\!\left\|\frac{1}{K}\sum_{k=1}^{K}\frac{1}{n}\sum_{i=1}^{n}x_{i,S}^{(k)}(x_{i,S}^{(k)})^T\right\|\!\right\|_2 \leq D_{\max}$$

by assumption. We then obtain

$$\max_{\alpha\in[0,1]}\|\!|A(\alpha)|\!\|_2 \leq D_{\max}M\lambda_n d \leq C_{\min}/2,$$

assuming that $\lambda_n \leq \frac{C_{\min}}{2MD_{\max}d}$. Under this condition, we have shown that

$$q^* := \Lambda_{\min}(\nabla^2\ell(\bar{\theta}_S+\alpha u); \{\mathfrak{X}_1^n\}_1^K)) \geq C_{\min}/2. \tag{114}$$

Finally, combining the three terms in $G(u)$, we conclude that

$$G(u_S) \geq (\lambda_n\sqrt{d})^2\left\{-\frac{1}{4}M + \frac{C_{\min}}{2}M^2 - M\right\},$$

which is strictly positive for $M = 5/C_{\min}$. Therefore, as long as

$$\lambda_n \leq \frac{C_{\min}}{2MD_{\max}d} = \frac{C_{\min}^2}{10D_{\max}d},$$

we are guaranteed that

$$\|\hat{u}_S\|_2 \leq M\lambda_n\sqrt{d} = \frac{5}{C_{\min}}\lambda_n\sqrt{d}.$$

$\square$

# I  Proof of Lemmas for Theorem 4.9

## I.1  Proof of Lemmas for Uniform Convergence of Sample Information Matrices in Novel Task

### I.1.1  Proof of Lemma F.2

*Proof.* The $(j, l)^{th}$ element of the difference matrix $Q^{(K+1)}(\bar{\theta}_S^{(K+1)}) - \bar{Q}^{(K+1)}(\bar{\theta}_S^{(K+1)})$ can be written as an i.i.d. sum of the form $Z_{jl}^{(K+1)} = \frac{1}{n^{(K+1)}} \sum_{i=1}^{n^{(K+1)}} Z_{jl,i}^{(K+1)}$, where each $Z_{jl,i}^{(K+1)}$ is zero-mean and bounded (in particular, $|Z_{jl,i}^{(K+1)}| \leq 4$) By the Azuma-Hoeffding's bound (Hoeffding, 1994), for any indices $j, l = 1, \ldots, d$ and for any $\varepsilon > 0$, we have

$$\mathbb{P}[(Z_{jl}^{(K+1)})^2 \geq \varepsilon^2] = \mathbb{P}\Big[\big|\frac{1}{n^{(K+1)}} \sum_{i=1}^{n^{(K+1)}} Z_{jl,i}^{(K+1)}\big| \geq \varepsilon\Big] \leq 2\exp\left(-\frac{\varepsilon^2 n^{(K+1)}}{32}\right). \tag{115}$$

By the Courant-Fischer variational representation (Horn & Johnson, 2012),

$$\begin{aligned}
\Lambda_{\min}(\bar{Q}_{S(K+1)S(K+1)}^{(K+1)}) &= \min_{\|x\|_2=1} x^T \bar{Q}_{S(K+1)S(K+1)}^{(K+1)} x \\
&= \min_{\|x\|_2=1} \{x^T Q_{S(K+1)S(K+1)}^{(K+1)} x + x^T (\bar{Q}_{S(K+1)S(K+1)}^{(K+1)} - Q_{S(K+1)S(K+1)}^{(K+1)})x\} \\
&\leq y^T Q_{S(K+1)S(K+1)}^{(K+1)} y + y^T (\bar{Q}_{S(K+1)S(K+1)}^{(K+1)} - Q_{S(K+1)S(K+1)}^{(K+1)})y,
\end{aligned}$$

where $y \in \mathbb{R}^d$ is a unit-norm minimal eigenvector of $Q_{S(K+1)S(K+1)}^{(K+1)}$. Therefore, we have

$$\begin{aligned}
\Lambda_{\min}(Q_{S(K+1)S(K+1)}^{(K+1)}) &\geq \Lambda_{\min}(\bar{Q}_{S(K+1)S(K+1)}^{(K+1)}) - \left\|\bar{Q}_{S(K+1)S(K+1)}^{(K+1)} - Q_{S(K+1)S(K+1)}^{(K+1)}\right\|_2 \\
&\geq C_{\min}^{(K+1)} - \left\|\bar{Q}_{S(K+1)S(K+1)}^{(K+1)} - Q_{S(K+1)S(K+1)}^{(K+1)}\right\|_2.
\end{aligned}$$

Observe that

$$\left\|\bar{Q}_{S(K+1)S(K+1)}^{(K+1)} - Q_{S(K+1)S(K+1)}^{(K+1)}\right\|_2 \leq \left(\sum_{j=1}^{d} \sum_{l=1}^{d} (Z_{jl}^{(K+1)})^2\right)^{1/2}$$

Setting $\varepsilon^2 = \delta^2/d^2$ in (115) and applying the union bound over the $d^2$ index pairs $(j, l)$ then yields

$$\mathbb{P}\Big[\left\|\bar{Q}_{S(K+1)S(K+1)}^{(K+1)} - Q_{S(K+1)S(K+1)}^{(K+1)}\right\|_2 \geq \delta\Big] \leq 2\exp\left(-\frac{\delta^2 n^{(K+1)}}{32d^2} + 2\log(d)\right) \tag{116}$$

So, we have the first concentration inequality in Lemma F.2:

$$\mathbb{P}[\Lambda_{\min}(Q_{S(K+1)S(K+1)}^{(K+1)}) \leq C_{\min}^{(K+1)} - \delta] \leq 2\exp\left(-\frac{\delta^2 n^{(K+1)}}{32d^2} + 2\log(d)\right). \tag{117}$$

This proves the first part of the lemma. For the second concentration inequality about maximum eigenvalue of the sample covariance matrix, with the same reasoning from the Courant-Fischer variational representation

([Horn & Johnson, 2012](#)), we have,

$$\Lambda_{\max}(\mathbb{E}[X_S^{(K+1)}(X_S^{(K+1)})^T]) = \max_{\|v\|_2=1} v^T \mathbb{E}[X_S^{(K+1)}(X_S^{(K+1)})^T]v$$

$$= \max_{\|v\|_2=1} \{v^T(\frac{1}{n^{(K+1)}}\sum_{i=1}^{n^{(K+1)}} x_{i,S}^{(K+1)}(x_{i,S}^{(K+1)})^T)v$$

$$+ v^T(\mathbb{E}[X_S^{(K+1)}(X_S^{(K+1)})^T] - \frac{1}{n^{(K+1)}}\sum_{i=1}^{n^{(K+1)}} x_{i,S}^{(K+1)}(x_{i,S}^{(K+1)})^T)v\}$$

$$\geq u^T(\frac{1}{n^{(K+1)}}\sum_{i=1}^{n^{(K+1)}} x_{i,S}^{(K+1)}(x_{i,S}^{(K+1)})^T)u$$

$$+ u^T(\mathbb{E}[X_S^{(K+1)}(X_S^{(K+1)})^T] - \frac{1}{n^{(K+1)}}\sum_{i=1}^{n^{(K+1)}} x_{i,S}^{(K+1)}(x_{i,S}^{(K+1)})^T)u,$$

where $u \in \mathbb{R}^d$ is a unit-norm maximal eigenvector of $\frac{1}{n^{(K+1)}}\sum_{i=1}^{n^{(K+1)}} x_{i,S}^{(K+1)}(x_{i,S}^{(K+1)})^T$. Therefore, we have

$$\Lambda_{\max}(\frac{1}{n^{(K+1)}}\sum_{i=1}^{n^{(K+1)}} x_{i,S}^{(K+1)}(x_{i,S}^{(K+1)})^T)$$

$$\leq \Lambda_{\max}(\mathbb{E}[X_S^{(K+1)}(X_S^{(K+1)})^T]) + u^T(\frac{1}{n^{(K+1)}}\sum_{i=1}^{n^{(K+1)}} x_{i,S}^{(K+1)}(x_{i,S}^{(K+1)})^T - \mathbb{E}[X_S^{(K+1)}(X_S^{(K+1)})^T])u$$

$$\leq D_{\max}^{(K+1)} + \left\|\!\left\|\!\left\|(\frac{1}{n^{(K+1)}}\sum_{i=1}^{n^{(K+1)}} x_{i,S}^{(K+1)}(x_{i,S}^{(K+1)})^T - \mathbb{E}[X_S^{(K+1)}(X_S^{(K+1)})^T])\right\|\!\right\|\!\right\|_2$$

The difference matrix $\frac{1}{n^{(K+1)}}\sum_{i=1}^{n^{(K+1)}} x_{i,S}^{(K+1)}(x_{i,S}^{(K+1)})^T - \mathbb{E}[X_S^{(K+1)}(X_S^{(K+1)})^T]$ can be written as an i.i.d. sum of the form $Y_{jl}^{(K+1)} = \frac{1}{n^{(K+1)}}\sum_{i=1}^{n^{(K+1)}} Y_{jl,i}^{(K+1)}$, where each $Y_{jl,i}^{(K+1)}$ is zero-mean and bounded (in particular, $|Y_{jl,i}^{(K+1)}| \leq 4$). By the Azuma-Hoeffding's bound ([Hoeffding, 1994](#)), for any indices $j, l = 1, \ldots, d$ and for any $\varepsilon > 0$, we have

$$\mathbb{P}[(Y_{jl}^{(K+1)})^2 \geq \varepsilon^2] = \mathbb{P}[|\frac{1}{n^{(K+1)}}\sum_{i=1}^{n^{(K+1)}} Y_{jl,i}^{(K+1)}| \geq \varepsilon] \leq 2\exp\left(-\frac{\varepsilon^2 n^{(K+1)}}{32}\right). \tag{118}$$

Observe that

$$\left\|\!\left\|\!\left\|\frac{1}{n^{(K+1)}}\sum_{i=1}^{n^{(K+1)}} x_{i,S}^{(K+1)}(x_{i,S}^{(K+1)})^T - \mathbb{E}[X_S^{(K+1)}(X_S^{(K+1)})^T]\right\|\!\right\|\!\right\|_2 \leq \left(\sum_{j=1}^{d}\sum_{l=1}^{d}(Y_{jl}^{(K+1)})^2\right)^{1/2}.$$

Setting $\varepsilon^2 = \delta^2/d^2$ in (118) and applying the union bound over the $d^2$ index pairs $(j, l)$ then yields

$$\mathbb{P}[\left\|\!\left\|\!\left\|\frac{1}{n^{(K+1)}}\sum_{i=1}^{n^{(K+1)}} x_{i,S}^{(K+1)}(x_{i,S}^{(K+1)})^T - \mathbb{E}[X_S^{(K+1)}(X_S^{(K+1)})^T]\right\|\!\right\|\!\right\|_2 \geq \delta]$$

$$\leq 2\exp\left(-\frac{\delta^2 n^{(K+1)}}{32d^2} + 2\log(d)\right)$$

So we have the second part of the lemma

$$\mathbb{P}\Big[\Lambda_{\max}\Big[\frac{1}{n^{(K+1)}}\sum_{i=1}^{n^{(K+1)}} x_{i,S}^{(K+1)}(x_{i,S}^{(K+1)})^T\Big] \geq D_{\max}^{(K+1)} + \delta\Big] \leq 2\exp\left(-\frac{\delta^2 n^{(K+1)}}{32d^2} + 2\log(d)\right)$$

$\square$

### I.1.2 Proof of Lemma F.3

*Proof.* Decomposing the sample matrix as the sum $Q_{[S^{(K+1)}]^c S^{(K+1)}}^{(K+1),S}(Q_{S^{(K+1)}S^{(K+1)}}^{(K+1)})^{-1} = T_1^{(K+1)} + T_2^{(K+1)} + T_3^{(K+1)} + T_4^{(K+1)}$, where we define

$$T_1^{(K+1)} := \bar{Q}_{[S^{(K+1)}]^c S^{(K+1)}}^{(K+1),S}[(Q_{S^{(K+1)}S^{(K+1)}}^{(K+1)})^{-1} - (\bar{Q}_{S^{(K+1)}S^{(K+1)}}^{(K+1)})^{-1}], \tag{119a}$$

$$T_2^{(K+1)} := [Q_{[S^{(K+1)}]^c S^{(K+1)}}^{(K+1),S} - \bar{Q}_{[S^{(K+1)}]^c S^{(K+1)}}^{(K+1),S}](\bar{Q}_{S^{(K+1)}S^{(K+1)}}^{(K+1)})^{-1}, \tag{119b}$$

$$T_3^{(K+1)} := [Q_{[S^{(K+1)}]^c S^{(K+1)}}^{(K+1),S} - \bar{Q}_{[S^{(K+1)}]^c S^{(K+1)}}^{(K+1),S}][(Q_{S^{(K+1)}S^{(K+1)}}^{(K+1)})^{-1} - (\bar{Q}_{S^{(K+1)}S^{(K+1)}}^{(K+1)})^{-1}], \tag{119c}$$

$$T_4^{(K+1)} := \bar{Q}_{[S^{(K+1)}]^c S^{(K+1)}}^{(K+1),S}(\bar{Q}_{S^{(K+1)}S^{(K+1)}}^{(K+1)})^{-1}. \tag{119d}$$

The fourth term is controlled by the incoherence assumption (A2)

$$\left\|\left|T_4^{(K+1)}\right|\right\|_\infty = \left\|\left|\bar{Q}_{[S^{(K+1)}]^c S^{(K+1)}}^{(K+1),S}(\bar{Q}_{S^{(K+1)}S^{(K+1)}}^{(K+1)})^{-1}\right|\right\|_\infty \leq 1 - \alpha^{(K+1)}.$$

If we can show that $\left\|\left|T_i^{(K+1)}\right|\right\|_\infty \leq \frac{\alpha^{(K+1)}}{6}$ for the remaining indices $i = 1, 2, 3$, then by our four term decomposition and the triangle inequality, the sample version satisfies the desired bound (68). For the remaining three terms, the following lemma is useful in the proof:

**Lemma I.1.** *For any $\delta > 0$, and constants $B, B_1, B_2$, the following bounds hold,*

$$\mathbb{P}\left[\left\|\left|Q_{[S^{(K+1)}]^c S^{(K+1)}}^{(K+1),S} - \bar{Q}_{[S^{(K+1)}]^c S^{(K+1)}}^{(K+1),S}\right|\right\|_\infty \geq \delta\right] \leq 2\exp\left(-B\frac{\varepsilon^2 n^{(K+1)}}{d^2} + 2\log(d)\right), \tag{120a}$$

$$\mathbb{P}\left[\left\|\left|Q_{S^{(K+1)}S^{(K+1)}}^{(K+1)} - \bar{Q}_{S^{(K+1)}S^{(K+1)}}^{(K+1)}\right|\right\|_\infty \geq \delta\right] \leq 2\exp\left(-B\frac{\varepsilon^2 n^{(K+1)}}{d^2} + 2\log(d)\right), \tag{120b}$$

$$\mathbb{P}\left[\left\|\left|(Q_{S^{(K+1)}S^{(K+1)}}^{(K+1)})^{-1} - (\bar{Q}_{S^{(K+1)}S^{(K+1)}}^{(K+1)})^{-1}\right|\right\|_\infty \geq \delta\right] \leq 4\exp\left(-B_1\frac{n^{(K+1)}\delta^2}{d^3} + B_2\log(d)\right). \tag{120c}$$

See Section J.2 for the proof of these claims.

**Control of the first term.** Turning to the first term, we re-factorize it as

$$T_1^{(K+1)} = \bar{Q}_{[S^{(K+1)}]^c S^{(K+1)}}^{(K+1),S}(\bar{Q}_{S^{(K+1)}S^{(K+1)}}^{(K+1)})^{-1}[\bar{Q}_{S^{(K+1)}S^{(K+1)}}^{(K+1)} - Q_{S^{(K+1)}S^{(K+1)}}^{(K+1)}](Q_{S^{(K+1)}S^{(K+1)}}^{(K+1)})^{-1}.$$

Then, we can upper bound $\left\|\left|T_1^{(K+1)}\right|\right\|_\infty$ by

$$\left\|\left|\bar{Q}_{[S^{(K+1)}]^c S^{(K+1)}}^{(K+1),S}(\bar{Q}_{S^{(K+1)}S^{(K+1)}}^{(K+1)})^{-1}\right|\right\|_\infty\left\|\left|\bar{Q}_{S^{(K+1)}S^{(K+1)}}^{(K+1)} - Q_{S^{(K+1)}S^{(K+1)}}^{(K+1)}\right|\right\|_\infty\left\|\left|(Q_{S^{(K+1)}S^{(K+1)}}^{(K+1)})^{-1}\right|\right\|_\infty$$

$$\leq (1-\alpha)\left\|\left|\bar{Q}_{S^{(K+1)}S^{(K+1)}}^{(K+1)} - Q_{S^{(K+1)}S^{(K+1)}}^{(K+1)}\right|\right\|_\infty\{\sqrt{d}\left\|\left|(Q_{S^{(K+1)}S^{(K+1)}}^{(K+1)})^{-1}\right|\right\|_2\},$$

where we have used the incoherence assumption in Assumption 4.6. Using the bound (67) in Lemma F.2 with $\delta = C_{\min}/2$, we have $\left\|\left|(Q_{S^{(K+1)}S^{(K+1)}}^{(K+1)})^{-1}\right|\right\|_2 = [\Lambda_{\min}(Q_{S^{(K+1)}S^{(K+1)}}^{(K+1)})]^{-1} \leq \frac{2}{C_{\min}}$ with probability greater

than $1 - 2\exp(-Bn^{(K+1)}/d^2 + 2\log(d))$. Next, applying the bound (120b) with $\delta = c/\sqrt{d}$, we conclude that with probability greater than $1 - 2\exp(-Bn^{(K+1)}c^2/d^3 + 2\log(d))$, we have

$$\left\|\left\|\bar{Q}^{(K+1)}_{S(K+1)S(K+1)} - Q^{(K+1)}_{S(K+1)S(K+1)}\right\|\right\|_\infty \le c/\sqrt{d}.$$

By choosing the constant $c > 0$ sufficiently small, we are guaranteed that

$$\mathbb{P}[\left\|\left\|T_1^{(K+1)}\right\|\right\|_\infty \ge \alpha^{(K+1)}/6] \le 2\exp\left(-B\frac{n^{(K+1)}c^2}{d^3} + \log(d)\right).$$

**Control of the second term.** To bound $T_2^{(K+1)}$, we first write

$$\left\|\left\|T_2^{(K+1)}\right\|\right\|_\infty \le \sqrt{d}\left\|\left\|(\bar{Q}^{(K+1)}_{S(K+1)S(K+1)})^{-1}\right\|\right\|_2 \left\|\left\|Q^{(K+1),S}_{[S(K+1)]^c S(K+1)} - \bar{Q}^{(K+1),S}_{[S(K+1)]^c S(K+1)}\right\|\right\|_\infty$$

$$\le \frac{\sqrt{d}}{C_{\min}}\left\|\left\|Q^{(K+1),S}_{[S(K+1)]^c S(K+1)} - \bar{Q}^{(K+1),S}_{[S(K+1)]^c S(K+1)}\right\|\right\|_\infty.$$

Then we apply the bound (120a) with $\delta = \frac{\alpha^{(K+1)}}{6}\frac{C_{\min}}{\sqrt{d}}$ to conclude that

$$\mathbb{P}[\left\|\left\|T_2^{(K+1)}\right\|\right\|_\infty \ge \alpha^{(K+1)}/6] \le 2\exp\left(-B\frac{n^{(K+1)}}{d^3} + \log(d)\right).$$

**Control of the third term.** We set $\delta = \sqrt{\alpha^{(K+1)}/6}$ in the bounds (120a) and (120c) to conclude that

$$\mathbb{P}[\left\|\left\|T_3^{(K+1)}\right\|\right\|_\infty \ge \alpha^{(K+1)}/6] \le 4\exp\left(-B\frac{n^{(K+1)}}{d^3} + \log(d)\right).$$

Putting together, we conclude that

$$\mathbb{P}[\left\|\left\|Q^{(K+1),S}_{[S(K+1)]^c S(K+1)}(Q^{(K+1)}_{S(K+1)S(K+1)})^{-1}\right\|\right\|_\infty \ge 1 - \alpha^{(K+1)}/2] = O\left(\exp\left(-B\frac{n^{(K+1)}}{d^3} + \log(d)\right)\right)$$

$\square$

## I.2 Proof of Lemmas for Proposition F.4

### I.2.1 Proof of Lemma F.5

*Proof.* Each entry of $W^{(K+1)}$, denoted by $W_u^{(K+1)}$, for $1 \le u \le |S(r)| \le d$, can be expressed as a sum of independent random variables $Z_{i,u}^{(K+1)}$:

$$W_u^{(K+1)} = \frac{1}{n^{(K+1)}}\sum_{i=1}^{n^{(K+1)}} Z_{i,u}^{(K+1)},$$

where

$$Z_{i,u}^{(K+1)} = x_{i,u}^{(K+1)}\{x_{i,r}^{(K+1)} - \frac{\exp(\sum_{t\in S\backslash r}\bar{\theta}_{rt}^{(K+1)}x_{i,t}^{(K+1)}) - \exp(-\sum_{t\in S\backslash r}\bar{\theta}_{rt}^{(K+1)}x_{i,t}^{(K+1)})}{\exp(\sum_{t\in S\backslash r}\bar{\theta}_{rt}^{(K+1)}x_{i,t}^{(K+1)}) + \exp(-\sum_{t\in S\backslash r}\bar{\theta}_{rt}^{(K+1)}x_{i,t}^{(K+1)})}\}$$

$$= x_{i,u}^{(K+1)}\{x_{i,r}^{(K+1)} - \mathbb{P}_{\bar{\theta}_S^{(K+1)}}[X_r^{(K+1)} = 1|x_{i,S}^{(K+1)}] + \mathbb{P}_{\bar{\theta}_S^{(K+1)}}[X_r^{(K+1)} = -1|x_{i,S}^{(K+1)}]\}.$$

Notice that the conditional expectation given the values of $\Delta^{(K+1)}$ has mean zero:

$$\mathbb{E}[Z_{i,u}^{(K+1)}|\Delta^{(K+1)}] = 0.$$

Then by law of total expectation ([Weiss et al., 2005](#)) we have

$$\mathbb{E}[Z_{i,u}^{(K+1)}] = \mathbb{E}[\mathbb{E}[Z_{i,u}^{(K+1)}|\Delta^{(K+1)}]] = \mathbb{E}[0] = 0. \tag{121}$$

(See the same logic in the proof in Section [H.2.1](#)). Also, since all the samples are either $-1$ or $+1$, we have $|Z_{i,u}^{(K+1)}| \leq 2$. Then by Azuma-Hoeffding's inequality ([Hoeffding, 1994](#)), we have, for any $\delta > 0$,

$$\mathbb{P}[|W_u^{(K+1)}| > \delta] \leq 2\exp(-\frac{n^{(K+1)}\delta^2}{8}).$$

Setting $\delta = \frac{\alpha^{(K+1)}\lambda^{(K+1)}}{4(2-\alpha^{(K+1)})}$, we obtain

$$\mathbb{P}[\frac{2-\alpha^{(K+1)}}{\lambda^{(K+1)}}|W_u^{(K+1)}| > \frac{\alpha^{(K+1)}}{4}] \leq 2\exp\left(-\frac{(\alpha^{(K+1)})^2(\lambda^{(K+1)})^2}{128(2-\alpha^{(K+1)})^2}n^{(K+1)}\right)$$

Applying a union bound over the indices $u$ of $W^{(K+1)}$ yields

$$\mathbb{P}[\frac{2-\alpha^{(K+1)}}{\lambda^{(K+1)}}\|W^{(K+1)}\|_\infty > \frac{\alpha^{(K+1)}}{4}] \leq 2\exp\left(-\frac{(\alpha^{(K+1)})^2(\lambda^{(K+1)})^2}{128(2-\alpha^{(K+1)})^2}n^{(K+1)} + \log d\right),$$

which converges to zero at rate $\exp(-c(\lambda^{(K+1)})^2 n^{(K+1)})$ as long as $\lambda^{(K+1)} \geq \frac{16(2-\alpha^{(K+1)})}{\alpha^{(K+1)}}\sqrt{\frac{\log d}{n^{(K+1)}}}$  ☐

### I.2.2 Proof of Lemma [F.6](#)

*Proof.* Similar to the proof for Lemma [D.6](#). We first show that the remainder term $R^{(K+1)}$ satisfies the bound $\|R^{(K+1)}\|_\infty \leq D_{\max}^{(K+1)}\|\hat{\theta}_{S^{(K+1)}}^{(K+1)} - \bar{\theta}_{S^{(K+1)}}^{(K+1)}\|_2^2$. Then the result of Lemma [F.7](#), namely $\|\hat{\theta}_{S^{(K+1)}}^{(K+1)} - \bar{\theta}_{S^{(K+1)}}^{(K+1)}\|_2 \leq \frac{5}{C_{\min}^{(K+1)}}\sqrt{d}\lambda^{(K+1)}$, can be used to conclude that

$$\frac{\|R^{(K+1)}\|_\infty}{\lambda^{(K+1)}} \leq \frac{25D_{\max}^{(K+1)}}{C_{\min}^{(K+1)2}}\lambda^{(K+1)}d,$$

as claimed in Lemma [D.6](#). Focusing on element $R_j^{(K+1)}$ for some index $j \in \{1, \ldots, |S_r|\}$, we have

$$R_j^{(K+1)}$$
$$= -[\nabla^2\ell^{(K+1)}(\theta_S^{(K+1)j}; \{\mathfrak{X}_{1,S}^{n^{(K+1)}}\}^{(K+1)}) - \nabla^2\ell^{(K+1)}(\bar{\theta}_S^{(K+1)}; \{\mathfrak{X}_{1,S}^{n^{(K+1)}}\}^{(K+1)})]_j^T(\hat{\theta}_S^{(K+1)} - \bar{\theta}_S^{(K+1)})$$
$$= \frac{1}{n^{(K+1)}}\sum_{i=1}^{n^{(K+1)}}[\eta(x_i^{(K+1)}; \theta_S^{(K+1)(j)}) - \eta(x_i^{(K+1)}; \bar{\theta}_S^{(K+1)}))](\hat{\theta}_S^{(K+1)} - \bar{\theta}_S^{(K+1)})$$

for some point $\theta_S^{(K+1)(j)} = \mu_j\hat{\theta}_S^{(K+1)} + (1-\mu_j)\bar{\theta}_S^{(K+1)}$ with $\mu_j \in [0,1]$. Setting $g(t) = \frac{4e^{2t}}{(e^{2t}+1)^2}$ by noting that that $\eta(\theta_S, x) = g(x_r\sum_{t\in S\backslash r}\theta_{rt}x_t)$. By the chain rule and another application of the mean value theorem, we write

$$R_j^{(K+1)} = \frac{1}{n^{(K+1)}}\sum_{i=1}^{n^{(K+1)}}\{g'((\theta_S'^{(K+1)(j)})^T x_{i,S}^{(K+1)})x_{i,j}^{(K+1)}\}\{[\theta_S^{(K+1)(j)} - \bar{\theta}_S^{(K+1)}]^T$$
$$x_{i,j}^{(K+1)}(x_{i,S}^{(K+1)})^T[\hat{\theta}_S^{(K+1)} - \bar{\theta}_S^{(K+1)}]\},$$

where $\theta_S'^{(K+1)(j)}$ is another point on the line joining $\hat{\theta}_S^{(K+1)}$ and $\bar{\theta}_S^{(K+1)}$.

Setting $a_i^{(K+1)} := \{g'((\theta_S'^{(j)})^T x_{i,S}^{(K+1)}) x_{i,j}^{(K+1)}\}$ and $b_i^{(K+1)} := \{[\theta_S^{(j)} - \bar{\theta}_S]^T x_{i,j}^{(K+1)} (x_{i,S}^{(K+1)})^T [\hat{\theta}_S - \bar{\theta}_S]\}$,

$$|R_j^{(K+1)}| = \frac{1}{n^{(K+1)}} \left| \sum_{i=1}^{n^{(K+1)}} a_i^{(K+1)} b_i^{(K+1)} \right| \leq \frac{1}{n^{(K+1)}} \|a^{(K+1)}\|_\infty \|b^{(K+1)}\|_1.$$

We have $\|a^{(K+1)}\|_\infty \leq 1$, and

$$\begin{aligned}
\frac{1}{n^{(K+1)}} \|b^{(K+1)}\|_1 &= \mu_j [\hat{\theta}_S^{(K+1)} - \bar{\theta}_S^{(K+1)}]^T \left\{ \frac{1}{n^{(K+1)}} \sum_{i=1}^{n^{(K+1)}} x_{i,S}^{(K+1)} (x_{i,S}^{(K+1)})^T \right\} [\hat{\theta}_S^{(K+1)} - \bar{\theta}_S^{(K+1)}] \\
&= \mu_j [\hat{\theta}_S^{(K+1)} - \bar{\theta}_S^{(K+1)}]^T \left\{ \frac{1}{n} \sum_{i=1}^{n} x_{i,S}^{(K+1)} (x_{i,S}^{(K+1)})^T \right\} [\hat{\theta}_S^{(K+1)} - \bar{\theta}_S^{(K+1)}] \\
&\leq D_{\max}^{(K+1)} \|\hat{\theta}_S^{(K+1)} - \bar{\theta}_S^{(K+1)}\|_2^2 \\
&= D_{\max}^{(K+1)} \|\hat{\theta}_{S(K+1)}^{(K+1)} - \bar{\theta}_{S(K+1)}^{(K+1)}\|_2^2,
\end{aligned}$$

where the last line uses the fact that $\hat{\theta}_{[S(K+1)]^c}^{(K+1)} = \bar{\theta}_{[S(K+1)]^c}^{(K+1)} = 0$ Therefore, we have

$$\|R^{(K+1)}\|_\infty \leq D_{\max}^{(K+1)} \|\hat{\theta}_{S(K+1)}^{(K+1)} - \bar{\theta}_{S(K+1)}^{(K+1)}\|_2^2$$

$\square$

### I.2.3 Proof of Lemma E.7

*Proof.* As in the proof for Lemma D.7, following the method of proof in Ravikumar et al. (2010) which was also previously used in another context (Rothman et al., 2008), we define the function $G^{(K+1)} : \mathbb{R}^d \to \mathbb{R}$ by

$$\begin{aligned}
G^{(K+1)}(u_{S(K+1)}) := \ell^{(K+1)}(\bar{\theta}_{S(K+1)}^{(K+1)} + u_{S(K+1)}) \\
- \ell^{(K+1)}(\bar{\theta}_{S(K+1)}^{(K+1)}) + \lambda^{(K+1)}(\|\bar{\theta}_{S(K+1)}^{(K+1)} + u_{S(K+1)}\|_1 - \|\bar{\theta}_{S(K+1)}^{(K+1)}\|_1). \quad (122)
\end{aligned}$$

It can be seen that $\hat{u}_{S(K+1)} = \hat{\theta}_{S(K+1)}^{(K+1)} - \bar{\theta}_{S(K+1)}^{(K+1)}$ minimizes $G^{(K+1)}$. Moreover, $G^{(K+1)}(0) = 0$ by construction; therefore, we must have $G^{(K+1)}(\hat{u}_{S(K+1)}) \leq 0$. Note also that $G^{(K+1)}$ is convex. Suppose that we show for some radius $B > 0$, and for $u \in \mathbb{R}^d$ with $\|u\|_2 = B$, we have $G^{(K+1)}(u) > 0$. We then claim that $\|\hat{u}\|_2 \leq B$. Indeed, if $\hat{u}$ lay outside the ball of radius $B$, then the convex combination $t\hat{u} + (1-t)(0)$ would lie on the boundary of the ball, for an appropriately chosen $t \in (0, 1)$. By convexity,

$$G^{(K+1)}(t\hat{u} + (1-t)(0)) \leq t G^{(K+1)}(\hat{u}) + (1-t) G^{(K+1)}(0) \leq 0,$$

contradicting the assumed strict positivity of $G^{(K+1)}$ on the boundary. It thus suffices to establish strict positivity of $G^{(K+1)}$ on the boundary of the ball with radius $B = M\lambda^{(K+1)}\sqrt{d}$, where $M > 0$ is a parameter to be chosen later in the proof. Let $u \in \mathbb{R}^d$ be an arbitrary vector with $\|u\|_2 = B$. Recalling the notation $W^{(K+1)} := -\nabla \ell^{(K+1)}(\bar{\theta}_S^{(K+1)}; \{\mathfrak{X}_{1,S}^{n^{(K+1)}}\}^{(K+1)})$, by a Taylor series expansion of the log likelihood component of $G^{(K+1)}$, we have

$$\begin{aligned}
G(u) = -(W_{S(K+1)}^{(K+1)})^T u + u^T [\nabla^2 \ell(\bar{\theta}_{S(K+1)}^{(K+1)} + \alpha u_{S(K+1)}; \{\mathfrak{X}_{1,S}^{n^{(K+1)}}\}^{(K+1)})] u \\
+ \lambda^{(K+1)}(\|\bar{\theta}_{S(K+1)}^{(K+1)} + u_{S(K+1)}\|_1 - \|\bar{\theta}_{S(K+1)}^{(K+1)}\|_1)
\end{aligned}$$

for some $\alpha \in [0, 1]$. For the first term, we have the bound

$$|(W_{S(K+1)}^{(K+1)})^T u| \leq \|W_{S(K+1)}^{(K+1)}\|_\infty \|u\|_1 \leq \|W_{S(K+1)}^{(K+1)}\|_\infty \sqrt{d}\|u\|_2 \leq (\lambda^{(K+1)}\sqrt{d})^2 \frac{M}{4},$$

since $\|W_{S^{(K+1)}}^{(K+1)}\|_\infty \leq \frac{\lambda^{(K+1)}}{4}$ by assumption. For the last term, applying triangle inequality yields

$$\lambda^{(K+1)}(\|\bar{\theta}_{S^{(K+1)}}^{(K+1)} + u_{S^{(K+1)}}\|_1 - \|\bar{\theta}_{S^{(K+1)}}^{(K+1)}\|_1) \geq -\lambda^{(K+1)}\|u_{S^{(K+1)}}\|_1.$$

Since $\|u_{S^{(K+1)}}\|_1 \leq \sqrt{d}\|u_{S^{(K+1)}}\|_2$, we have

$$\lambda^{(K+1)}(\|\bar{\theta}_{S^{(K+1)}}^{(K+1)} + u_{S^{(K+1)}}\|_1 - \|\bar{\theta}_{S^{(K+1)}}^{(K+1)}\|_1) \geq -\lambda^{(K+1)}\sqrt{d}\|u_{S^{(K+1)}}\|_2 = -M(\sqrt{d}\lambda^{(K+1)})^2.$$

Finally, turning to the middle Hessian term, we have

$$q^* := \Lambda_{\min}(\nabla^2\ell(\bar{\theta}_{S^{(K+1)}}^{(K+1)} + \alpha^{(K+1)}u_{S^{(K+1)}}; \{\mathfrak{X}_{1,S}^{n^{(K+1)}}\}^{(K+1)}))$$

$$\geq \min_{\alpha^{(K+1)}\in[0,1]} \Lambda_{\min}(\nabla^2\ell(\bar{\theta}_{S^{(K+1)}}^{(K+1)} + \alpha^{(K+1)}u_{S^{(K+1)}}; \{\mathfrak{X}_{1,S}^{n^{(K+1)}}\}^{(K+1)}))$$

$$= \min_{\alpha^{(K+1)}\in[0,1]} \Lambda_{\min}\left[\frac{1}{n^{(K+1)}} \sum_{i=1}^{n^{(K+1)}} \eta(x_i^{(K+1)}; \bar{\theta}_{S^{(K+1)}}^{(K+1)} + \alpha^{(K+1)}u_{S^{(K+1)}})x_{i,S^{(K+1)}}^{(K+1)}(x_{i,S^{(K+1)}}^{(K+1)})^T\right].$$

By a Taylor series expansion of $\eta(x_i^{(K+1)}; \cdot)$, we have, for some $\alpha_0 \in [0, \alpha^{(K+1)}]$,

$$q^* \geq \min_{\alpha^{(K+1)}\in[0,1]} \Lambda_{\min}\left\{\frac{1}{n^{(K+1)}} \sum_{i=1}^{n^{(K+1)}} \left[\eta(x_i^{(K+1)}; \bar{\theta}_{S^{(K+1)}}^{(K+1)})x_{i,S^{(K+1)}}^{(K+1)}(x_{i,S^{(K+1)}}^{(K+1)})^T\right]\right\}$$

$$+ \alpha^{(K+1)}g'\left(x_{i,r}^{(K+1)} \sum_{t\in S^{(K+1)}\backslash r}(\bar{\theta}_{rt}^{(K+1)} + \alpha_0 u_{rt})x_{i,t}^{(K+1)}\right)x_{i,r}^{(K+1)}(u_{S^{(K+1)}}^T x_{i,S^{(K+1)}}^{(K+1)})x_{i,S^{(K+1)}}^{(K+1)}(x_{i,S^{(K+1)}}^{(K+1)})^T$$

$$\geq \Lambda_{\min}\left[\frac{1}{n^{(K+1)}} \sum_{i=1}^{n^{(K+1)}} \eta(x_i^{(K+1)}; \bar{\theta}_{S^{(K+1)}})x_{i,S^{(K+1)}}^{(K+1)}(x_{i,S^{(K+1)}}^{(K+1)})^T\right]$$

$$+ \min_{\alpha^{(K+1)}\in[0,1]} \alpha^{(K+1)}\Lambda_{\min}\left[\frac{1}{n^{(K+1)}} \sum_{i=1}^{n^{(K+1)}} g'\left(x_{i,r}^{(K+1)}(\bar{\theta}_{S^{(K+1)}}^{(K+1)} + \alpha_0 u_{S^{(K+1)}})^T x_{i,S^{(K+1)}}^{(K+1)}\right)\right.$$

$$\left. x_{i,r}^{(K+1)}(u_{S^{(K+1)}}^T x_{i,S^{(K+1)}}^{(K+1)})x_{i,S^{(K+1)}}^{(K+1)}(x_{i,S^{(K+1)}}^{(K+1)})^T\right]$$

$$\geq \Lambda_{\min}(Q_{S^{(K+1)}S^{(K+1)}}^{(K+1)}) - \max_{\alpha^{(K+1)}\in[0,1]}$$

$$\left\|\frac{1}{n^{(K+1)}} \sum_{i=1}^{n^{(K+1)}} g'(x_{i,r}^{(K+1)}(\bar{\theta}_{S^{(K+1)}}^{(K+1)} + \alpha_0 u_{S^{(K+1)}})^T x_{i,S^{(K+1)}}^{(K+1)})(u_{S^{(K+1)}}^T x_{i,S^{(K+1)}}^{(K+1)})x_{i,S^{(K+1)}}^{(K+1)}(x_{i,S^{(K+1)}}^{(K+1)})^T\right\|_2$$

$$\geq C_{\min} - \max_{\alpha^{(K+1)}\in[0,1]}$$

$$\left\|\frac{1}{n^{(K+1)}} \sum_{i=1}^{n^{(K+1)}} g'(x_{i,r}^{(K+1)}(\bar{\theta}_{S^{(K+1)}}^{(K+1)} + \alpha_0 u_{S^{(K+1)}})^T x_{i,S^{(K+1)}}^{(K+1)})(\langle u_{S^{(K+1)}}, x_{i,S^{(K+1)}}^{(K+1)}\rangle)x_{i,S^{(K+1)}}^{(K+1)}(x_{i,S^{(K+1)}}^{(K+1)})^T\right\|_2$$

It remains to control the spectral norm of the matrix , denoted as $A(\alpha^{(K+1)})$ here, for $\alpha^{(K+1)} \in [0, 1]$. For any fixed $\alpha^{(K+1)} \in [0, 1]$, and $y \in \mathbb{R}$ with $\|y\|_2 = 1$, we have

$$\langle y, A(\alpha^{(K+1)})y\rangle = \frac{1}{n^{(K+1)}} \sum_{i=1}^{n^{(K+1)}} g'\left(\bar{\theta}_{S^{(K+1)}}^{(K+1)} + \alpha_0 u_{S^{(K+1)}}\right)[\langle u_{S^{(K+1)}}, x_{i,S^{(K+1)}}^{(K+1)}\rangle][\langle x_{i,S^{(K+1)}}^{(K+1)}, y\rangle]^2$$

$$\leq \frac{1}{n^{(K+1)}} \sum_{i=1}^{n^{(K+1)}} \left|g'\left(\bar{\theta}_{S^{(K+1)}}^{(K+1)} + \alpha_0 u_{S^{(K+1)}}\right)\right| |\langle u_{S^{(K+1)}}, x_{i,S^{(K+1)}}^{(K+1)}\rangle|[\langle x_{i,S^{(K+1)}}^{(K+1)}, y\rangle]^2.$$

Note that $\left| g'\left( \bar{\theta}_{S^{(K+1)}}^{(K+1)} + \alpha_0 u_{S^{(K+1)}} \right) \right| \leq 1$, and

$$|\langle u_{S^{(K+1)}}, x_{i,S^{(K+1)}}^{(K+1)} \rangle| \leq \|u_{S^{(K+1)}}\|_1 \leq \sqrt{d}\|u_{S^{(K+1)}}\|_2 = M\lambda^{(K+1)}d.$$

Moreover, we have

$$\frac{1}{n^{(K+1)}}\sum_{i=1}^{n^{(K+1)}} \left( \langle x_{i,S^{(K+1)}}^{(K+1)}, y \rangle \right)^2 \leq \left\| \left\| \frac{1}{n^{(K+1)}}\sum_{i=1}^{n^{(K+1)}} x_{i,S^{(K+1)}}^{(K+1)}(x_{i,S^{(K+1)}}^{(K+1)})^T \right\| \right\|_2 \leq D_{\max}^{(K+1)}$$

by assumption. We then obtain

$$\max_{\alpha^{(K+1)} \in [0,1]} \left\| \left\| A(\alpha^{(K+1)}) \right\| \right\|_2 \leq D_{\max}^{(K+1)}M\lambda^{(K+1)}d \leq C_{\min}^{(K+1)}/2,$$

assuming that $\lambda^{(K+1)} \leq \frac{C_{\min}^{(K+1)}}{2MD_{\max}^{(K+1)}d}$.

Under this condition, we have shown that

$$q^* := \Lambda_{\min}(\nabla^2 \ell(\bar{\theta}_{S^{(K+1)}}^{(K+1)} + \alpha^{(K+1)} u_{S^{(K+1)}})) \geq C_{\min}^{(K+1)}/2.$$

Finally, combining the three terms in $G^{(K+1)}(u)$, we conclude that

$$G^{(K+1)}(u_{S^{(K+1)}}) \geq (\lambda^{(K+1)}\sqrt{d})^2 \left\{ -\frac{1}{4}M + \frac{C_{\min}^{(K+1)}}{2}M^2 - M \right\},$$

which is strictly positive for $M = 5/C_{\min}^{(K+1)}$. So as long as

$$\lambda^{(K+1)} \leq \frac{C_{\min}^{(K+1)}}{2MD_{\max}^{(K+1)}d} = \frac{(C_{\min}^{(K+1)})^2}{10D_{\max}^{(K+1)}d},$$

we are guaranteed that

$$\|\hat{u}_{S^{(K+1)}}\|_2 \leq M\lambda^{(K+1)}\sqrt{d} = \frac{5}{C^{(K+1)}}\lambda^{(K+1)}\sqrt{d}.$$

$\square$

# J   Proof of Lemmas Used in Proving Other Lemmas

## J.1   Proof of Lemma H.1

*Proof.* By the definition of the $\ell_\infty$-matrix norm, and using $Z_{jl}$ defined in Section H.1.1 we have

$$\mathbb{P}[\|\|Q_{S^cS}^N - \bar{Q}_{S^cS}\|\|_\infty \geq \delta] = \mathbb{P}\Big[\max_{j \in S^c}\sum_{l \in S}|Z_{jl}| \geq \delta\Big]$$

$$\leq p\mathbb{P}\Big[\sum_{l \in S}|Z_{jl}| \geq \delta\Big],$$

where the final inequality uses a union bound and the fact that $|S^c| \leq p$.

$$\mathbb{P}\Big[\sum_{k \in S}|Z_{jl}| \geq \delta\Big] \leq \mathbb{P}[\exists k \in S||Z_{jl}| \geq \delta/d]$$

$$\leq d\mathbb{P}[|Z_{jl}| \geq \delta/d].$$

We then obtain (92a) by setting $\varepsilon = \delta/d$ in the Hoeffding bound (87):

$$\mathbb{P}[\||Q^N_{S^cS} - \bar{Q}_{S^cS}\||_\infty \geq \delta] \leq pd\mathbb{P}[|Z_{jl}| \geq \delta/d]$$
$$\leq 2\exp\left(-\frac{\varepsilon^2 nK}{32d^2} + \log(d) + \log(p)\right)$$

Analogously, for (92b), we have

$$\mathbb{P}[\||Q^N_{SS} - \bar{Q}_{SS}\||_\infty \geq \delta] = \mathbb{P}\big[\max_{j \in S} \sum_{k \in S} |Z_{jl}| \geq \delta\big]$$
$$\leq d\mathbb{P}\big[\sum_{l \in S} |Z_{jl}| \geq \delta\big]$$
$$\leq d\mathbb{P}[\exists l \in S | |Z_{jl}| \geq \delta/d]$$
$$\leq d^2\mathbb{P}[|Z_{jl}| \geq \delta/d]$$
$$\leq 2\exp\left(-\frac{\varepsilon^2 nK}{32d^2} + 2\log(d)\right).$$

To prove (92c), we can write

$$\||(Q^N_{SS})^{-1} - (\bar{Q}_{SS})^{-1}\||_\infty = \||(\bar{Q}_{SS})^{-1}[\bar{Q}_{SS} - Q^N_{SS}](Q^N_{SS})^{-1}\||_\infty$$
$$\leq \sqrt{d}\||(\bar{Q}_{SS})^{-1}[\bar{Q}_{SS} - Q^N_{SS}](Q^N_{SS})^{-1}\||_2$$
$$\leq \sqrt{d}\||(\bar{Q}_{SS})^{-1}\||_2\||\bar{Q}_{SS} - Q^N_{SS}\||_2\||(Q^N_{SS})^{-1}\||_2$$
$$\leq \frac{\sqrt{d}}{C_{\min}}\||\bar{Q}_{SS} - Q^N_{SS}\||_2\||(Q^N_{SS})^{-1}\||_2$$

Using the bound (88) in the proof of Lemma D.2, we get

$$\mathbb{P}[\||(Q^N_{SS})^{-1}\||_2 \geq \frac{2}{C_{\min}}] \leq 2\exp\left(-\frac{\delta^2 nK}{32d^2} + 2\log(d)\right),$$

and

$$\mathbb{P}[\||Q^N_{SS} - \bar{Q}_{SS}\||_2 \geq \delta/\sqrt{d}] \leq 2\exp\left(-\frac{\delta^2 nK}{32d^3} + 2\log(d)\right).$$

So finally we have

$$\mathbb{P}\left(\||(Q^N_{SS})^{-1} - (\bar{Q}_{SS})^{-1}\||_\infty \geq \delta\right) \leq 4\exp\left(-B_1\frac{nK\delta^2}{d^3} + B_2\log(d)\right),$$

where $B_1, B_2$ are some positive constants. □

## J.2 Proof of Lemma I.1

*Proof.* By the definition of the $\ell_\infty$-matrix norm, and using the $Z^{(K+1)}_{jl}$ defined in Section I.1.1, we have

$$\mathbb{P}[\||Q^{(K+1),S}_{[S^{(K+1)}]^cS^{(K+1)}} - \bar{Q}^{(K+1),S}_{[S^{(K+1)}]^cS^{(K+1)}}\||_\infty \geq \delta] = \mathbb{P}\big[\max_{j \in ([S^{(K+1)}]^c \cap S)} \sum_{k \in S^{(K+1)}} |Z^{(K+1)}_{jl}| \geq \delta\big]$$
$$\leq d\mathbb{P}\big[\sum_{l \in S^{(K+1)}} |Z^{(K+1)}_{jl}| \geq \delta\big],$$

where the final inequality uses a union bound and the fact that $|([S^{(K+1)}]^c \cap S)| \leq d$.

$$\mathbb{P}\big[\sum_{l \in S^{(K+1)}} |Z_{jl}^{(K+1)}| \geq \delta\big] \leq \mathbb{P}[\exists k \in S^{(K+1)}||Z_{jl}^{(K+1)}| \geq \delta/d]$$

$$\leq \mathbb{P}[\exists k \in |S^{(K+1)}|||Z_{jl}^{(K+1)}| \geq \delta/d]$$

$$\leq |S^{(K+1)}|\mathbb{P}[|Z_{jl}^{(K+1)}| \geq \delta/d]$$

$$\leq d\mathbb{P}[|Z_{jl}^{(K+1)}| \geq \delta/d].$$

We then obtain (120a) by setting $\varepsilon = \delta/d$ in the Hoeffding's bound (115),

$$\mathbb{P}\big[\big\|\big\|Q_{[S^{(K+1)}]^c S^{(K+1)}}^{(K+1),S} - \bar{Q}_{[S^{(K+1)}]^c S^{(K+1)}}^{(K+1),S}\big\|\big\|_\infty \geq \delta\big] \leq d^2\mathbb{P}[|Z_{jl}^{(K+1)}| \geq \delta/d]$$

$$\leq 2\exp\left(-\frac{\varepsilon^2 n^{(K+1)}}{32d^2} + 2\log(d)\right).$$

Analogously for (120b), we have

$$\mathbb{P}\big[\big\|\big\|Q_{S^{(K+1)} S^{(K+1)}}^{(K+1)} - \bar{Q}_{S^{(K+1)} S^{(K+1)}}^{(K+1)}\big\|\big\|_\infty \geq \delta\big] = \mathbb{P}\big[\max_{j \in S^{(K+1)}} \sum_{k \in S^{(K+1)}} |Z_{jl}^{(K+1)}| \geq \delta\big]$$

$$\leq d\mathbb{P}\big[\sum_{k \in S^{(K+1)}} |Z_{jl}^{(K+1)}| \geq \delta\big]$$

$$\leq d\mathbb{P}[\exists k \in S^{(K+1)}||Z_{jl}^{(K+1)}| \geq \delta/d]$$

$$\leq d^2\mathbb{P}[|Z_{jl}^{(K+1)}| \geq \delta/d]$$

$$\leq 2\exp\left(-\frac{\delta^2 n^{(K+1)}}{32d^2} + 2\log(d)\right).$$

To prove (120c), we have

$$\big\|\big\|(Q_{S^{(K+1)} S^{(K+1)}}^{(K+1)})^{-1} - (\bar{Q}_{S^{(K+1)} S^{(K+1)}}^{(K+1)})^{-1}\big\|\big\|_\infty$$

$$= \big\|\big\|(\bar{Q}_{S^{(K+1)} S^{(K+1)}}^{(K+1)})^{-1}[\bar{Q}_{S^{(K+1)} S^{(K+1)}}^{(K+1)} - Q_{S^{(K+1)} S^{(K+1)}}^{(K+1)}](Q_{S^{(K+1)} S^{(K+1)}}^{(K+1)})^{-1}\big\|\big\|_\infty$$

$$\leq \sqrt{d}\big\|\big\|(\bar{Q}_{S^{(K+1)} S^{(K+1)}}^{(K+1)})^{-1}[\bar{Q}_{S^{(K+1)} S^{(K+1)}}^{(K+1)} - Q_{S^{(K+1)} S^{(K+1)}}^{(K+1)}](Q_{S^{(K+1)} S^{(K+1)}}^{(K+1)})^{-1}\big\|\big\|_2$$

$$\leq \frac{\sqrt{d}}{C_{\min}^{(K+1)}}\big\|\big\|\bar{Q}_{S^{(K+1)} S^{(K+1)}}^{(K+1)} - Q_{S^{(K+1)} S^{(K+1)}}^{(K+1)}\big\|\big\|_2\big\|\big\|(Q_{S^{(K+1)} S^{(K+1)}}^{(K+1)})^{-1}\big\|\big\|_2,$$

where the sub-multiplicative property $\|\|AB\|\|_2 \leq \|\|A\|\|_2\|\|B\|\|_2$ for matrices $A, B$ is used for the last line, and Assumption 4.5 is also applied. Then using the bound (116) in the proof of Lemma F.2, we get

$$\mathbb{P}\big[\big\|\big\|(Q_{S^{(K+1)} S^{(K+1)}}^{(K+1)})^{-1}\big\|\big\|_2 \geq \frac{2}{C_{\min}^{(K+1)}}\big] \leq 2\exp\left(-\frac{\delta^2 n^{(K+1)}}{32d^2} + 2\log(d)\right),$$

and

$$\mathbb{P}\big[\big\|\big\|Q_{S^{(K+1)} S^{(K+1)}}^{(K+1)} - \bar{Q}_{S^{(K+1)} S^{(K+1)}}^{(K+1)}\big\|\big\|_2 \geq \delta/\sqrt{d}\big] \leq 2\exp\left(-\frac{\delta^2 n^{(K+1)}}{32d^3} + 2\log(d)\right)$$

So we have

$$\mathbb{P}\left(\big\|\big\|(Q_{S^{(K+1)} S^{(K+1)}}^{(K+1)})^{-1} - (\bar{Q}_{S^{(K+1)} S^{(K+1)}}^{(K+1)})^{-1}\big\|\big\|_\infty \geq \delta\right) \leq 4\exp\left(-B_1\frac{n^{(K+1)}\delta^2}{d^3} + B_2\log(d)\right),$$

where $B_1, B_2$ are some positive constants. $\qquad \square$