# OpenReview forum: "Meta Learning for Support Recovery of High-Dimensional Ising Models"
_TMLR — Accepted by TMLR_

### Review · Reviewer_Dgbh · 2024-04-03

**Summary Of Contributions:**

The submission considers meta learning in the context of Ising model learning.

The major contribution is on theoretical side. Specifically, the proposed method is shown to recover the support unition and also the support of a novel task with high probability. Finite-sample results are also provided, showing comparable and improved complexity.

Empirically, both synthetic and real-world results are provided.

**Audience:**

Yes

**Claims And Evidence:**

Yes

**Requested Changes:**

Please see the two suggestions in the weakness section.

**Strengths And Weaknesses:**

Strengths

1. The submission is clearly written and easy to follow.

2. The theoretical analysis is thorough.

3. The assumptions while might be strong are clearly discussed with clear motivation.

4. Both synthetic and empirical results are provided supporting the proposed method.

Weaknesses

I find the empirical results are lacking a little.

1. For the real-world experiment, one naively solution is to use all the data and pretend they are all for the novel task. This can be a competing method to try.
2. How many random seeds have we tried in the synthetic experiments. Can we also provide the std of the results or a histogram over different seeds? It would be great to show that the improvement of the proposed method is significant.

---

> ### Author Response · Authors · 2024-07-08
>
> > ... Finite-sample results... showing comparable and improved complexity.
> >
> > 1. The submission is clearly written and easy to follow.
> >
> > 2. The theoretical analysis is thorough.
> >
> > 3. The assumptions... are clearly discussed with clear motivation.
>
> We thank the reviewer for the appreciation of our work.
>
> > 1. For the real-world experiment, one naively solution is to use all the data and pretend they are all for the novel task. This can be a competing method to try.
>
> On real-world experiments, we initially did not consider the novel task. We only initially considered the estimation of the support union from auxiliary tasks. Having said that, we took this as an opportunity to improve all of our experiments.
>
> For both synthetic and real-world experiments, we are now comparing our meta-learning support union result against a multi-task alternative. For the novel task, besides using the single-task method of (Ravikumar et al 2010) on the novel task data only, we also follow the reviewer's suggestion of using all data together (i.e., auxiliary tasks and novel task).
>
> Note that using all data together might be good for estimating the support union, but not for estimating the support of the novel task, which is a subset of the support union. Please see Appendix B.1.
>
> We have replicated a similar conclusion on real-world data. Please see Appendix B.2.
>
> > 2. How many random seeds have we tried in the synthetic experiments. Can we also provide the std of the results or a histogram over different seeds? It would be great to show that the improvement of the proposed method is significant.
>
> We run 100 repetitions in our experiments. Our initial submission had these details on Appendix B.1. We have added the number of repetitions on the main text and all figures. We have also added standard error bars.

---

### Review · Reviewer_DpYs · 2024-04-04

**Summary Of Contributions:**

This paper studies the sample complexity of meta-learning of signed-support recovery of Ising models.

Given a group of Ising models whose parameters are sampled independently from a probability distribution (as defined in Definition 3.1), the paper studies how many samples are needed from each model in order to (a) estimate the "common parameter vector" of the models and (b) estimate the signed-support vector of a new independent copy of the model given enough samples from it.

The paper also shows that the proved sample complexities are minimax-optimal.

**Audience:**

Yes

**Broader Impact Concerns:**

Not applicable.

**Claims And Evidence:**

Yes

**Requested Changes:**

The paper is acceptable from my point of view, but I'd request some clarifications to improve readability.

* Please explain what you mean by *improper* estimation.
* At the top of Section 3.1, it is claimed that your generative model is "more reasonable and flexible than the deterministic settings in previous work." Why? Specifically, why is this a *reasonable* model? Please elaborate.
* After Definition 3.1, you immediately talk about the "novel task." But it becomes clear much later what exactly is this novel task (I.e., estimating the support of the K+1-th Ising model). It seems you're using the words "Ising model" and "task" interchangeably, which is confusing.
* There are a few undefined (but guessable) matrix notation. Please define these: $\overline{Q}_{S_r S_r}, \Lambda, |||\cdot|||, \bigodot$.
* Writing $\overline{Q}\_{S S}$ for $\overline{Q}\_{S_rS_r}$ is extremely confusing.
* Is Assumption 4.1 required for *all* nodes $r$? Please clarify.
* Assumption 4.3 is unnatural and hard to check for an arbitrary model. Can you simplify it or provide conditions under which it holds?
* Using $C^*$ as a parameter name is unfortunate. Some people may confuse $*$ with multiplication.
* The results on real-world data experiments were quite surprising to me, as it doesn't seem to satisfy the assumptions you have made. Perhaps that section can be expanded and maybe you can explain what are some other cases where your method works although some of the assumptions do not hold.
* In your main theorems, when you write, for instance, "the estimated signed neighborhood correctly includes all edges such that etc.," perhaps you should add that these edges are included *with the correct sign*. (These are signed sets, so inclusion is not really well-defined for them.)

**Strengths And Weaknesses:**

Strengths:
* This is a natural problem.
* The assumptions are stated clearly.
* The paper proves matching lower and upper bounds for the sample complexity.
* Experimental evidence is provided to support the theoretical results.
* The paper can motivate studying other Markov random fields and graphical models.

Weaknesses:
* Some of the notation is undefined.
* Some assumptions are written in a convoluted way or are hard to check.

---

> ### Author Response · Authors · 2024-07-09
> **Response 1/2**
>
> > The paper also shows that the proved sample complexities are minimax-optimal.
> >
> > - This is a natural problem.
> >
> > - The assumptions are stated clearly.
> >
> > - The paper can motivate studying other Markov random fields and graphical models.
>
> We thank the reviewer for the appreciation of our work.
>
> > - Please explain what you mean by improper estimation.
>
> The introduction stated "We also propose an improper estimation method in the meta learning problem for Ising model selection where we pool all the samples from the auxiliary tasks together to estimate a single common parameter vector (see Definition 3.1)"
>
> After Equation (11) we mentioned "Note that (10) is an improper estimation as we estimate a single parameter vector using data from different distributions."
>
> That is, we estimate a single parameter $\theta$ although we know that each of the auxiliary tasks $k=1,\dots,K$ has its own true parameter $\bar{\theta}^{(k)}$. Since we are only interested on the joint support of the auxiliary tasks, this improper estimation does not need to estimate $K$ parameters instead.
>
> > - At the top of Section 3.1, it is claimed that your generative model is "more reasonable and flexible than the deterministic settings in previous work." Why?...
>
> We have added "on multi-task learning (Guo et al 2015)" in that sentence, and added more discussion after Assumption 4.2, and after Assumption 4.6: Our assumptions made only twice (on the true common parameter $\bar{\theta}$ and the novel task parameter $\bar{\theta}^{(K+1)}$) for meta-learning, are similar to assumptions made *for all tasks* in multi-task learning (Guo et al 2015, Annals of Applied Statistics). Thus, our assumptions are less restrictive.
>
> Remark 3.2 is also relevant to this point, as it stated "... Suppose on the contrary we do not impose condition (9), then it will be hard for us to estimate a common parameter or a support union useful for all the tasks. On the other hand, there is still great flexibility in the family of distributions since graphs from different tasks can have edge structures with no intersection with arbitrary probability, and we do not assume entries in $\Delta^{(k)}$ to be small in absolute value."
>
> The original submission also included the following paragraph "*Additional Assumptions on $\{\Delta^{(k)}\}_{1\leq k \leq K}$.* The success of our method also relies on some reasonable and flexible assumptions on the centering of the random variables $\{\Delta^{(k)}\}_{1\leq k \leq K}$ underlying the parameters of each task --- reasonable in the sense that the tasks are similar enough to provide useful information, and flexible so that there is as little inductive bias as possible."
>
> > - After Definition 3.1, you immediately talk about the "novel task." But it becomes clear much later what exactly is this novel task (i.e., estimating the support of the K+1-th Ising model). It seems you're using the words "Ising model" and "task" interchangeably...
>
> A task in this manuscript is not "the task of estimating the support of an Ising model" as the reviewer might seem to be understanding.
>
> Our use of the term "task" is consistent with prior literature in the sense that task refers to the parameter and the data. More specifically, a task consist of: a graph with edge set $E^{(k)}$, an Ising model parameterized by $\bar{\theta}^{(k)}$ with support defined by $E^{(k)}$, and samples coming from the probability distribution defined by the Ising model parameterized by $\bar{\theta}^{(k)}$. Please see Definition 3.1.
>
> Having said that, we found few instances that needed correction:
> - true support for the novel task -> true support of the novel task parameter
> - tasks to have only 2 edges -> tasks parameter supports to have only 2 edges
> - novel task of Ising model selection with parameter -> novel task with parameter
> - perform restricted novel task estimation -> estimate the novel task parameter
>
> All the references to Ising models are correct in our manuscript.

---

> > ### Comment · Reviewer_DpYs · 2024-07-09
> > **Response to Response 1/2**
> >
> > > That is, we estimate a single parameter although we know that each of the auxiliary tasks has its own true parameter. Since we are only interested in the joint support of the auxiliary tasks, this improper estimation does not need to estimate parameters instead.
> >
> > This is useful, thanks. Please add these sentences to the paper.
> >
> > > A task in this manuscript is not "the task of estimating the support of an Ising model" as the reviewer might seem to understanding.
> >
> > Since other readers may also get confused by this, please add a short discussion about this in the paper to avoid confusion.

---

> ### Author Response · Authors · 2024-07-09
> **Response 2/2**
>
> > - ... Please define these: $\overline{Q}\_{S_rS_r}$, $\Lambda$, $|||\cdot|||$, $\odot$.
> >
> > - Writing $\overline{Q}\_{SS}$ for $\overline{Q}\_{S_rS_r}$ is extremely confusing.
>
> In our initial submission, we stated "The notations to be used throughout the
> paper is summarized in Table 1 in Appendix A." Here we have $\Lambda_{\min}(A)$, $\Lambda_{\max}(A)$, $|||A|||_\infty$, $|||A|||_2$ and $A \odot B$ are defined in Appendix A. We could gladly move Table 1 to the main text, if required.
>
> $S$ and $S_r$ are defined after equation (9). We use $\overline{Q}\_{SS}$ as a shorthand notation of ${(\bar{Q}_r)}\_{S_r S_r}$. Similarly, we use $\bar{Q}^{(K+1)}\_{S^{(K+1)}S^{(K+1)}}$ as a shorthand notation of ${(\bar{Q}^{(K+1)}_r)}\_{S^{(K+1)}_r S^{(K+1)}_r}$.
>
> We now use shorthand notations only in the appendix, since the proofs can become too long.
>
> > - Is Assumption 4.1 required for all nodes $r$? Please clarify.
>
> We have clarified this by removing the shorthand notation.
>
> > - Assumption 4.3 is unnatural and hard to check for an arbitrary model. Can you simplify it or provide conditions under which it holds?
>
> In our initial submission, we included an illustrative example (for 3 nodes, and then extended it to high dimensions). The comment was somewhat hidden inside a remark. We have given its own title now: "Illustrative Example". Please see Page 8 and Appendix C.
>
> > - Using $C^*$ as a parameter name is unfortunate. Some people may confuse $*$ with multiplication.
>
> We replaced $C^*$ with $C'$.
>
> > - The results on real-world data experiments were quite surprising to me, as it doesn't seem to satisfy the assumptions you have made. Perhaps that section can be expanded and maybe you can explain what are some other cases where your method works although some of the assumptions do not hold.
>
> To give more context, we have now included F1-scores for support union recovery (0.8869) and novel task support recovery (0.6228). While these results are better than alternatives (see Appendix B.2), we believe that the fact that some assumptions might not hold make our results a bit far from 100% on real-world data.
>
> We believe that the superiority of improper estimation of the support union, comes from using all the samples for estimating a single parameter, instead of estimating $K$ parameters. As we mentioned in the introduction "Note that there is an intrinsic difference between multi-task learning and meta learning, where multi-task learning is learning one model for each of the different $K$ tasks simultaneously while for meta learning we learn a single model from different tasks for the easier learning of a novel task. The challenging situation of having many tasks but only a few samples per task (e.g., $K=5000$ tasks each with 2 samples) would also render the multi-task learning method meaningless, as each task cannot be learned individually with these few samples per task."
>
> > - ... "the estimated signed neighborhood correctly includes all edges such..." perhaps you should add that these edges are included with the correct sign.
>
> Corrected.

---

> > ### Comment · Reviewer_DpYs · 2024-07-09
> > **Response to Response 2/2**
> >
> > > We could gladly move Table 1 to the main text, if required.
> >
> > Yes, I'd certainly prefer that. The general rule is: if a notation is used in the paper body, it should be defined in the paper body. I'd also advise against using the shorthand notation, as there are already a lot of different notations to keep track of.
> >
> > I am happy with the other changes and clarifications you have made. Thanks!

---

> ### Author Response · Authors · 2024-07-10
>
> We have added a paragraph at the beginning of Section 3.2 regarding improper estimation.
>
> We have added a remark after Definition 3.1 regarding tasks.
>
> We have moved Table 1 to the main text. We now use shorthand notations only in the appendix, since the proofs can become too long.

---

### Review · Reviewer_PgBo · 2024-06-26

**Summary Of Contributions:**

This paper proposes to use meta learning to estimate the graphs for high-dimensional Ising models. Specifically, a generative model is used to produce random samples of the Ising model parameters.
An Improper Estimation Method is used which involves two steps. The first step is to estimate the Support Union from $K$ existing tasks, and the second step is to estimate the Support of the novel task.

Theoretical results are provided on the sample complexity of the method with a matching lower bound, which indicates the algorithm is minimax optimal.
Furthermore, it is also guaranteed that when the novel task is restricted to the estimated support union, the sample complexity is reduced, depending only on the max neighborhood size instead of the graph size.

**Audience:**

Yes

**Broader Impact Concerns:**

No separate Broader Impact section is found.

But in the Concluding Remarks section some discussion of future extensions of this work is dicussed.

**Claims And Evidence:**

Yes

**Requested Changes:**

1. Discuss the existing theoretical results on meta learning, see an example related work list below. Also, discuss how the results in these works compare to yours.

[1] Giulia Denevi, Carlo Ciliberto, Dimitris Stamos, and Massimiliano Pontil. "Learning to learn around a common mean". NeurIPS, 2018.

[2] Ron Amit and Ron Meir. "Meta-learning by adjusting priors based on extended PAC-Bayes theory". ICML, 2018.

[3] Katelyn Gao and Ozan Sener. "Modeling and optimiza- tion trade-off in meta-learning". NeurIPS, 2020.

[4] Alec Farid and Anirudha Majumdar. "Generalization bounds for meta-learning via PAC-Bayes and uniform stability". NeurIPS, 2021.

[5] Lisha Chen and Tianyi Chen. "Is Bayesian Model-Agnostic Meta Learning Better than Model-Agnostic Meta Learning, Provably?". AISTATS, 2022.

[6] Jiechao Guan, Liu Yong, and Lu Zhiwu. "Fine-grained analysis of stability and generalization for modern meta learning algorithms". NeurIPS, 2022.

2. Discuss the relation of the proposed generative model with Bayesian meta learning.

3. Discuss when the assumptions in Section 4.1.2 can hold.

**Strengths And Weaknesses:**

## Strengths

Overall, the paper is well written with clearly presented theoretical results.

Experiments on real-world data are performed to validate the theories.

## Weaknesses

1. Although there are sufficient discussions on the related works of Ising models, very limited discussion is provided on the theoretical results of meta learning, which also provide sample complexity bounds.
It would be better if a comparison can be made for the results in this paper and some other related theoretical results of meta learning.



2. The generative model essentially models the distribution of the Ising model parameters. How is the generative model related to Bayesian meta learning, which also models the distribution of the task-specific parameters?


3. It would be better if the experiments in Figures 1 and 2 can perform on a wider range of $p$ and $d$ to validate the dependence on $p$ and $d$ for the theoretical sample complexity.

4. It would be better if some discussions can be provided on when the assumptions in Section 4.1.2 can hold.

---

> ### Author Response · Authors · 2024-07-10
>
> > Overall, the paper is well written with clearly presented theoretical results.
>
> We thank the reviewer for the appreciation of our work.
>
> > 1. Although there are sufficient discussions on the related works of Ising models, very limited discussion is provided on the theoretical results of meta learning, which also provide sample complexity bounds...
> >
> > Change 1. Discuss the existing theoretical results on meta learning... discuss how the results in these works compare to yours.
>
> In Page 2, we have included discussions of the papers brought up by the reviewer and some additional papers as well:
>
> "Meta learning has been widely used in machine learning problems to help increase sample efficiency, but a majority of prior works are experimental in nature, without theoretical guarantees (Lake et al 2015; Lemke et al 2015; Vinyals et al 2016; Ravi & Larochelle 2017; Finn et al 2017; Snell et al 2017; Grant et al 2018; Yoon et al 2018; Hospedales et al 2022). There have been some efforts on building the theoretical foundation of meta-learning, but they only pertain generalization bounds in learning theory (Maurer 2005; Pentina & Lampert 2014; Amit & Meir 2018; Denevi et al 2018; Khodak et al 2019; Huang et al 2020; Tripuraneni et al 2020; Farid & Majumdar 2021; Chen & Chen 2022; Guan et al 2022) and convergence rates in optimization (Fallah et al 2020; Finn et al 2019; Khodak et al 2019; Gao & Sener 2020). In the above works, performance is only viewed in terms of risks (e.g., misclassification rate, mean squared error) and not in terms of support recovery. For Markov random fields..."
>
> > 2. The generative model essentially models the distribution of the Ising model parameters. How is the generative model related to Bayesian meta learning, which also models the distribution of the task-specific parameters?
> >
> > Change 2. Discuss the relation of the proposed generative model with Bayesian meta learning.
>
> We have added this discussion at the end of the introduction:
>
> "Finally, we want to point out that, as part of our contribution, we define a Bayesian generative model (see Definition 3.1). Bayesian meta learning has been previously studied (Grant et al 2018; Yoon et al 2018) but the focus is on proposing general problem-independent algorithms, in which a marginal likelihood (with respect to a common parameter) is approximated by estimating the parameter of each auxiliary task.
> In contrast, we avoid estimating each auxiliary task parameter. Furthermore, our main contribution is on theoretical guarantees of our procedure to efficiently recover the support union from auxiliary tasks through improper estimation, and to then estimate the parameter of the novel task, both with significantly better sample complexities."
>
> > 3. It would be better if the experiments in Figures 1 and 2 can perform on a wider range of $p$ and $d$...
>
> First, note that the $d^3 \log p$ factor (after removing the favorable dependence on $K$ in our paper) has also been observed in the single-task problem (Ravikumar et al 2010, Annals of Statistics) and the multi-task problem (Guo et al 2015, Annals of Applied Statistics).
>
> Instead of $p \in$ {$50,75,100$}, we are now using $p \in$ {$50,100,200$}. Note that the number of parameters to be estimated is in the order of $p^2$, thus the dimension are actually {$2500,10000,40000$}. We have also added comparisons for different degrees $d \in$ {$3,5,7$}.
>
> > 4. It would be better if some discussions can be provided on when the assumptions in Section 4.1.2 can hold.
> >
> > Change 3. Discuss when the assumptions in Section 4.1.2 can hold.
>
> We have added some discussion after Assumption 4.2, and after Assumption 4.6: Our assumptions made only twice (on the true common parameter $\bar{\theta}$ and the novel task parameter $\bar{\theta}^{(K+1)}$) for meta-learning, are similar to assumptions made *for all tasks* in multi-task learning (Guo et al 2015, Annals of Applied Statistics). Thus, our assumptions are less restrictive.
>
> Furthermore, the assumptions are on the population Fisher information matrix, not on the sample Fisher information matrix.
>
> > No separate Broader Impact section is found.
>
> We have added a "Broader Impact" section stating "This theoretical work does not present any foreseeable societal consequence."

---

### Decision · Action_Editor_S6UA · 2024-08-06

**Recommendation:** Accept as is

**Comment:**

The reviewers all appreciate the contributions of the paper.

The authors and the reviewers had a good interaction that helped improve the paper.

**Audience:**

Yes.

**Claims And Evidence:**

Yes.